# ZBTB16/PLZF regulates juvenile spermatogonial stem cell development through an extensive transcription factor poising network

Chongil Yi [1], Yuka Kitamura [2], So Maezawa [3], Satoshi H. Namekawa [2] & Bradley R. Cairns [1]✉

Spermatogonial stem cells balance self-renewal with differentiation and spermatogenesis to ensure continuous sperm production. Here, we identify roles for the transcription factor zinc finger and BTB domain-containing protein 16 (ZBTB16; also known as promyelocytic leukemia zinc finger (PLZF)) in juvenile mouse undifferentiated spermatogonia (uSPG) in promoting self-renewal and cell-cycle progression to maintain uSPG and transit-amplifying states. Notably, ZBTB16, Spalt-like transcription factor 4 (SALL4) and SRY-box transcription factor 3 (SOX3) colocalize at over 12,000 promoters regulating uSPG and meiosis. These regions largely share broad histone 3 methylation and acetylation (H3K4me3 and H3K27ac), DNA hypomethylation, RNA polymerase II (RNAPol2) and often CCCTC-binding factor (CTCF). Hi-C analyses show robust three-dimensional physical interactions among these cobound promoters, suggesting the existence of a transcription factor and higher-order active chromatin interaction network within uSPG that poises meiotic promoters for subsequent activation. Conversely, these factors do not notably occupy germline-specific promoters driving spermiogenesis, which instead lack promoter–promoter physical interactions and bear DNA hypermethylation, even when active. Overall, ZBTB16 promotes uSPG cell-cycle progression and colocalizes with SALL4, SOX3, CTCF and RNAPol2 to help establish an extensive and interactive chromatin poising network.

Murine spermatogonia (SPG)-derived stem cells continuously balance self-renewal and differentiation, orchestrated by multiple transcription factors (TFs). In the mouse adult testis, undifferentiated SPG (uSPG) have mesenchymal stem cell (MS cell)-like properties and forage for limited cytokines[1–4]. In vitro systems to study uSPG include primary cultures of uSPG (PC-uSPG) that, at low frequency, lose MS cell-like cell identity during in vitro expansion and instead acquire embryonic stem cell (ES cell)-like cell properties (with an active pluripotency network)[5,6]. Here, we aimed to understand how TF networks regulate uSPG and their differentiation and began our studies with zinc finger and BTB domain-containing protein 16 (ZBTB16; also known as promyelocytic leukemia zinc finger (PLZF)), which is specifically expressed in undifferentiated A-single, A-paired and A-aligned uSPG[7] and A1 differentiating SPG (dSPG)[8,9]. ZBTB16 is not essential for spermatogenesis but rather helps ensure long-term uSPG maintenance, as its absence results in the gradual of depletion of uSPG in a substantial

[1]Howard Hughes Medical Institute, Huntsman Cancer Institute, Department of Oncological Sciences, University of Utah School of Medicine, Salt Lake City, UT, USA. [2]Department of Microbiology and Molecular Genetics, University of California, Davis, CA, USA. [3]Department of Applied Biological Science, Faculty of Science and Technology, Tokyo University of Science, Chiba, Japan. ✉e-mail: brad.cairns@hci.utah.edu

subset of seminiferous tubules, especially during aging[7,10,11]. Previous studies used microarrays or small interfering RNA-mediated knockdown to explore ZBTB16 function in uSPG or PC-uSPG[7,12]. In both testis and blood development, ZBTB16 functions as both an activator and a repressor[10,12–15] but the mechanisms remain incompletely understood. Repression involves ZBTB16 recruiting Polycomb repressive complexes 1 and 2 (PRC1 and PRC2) and interaction between B cell-specific Moloney murine leukemia virus integration site 1 (BMI1) and histone deacetylases (HDACs)[16,17], while activation entails interaction with both chromatin and post-translational modifiers[15,18]. Here, we initially examined the relationship between ZBTB16 and two other uSPG TFs, SRY-box TF 3 (SOX3) and Spalt like TF 4 (SALL4), in establishing the chromatin and transcription landscape of uSPG.

Human and mouse spermiogenesis (postmeiotic differentiation) exhibits a chromatin transcription logic that greatly differs from somatic cells. Mitotic SPG and meiotic cells predominantly use 'typical promoters' with active chromatin modifications, while spermiogenesis primarily relies on 'atypical promoters' that bear repressive chromatin modifications such as DNA methylation (DNAme) and histone 3 methylation (H3K27me3) during their transcription, alongside active modifications such as histone acetylation (H3K27ac)[19,20]. Here, we examined the logic and mechanisms underlying the establishment of active chromatin at typical promoters in uSPG. Remarkably, ZBTB16, SALL4 and SOX3, along with several positive epigenetic marks (H3K4me2/3, H3K27ac and DNA hypomethylation), co-occupy >12,000 typical promoters that coincide with open higher-order chromatin architectures in uSPG. This encompasses the genes active in uSPG, as well as the vast majority of genes later activated in dSPG and during meiosis, suggesting that this TF network establishes chromatin poising in uSPG for both differentiation and meiotic activation. We further dovetail our work with recent work of others reporting the presence of paused RNA polymerase II (RNAPol2) at uSPG genes[21], which we show resides at typical promoters. Functionally, we reveal that ZBTB16 helps activate a portion of typical network genes to help regulate uSPG self-renewal, MS cell-like cell identity and uSPG progression by promoting cell-cycle progression.

## Results

### Defining ZBTB16-occupied and affected genes in vivo
Prior work characterized ZBTB16-binding sites and affected genes in PC-uSPG rather than in cells directly isolated from the testis[12]. To characterize ZBTB16 in vivo, we isolated uSPG directly from the testis and conducted RNA sequencing (RNA-seq) and histone modification chromatin immunoprecipitation followed by deep sequencing (ChIP-seq). ZBTB16 ChIP-seq was also performed from the whole testis as ZBTB16 expression is limited to uSPG at P7 (refs. 1,7). Using immunomagnetic cell sorting (IMCS) with anti-THY1 and anti-KIT antibodies[1,22], we isolated uSPG and dSPG from wild-type (WT) and *Zbtb16*[lu/lu] (null) mice at P7. Additionally, to study later stages of germ cell development, we used the STA-PUT method[23,24] to isolate pachytene spermatocytes (PSs) and round spermatids (RSs) (Fig. 1a). Our RNA-seq approach involved three replicates with high similarity (Extended Data Fig. 1a). THY1+-sorted and KIT+-sorted SPG were highly enriched for *Thy1* or *Kit* mRNA, verifying the utility of the IMCS approach (Extended Data Fig. 1b). When comparing THY1+ uSPG from WT and null mice, we found 739 genes upregulated and 695 genes downregulated in *Zbtb16*-deficient THY1+ uSPG (Fig. 1b).

We next identified ZBTB16-occupied loci by ChIP-seq, using biological duplicates and two different ZBTB16 antibodies (rabbit or goat), confirmed by immunoblot analysis (Extended Data Fig. 1c). First, the peaks (ChIP signal) occupied by ZBTB16 in the datasets derived from the rabbit and the goat antibodies were numerous and nearly identical ($r$ values ≈ 0.93; Extended Data Fig. 1d). About 40% of ZBTB16-binding sites reside in gene promoters, with another ~40% in distal intergenic regions and ~10% in gene bodies (Extended Data Fig. 1e,f). Although ZBTB16 bound the proximal promoter of 20,116 genes, in our RNA-seq,

only 4.7% of ZBTB16-bound genes were affected in *Zbtb16*-deficient THY1+ uSPG. Within the affected genes, ZBTB16 bound 459 of the 695 downregulated genes (66.0%) and 496 of the 739 upregulated genes (67.1%) (Fig. 1c).

Analysis of affected genes by Gene Ontology (GO) using Panther[25] identified enriched GO terms related to receptors and TFs for development and cell migration or regulation of locomotion (for genes downregulated in the null cells; Fig. 1d) and terms for meiosis and carboxylic acid biosynthetic process (for genes upregulated in the null cells; Fig. 1e). Notably, ZBTB16 activates itself (*Zbtb16*; autoactivation), *T* (*Brachyury*, a TF important for uSPG maintenance regulated by glial cell-derived neurotrophic factor (GDNF) and Ets variant gene 5 (ETV5))[26], *Eomes* (a TF for long-term uSPG maintenance)[11] and *Fgfr1* (a receptor important for uSPG maintenance)[27] (Fig. 1f). In contrast, ZBTB16 binds and attenuates *Krt18*, *Sycp3* (meiosis), *Cth* (metabolism) and *Rhox10* (gonocyte-to-uSPG conversion[28]) (Fig. 1g). The genes indirectly regulated by ZBTB16 are linked to immune response (Extended Data Fig. 2a). Overall, ZBTB16 occupies an interesting set of genes but curiously only impacts a small fraction of its occupied promoters, which is explored further below.

### ZBTB16 occupancy correlates with broad H3K4me3 and H3K27ac
We then examined chromatin attributes correlated with ZBTB16 occupancy in uSPG. In nontestis cells, ZBTB16 represses genes by recruiting PRC1, PRC2 and HDACs[16,17]. Regarding activation, phosphorylation of ZBTB16 promotes transcriptional action, while acetylation of ZBTB16 correlates with repression[18] (Fig. 2a). However, in uSPG, ZBTB16 occupancy at promoters strikingly overlapped with high and broad H3K4me3 (>90% overlap), H3K27ac (~56% overlap) and low DNAme (Fig. 2b–e), regardless of transcriptional status. Overlap with H3K27me3 (~17%) or BMI1 (~7%) was modest and similar between upregulated and downregulated genes (Fig. 2b,d,e and Extended Data Fig. 2b,c). ZBTB16-regulated genes (either upregulated or downregulated) showed no distinct pattern in terms of their initial DNAme or chromatin status (for example, H3K4me3, H3K27me3 or their combination (that is, bivalency)) (Fig. 2d–g). Thus, loss of ZBTB16 does not specifically impact genes bearing a particular chromatin state, including bivalency. Overall, the clear feature in uSPG is that ZBTB16-bound genes uniformly bear high and broad H3K4me3 and lack DNAme and a majority bear H3K27ac.

### ZBTB16 lacks a single consensus DNA-binding site in uSPG
Prior efforts have yielded surprisingly diverse ZBTB16-binding motifs in different cell types[12,13,29]. To identify a motif in uSPG, we used MEME-ChIP[30] and GEM[31] analysis on ZBTB16-bound peaks from the regular ChIP-seq and also ZBTB16-bound footprints from our ChIP-nexus approach[32]. However, MEME-ChIP produced a top-ranked site overlapping with only ~3% of bound peaks, affecting only 15 genes in *Zbtb16*-deficient uSPG (Extended Data Fig. 2d). Notably, GEM analysis from both datasets yielded a telomere motif (Extended Data Fig. 2e, f) but this observation was absent in MEME-ChIP. Thus, ZBTB16 lacks a single defined binding motif in uSPG and may instead be tethered to the DNA with other protein partners (explored below)[17].

### ZBTB16 attenuates particular retrotransposons possessing H3K4me3
Around 40% of ZBTB16-bound regions are associated with distal intergenic areas (Extended Data Fig. 1e,f). Indeed, ZBTB16 repressed long interspersed element (LINE) L1 in the whole testis and bone marrow[33], prompting an examination of transposable elements in the testis. We found that ZBTB16 binds LINEs, as well as particular long terminal repeats (LTRs) and short interspersed elements (SINEs) (Extended Data Fig. 3a). Surprisingly, a high proportion of ZBTB16-occupied

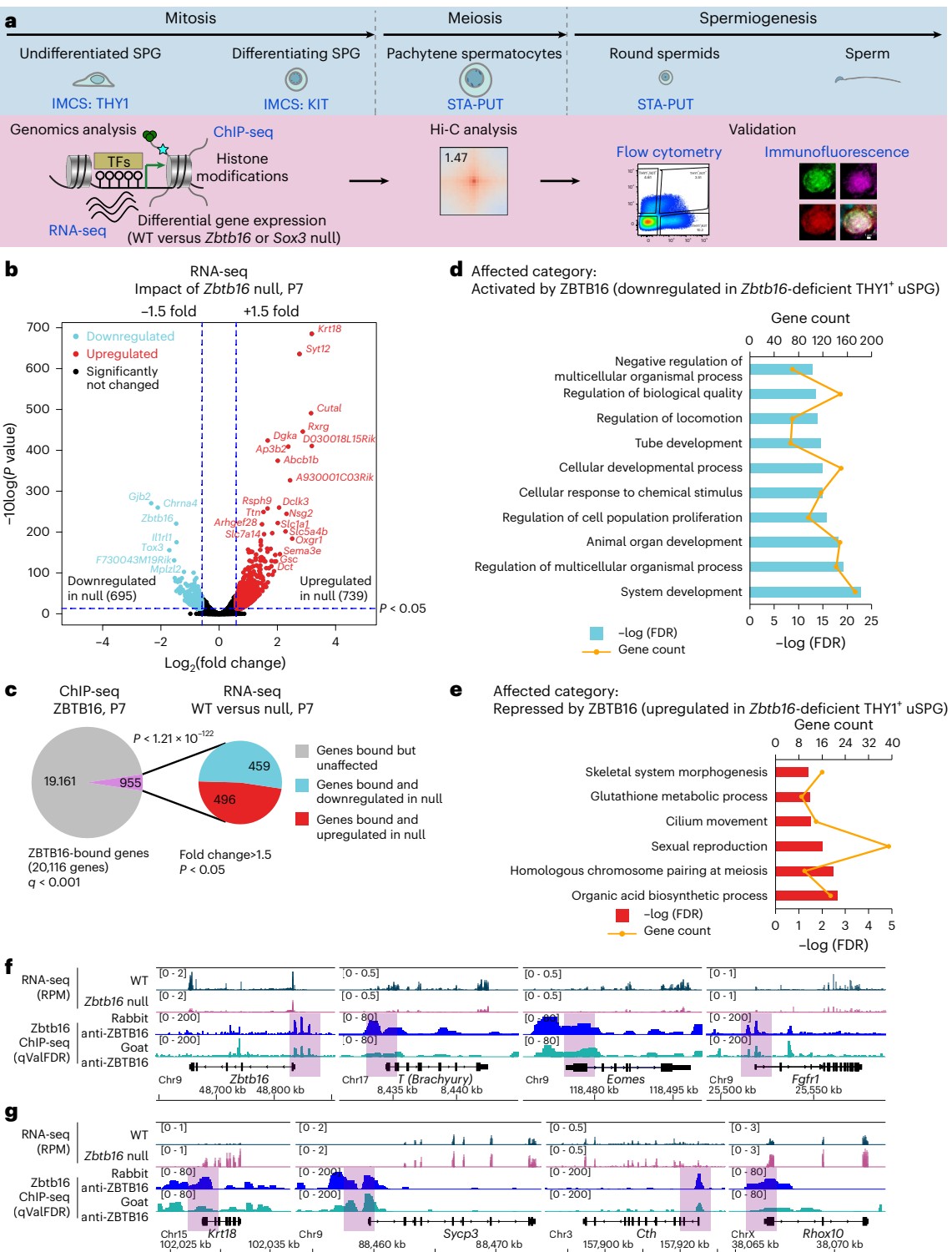

**Fig. 1 | Defining the gene targets of ZBTB16. a**, Summary of experiment flow. Testes were digested into single cells and THY1⁺ uSPG and KIT⁺ dSPG were isolated. **b**, Volcano plot depicting fold changes (log₂) and *P* value (−10log₁₀) of differential expression in *Zbtb16*-deficient THY1⁺ uSPG compared to WT. DEGs are defined by a fold change cutoff of >1.5 and an adjusted *P* value (derived from DESeq2 analysis) of <0.05. Points above the threshold represent significantly upregulated (red) or downregulated (sky blue) genes. Nonsignificant genes are shown in gray. Adjusted *P* values account for multiple comparisons using the Benjamini–Hochberg method. **c**, Left, pie chart illustrating the fraction of ZBTB16-bound sites within ±2 kb of the TSS of the closest RefSeq-annotated transcript. The number of ZBTB16-bound genes that were unaffected in RNA-seq of *Zbtb16*-null cells (gray) compared to those differentially expressed

(pink). Right, pie chart partitioning the number of downregulated (sky blue) or upregulated (red) ZBTB16-bound genes with *Zbtb16*-affected genes from RNA-seq (analyzed as in **b**). Statistical analysis was performed using a hypergeometric test. **d**, GO terms for functional clustering of genes downregulated in *Zbtb16*-null cells associated with ZBTB16 ChIP-seq peaks (top ten categories are shown). **e**, GO terms for functional clustering of genes upregulated in *Zbtb16*-null cells associated with ZBTB16 ChIP-seq peaks. **f,g**, Gene targets of ZBTB16. Browser snapshots displaying genes downregulated (**f**; *Zbtb16*, *T*, *Eomes* and *Fgfr1*) or upregulated (**g**; *Krt18*, *Sycp3*, *Cth* and *Rhox10*) in the null cells. Shown are tracks for RNA-seq from purified THY1⁺ uSPG in WT and *Zbtb16*-null mouse testes at P7 and ZBTB16 ChIP-seq data using anti-rabbit ZBTB16 and anti-goat ZBTB16 antibodies (qValFDR).

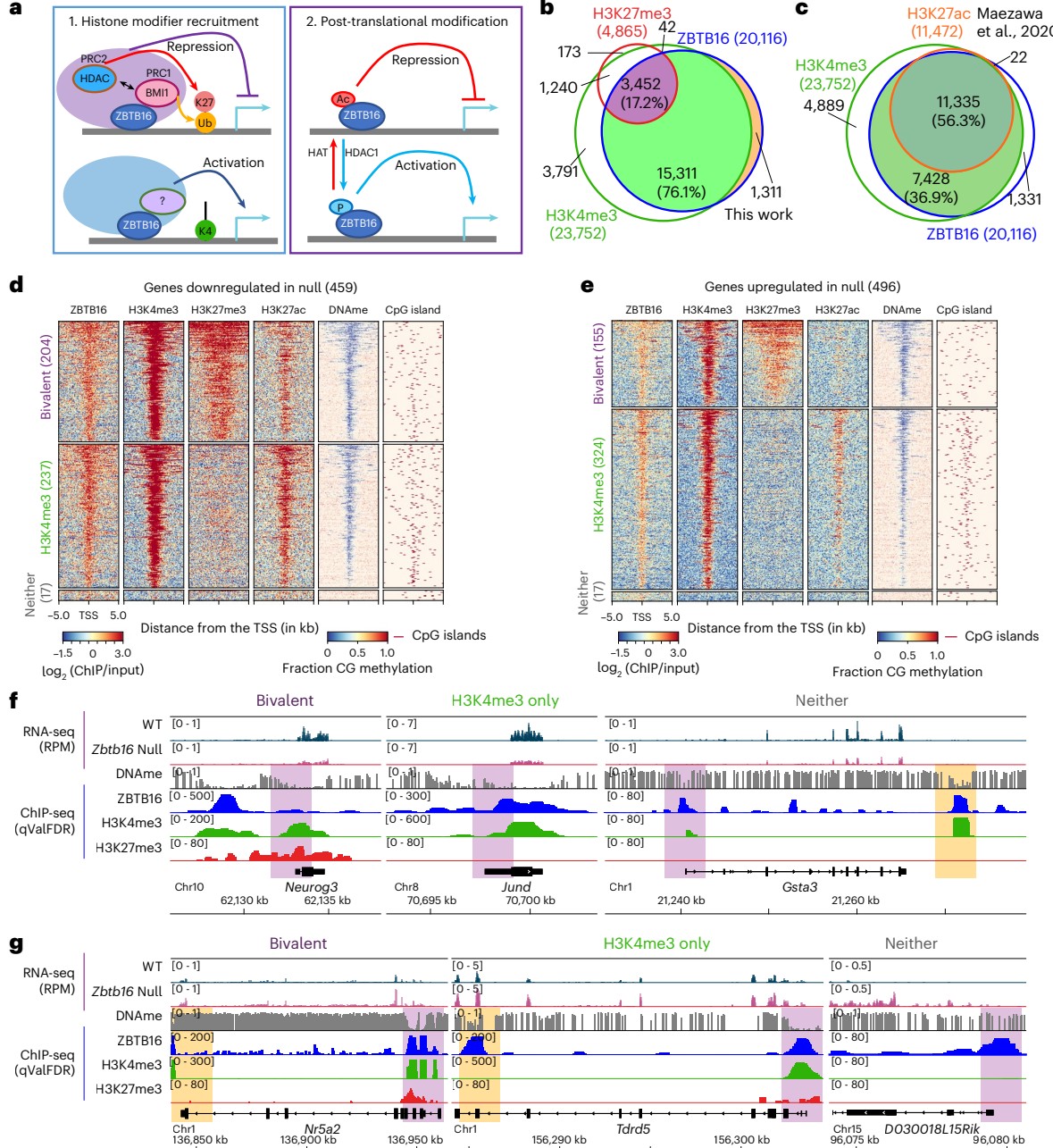

**Fig. 2 | Relationships of ZBTB16 to chromatin modifications. a**, Models for gene regulation by ZBTB16. Left, histone modifier recruitment by ZBTB16. Right, post-translational modifications of ZBTB16. **b**, Venn diagram showing the overlap of ZBTB16-bound genes with H3K4me3 and H3K27me3 in THY1⁺ uSPG (this work). **c**, Venn diagram showing the overlap of ZBTB16-bound genes with H3K4me3 (this work) and H3K27ac (ref. 51) in THY1⁺ uSPG. **d,e**, Heat map showing clustering of active and repressive histone modification at the TSS (±5 kb) of downregulated (**d**) or upregulated (**e**) genes in *Zbtb16*-null cells with DNAme and CpG islands. **f,g**, Genome browser panels showing ZBTB16 target genes, including genes downregulated (**f**; *Neurog3*, *Jund* and *Gsta3*) or upregulated (**g**; *Nr5a2*, *Tdrd5* and *DO30O18L15Rik*) in the null cells, alongside chromatin marks H3K4me3 and H3K27me3 and DNAme. RPM, reads per million mapped reads.

LTRs (46.2%), LINEs (48.3%) and SINEs (57.9%) overlapped with broad H3K4me3 and low DNAme, a pattern similar to that seen with ZBTB16 at RefSeq genes, and only rarely overlapped with H3K9me3, H3K27me3 and Piwi-interacting RNA (piRNA) clusters (Fig. 2b and Extended Data Fig. 3b–d). Thus, in uSPG, ZBTB16 attenuates specific retrotransposons independently of H3K9me3, H3K27me3 and DNA hypermethylation. Overall, ZBTB16 appears to have dual functions (activation and repression) at RNAPol2-transcribed protein-coding genes, whereas it only attenuates retrotransposons. Here, we note that our data leave open the possibility of both direct and indirect mechanisms for ZBTB16-dependent repeat element regulation.

## Limited overlap of ZBTB16 targets between uSPG and PC-uSPG

We then considered using established in vitro culturing approaches to advance our work but first needed to examine whether there are differences in ZBTB16 binding between in vivo (testis-derived) and PC-uSPG. To test this, we compared our *Zbtb16*-affected genes in THY1⁺ P7 uSPG to those reported in ITGA6⁺ uSPG[10] and PC-uSPG[12,14]. Surprisingly, only about 2% (33 genes) of *Zbtb16*-affected genes in THY1⁺ uSPG were shared with those impacted in *Zbtb16*-deficient ITGA6⁺ uSPG (Extended Data Fig. 4a,b). Furthermore, just two of the top ten ZBTB16 targets from PC-uSPG (*Zbtb16* and *Lhx1*) were also affected by ZBTB16 deficiency in vivo. Most genes identified in vitro, such as *Etv5*, *Bcl6b* and *Uchl1*,

remained unaffected in *Zbtb16*-deficient P7 THY1⁺ uSPG (Extended Data Fig. 4c).

Principal component analysis (PCA) revealed striking transcriptome differences between THY1⁺ uSPG and PC-uSPG, with PC-uSPG closely resembling multipotent adult SPG-derived stem cells (MASCs) and ES cells (Extended Data Fig. 5a). Notably, *Zbtb16* expression in THY1⁺ uSPG remained unaltered in PC-uSPG but was silenced in MASCs (Extended Data Fig. 5b). We next compared differential ZBTB16 enrichment at gene promoters in uSPG and PC-uSPG[12] by ChIP-seq using a goat anti-ZBTB16 antibody. ZBTB16 bound 16,852 gene promoters in uSPG but only 8,261 gene promoters in PC-uSPG, a subset of those bound in vivo (Extended Data Fig. 5c). However, only 508 of the genes occupied in PC-uSPG were affected in the *Zbtb16*-null cells (Extended Data Fig. 5c,d). Notably, ZBTB16 occupied and attenuated *Utf1* in uSPG while it remained unoccupied in PC-uSPG, correlated with *Utf1* upregulation (Extended Data Fig. 6e,f), a result validated by examining UTF1 levels in *Zbtb16*-deficient LIN28A⁺ uSPG (Extended Data Fig. 6a,b). Additionally, whereas adult uSPG lack pluripotency factors *Sox2* and *Nanog* (ref. 19), PC-uSPG upregulate these factors, alongside *Myc*, *Lin28a* and *Klf4* (Extended Data Fig. 6c), suggesting a transition to a more ES cell-like state. Lastly, ZBTB16 indirectly attenuated *Pou5f1* (*Oct4*) expression (Extended Data Fig. 2a).

uSPG are migratory, display MS cell charatericstics[1,2] and undergo a mesenchymal-to-epithelial transition at low frequency during uSPG in vitro expansion into PC-uSPG[6]. Indeed, ZBTB16 target genes downregulated in the null were enriched in cell migration categories (Fig. 1d). Moreover, in PC-uSPG, where ZBTB16 occupancy was lowered, cell-migration-related genes were further downregulated (Extended Data Fig. 6d). uSPG appear to migrate by extracellular matrix remodeling, as a subset of LIN28A⁺ uSPG displayed ADAMTS5 metalloproteinase expression (Extended Data Fig. 6e) and cleaved Versican (Versikine) was detected in SOX3⁺ uSPG (Extended Data Fig. 6f). Taken together, ZBTB16-bound sites were greatly reduced and gene expression patterns, including genes for migration, were highly altered in PC-uSPG versus in vivo uSPG. Thus, ZBTB16 occupancy and activity depend on the cell type, developmental stage and growth conditions, requiring investigation of the direct in vivo cellular context.

### dSPG decrease in *Zbtb16*-null mice

Previous studies have suggested that ZBTB16 represses *Kit* (refs. 13,34). Although ZBTB16 occupies the *Kit* promoter in vivo, the loss of *Zbtb16* did not affect *Kit* expression in either THY1⁺ uSPG or KIT⁺ dSPG (Extended Data Fig. 1b). In *Zbtb16*-null mice testes at P7, THY1⁻/KIT⁺ dSPG were decreased ~2.1-fold while the other two populations of uSPG (THY1⁺/KIT⁻ and THY1⁺/KIT⁺) were not significantly changed (Fig. 3a,b and Extended Data Fig. 7a), implying that ZBTB16 promotes the THY1⁺ uSPG-to-KIT⁺ dSPG transition, and does not inhibit *Kit* expression or suppress uSPG differentiation. Thus, ZBTB16 promotes the undifferentiated-to-differentiating SPG transition, termed hereafter U–DT.

### ZBTB16 activates neurogen 3 (*Neurog3*) and cyclin D1 (*Ccnd1*) for SPG differentiation

To understand how ZBTB16 promotes U–DT and supports progenitor development, we examined genes linked to progenitor initiation and cell-cycle regulation. Notably, ZBTB16 directly activated *Neurog3* (Fig. 2h and Extended Data Fig. 7b,c), which promotes uSPG differentiation and is inhibited by high GDNF, thereby preventing uSPG differentiation[35–37]. Furthermore, ZBTB16 directly activated *Ccnd1* (Fig. 3c), which is involved in entry into the cell cycle and $G_1$–S transition, consistent with previous results in *Zbtb16*-knockout (KO) mice[10]. Thus, these findings imply that ZBTB16 antagonizes GDNF in uSPG to promote U–DT and developmental progression.

Next, we performed flow cytometry analysis to evaluate cell-cycle status as a function of DNA content in P7 testicular cells from *Zbtb16*-null and littermate control (WT) mice. While THY1⁺/KIT⁻ uSPG were relatively quiescent (72.57% of $G_0/G_1$ phase cells), THY1⁺/KIT⁺ uSPG (31.20% of $G_0/G_1$ phase cells) and THY1⁻/KIT⁺ dSPG (58.17% of $G_0/G_1$ phase cells) displayed increased activity. Furthermore, $G_0/G_1$ phase cells were enriched in *Zbtb16*-deficient THY1⁺/KIT⁺ uSPG (~1.3-fold) and THY1⁻/KIT⁺ dSPG (~1.4-fold), while fewer $G_0/G_1$ phase cells were observed in *Zbtb16*-deficient THY1⁺/KIT⁻ uSPG at P7 (Fig. 3d). These results imply that ZBTB16 ensures a slow cell cycle in self-renewing early uSPG (THY1⁺/KIT⁻) but helps accelerate the cell cycle in late uSPG (THY1⁺/KIT⁺).

### Delay in spermatocyte development in *Zbtb16*-null mice

We next assessed the impact of delayed U–DT on spermatocyte emergence at P14. *Zbtb16*-null testis growth halted at P56, followed by shrinkage by P70, possibly because of age-related germ cell loss (Extended Data Fig. 7d,e). At P14, testis weight was ~1.6-fold lower in *Zbtb16*-null mice at P14 (Fig. 3e,f and Extended Data Fig. 7d,e). Immunofluorescence (IF) revealed a ~2.1-fold decrease in spermatocytes and a ~3.2-fold decrease in PSs with the XY body displaying H2AX phosphorylation (γH2AX) in *Zbtb16*-null mice (Fig. 3g–i). Notably, LIN28A⁺ uSPG were increased ~1.3-fold in *Zbtb16*-null cells. However, no abnormal γH2AX was detected in *Zbtb16*-deficient uSPG (Fig. 3g), implying that DNA damage response is not triggered in *Zbtb16*-deficient uSPG. Together, spermatocyte reduction and the accumulation of SPG appear to be linked to late uSPG transition delay.

### Distinct cyclin D (CCND)–cyclin-dependent kinase 4 (CDK4) complexes in early and late uSPG

To better understand cell-cycle regulation in uSPG, we examined CCND family proteins and their catalytic partners CDK4 and CDK6. Analysis of RNA-seq data from *Id4*⁻GFP^Bright uSPG, characterized by high transplantation efficiency[38], revealed elevated expression of *Ccnd2* and *Cdk4*, which are associated with uSPG self-renewal, in comparison to *Ccnd1* and *Cdk6*. Additionally, *Ccnd1* and *Cdk6* are known to be dispensable for uSPG self-renewal[38–40] (Extended Data Fig. 7f). Intriguingly, the expression pattern of *Ccnd*, *Cdk4* and *Cdk6* in uSPG differed from PC-uSPG, primordial germ cells (PGCs), ES cells and hematopoietic stem cells (HSCs), suggesting that cell-cycle acceleration varies in cell types and developmental stages (Extended Data Fig. 7g). Here, we hypothesized that CCND2 and CDK4 promote a more rapid cell cycle in early uSPG, while CCND1 and CDK6 enhance $G_0/G_1$–S transition in late uSPG. IF in P14 testes confirmed CCND1 expression in uSPG and dSPG while CCND2 was present in a subset of ZBTB16⁺/CCND1⁺ uSPG, marking early uSPG, but absent in ZBTB16⁻/CCND1⁺ dSPG (Extended Data Fig. 7h). Notably, CCND2⁺/CCND1⁺ early uSPG expressed higher ZBTB16 levels than CCND2⁻/CCND1⁺ TA late uSPG (Extended Data Fig. 7i), suggesting that ZBTB16 gradually decreases from early uSPG to late uSPG.

We next explored how uSPG differently use CDK4 and CDK6 for $G_0/G_1$–S transition. Phosphorylated CDK4 (pCDK4) was detected from uSPG to spermatocytes (Extended Data Fig. 7j). Unexpectedly, pCDK6 was localized in leptotene and PSs but not in SPG (Extended Data Fig. 7k). These results indicate a role of pCDK4 in $G_0/G_1$–S progression from mitosis to meiosis.

### SOX3 promotes SPG differentiation and reduces migration genes

We then returned to the highly unexpected observation above involving ZBTB16 binding at ~20,000 promoters but affecting only a few hundred genes. This prompted consideration of partial redundancy among additional uSPG-specific (for testis) TFs SOX3 and SALL4 (refs. 41,42). This extension was also prompted by the reported coactivation of *Neurog3* by ZBTB16 (Fig. 2f) and SOX3 (refs. 41,43,44), as well as the reported co-occupancy and antagonistic relationship between ZBTB16 and SALL4 (refs. 12,34,42).

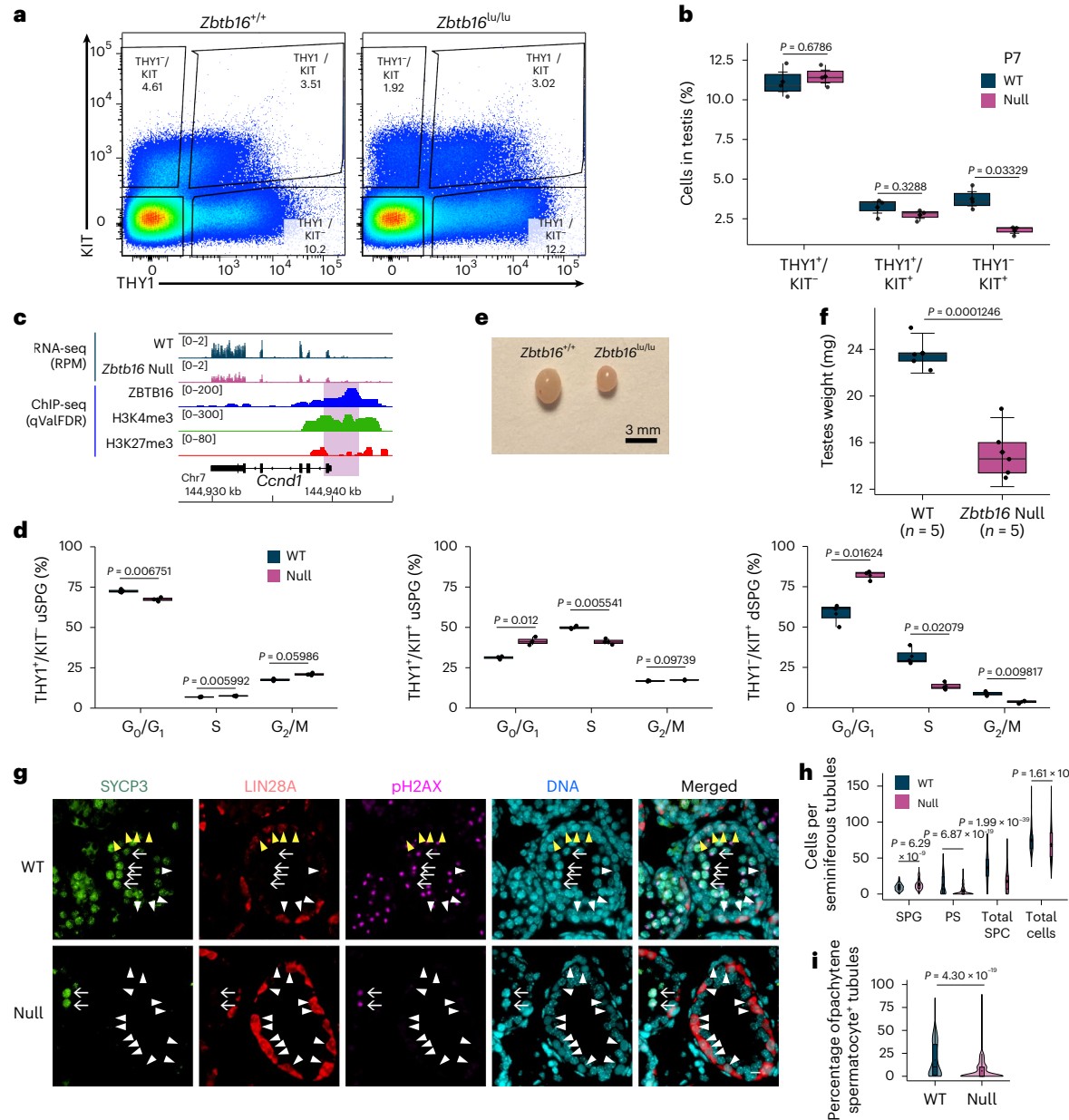

**Fig. 3 | Fewer KIT⁺ dSPG and spermatocytes in *Zbtb16*-null testes.**
**a**, Representative profile of FACS-sorted populations for THY1 (uSPG marker)-positive and/or KIT (dSPG marker)-positive cells from WT and null testicular cells at P7 for cell-cycle analysis (Fig. 3c). **b**, Box-and-whisker plot showing the percentage of THY1⁺/KIT⁻, THY1⁺/KIT⁺ and THY1⁻/KIT⁺ cell populations in the testes of *Zbtb16*-null and littermate control (WT or heterozygotes) mice at P7 (*n* = 3 per group; analyzed as in **a**). The center line represents the median, the box extends from the first to the third quartiles (IQR) and the whiskers extend to 1.5× the IQR. A diamond indicates the mean. Each dot represents an individual biological replicate. Statistical significance was determined using the Wilcoxon rank-sum test. **c**, Genome browser snapshot for *Ccnd1*. **d**, Bar charts showing cell-cycle analysis of each cell type, labeled THY1⁺/KIT⁻, THY1⁺/KIT⁺ uSPG and THY1⁻/KIT⁺ dSPG from null and littermate control (WT or heterozygotes) mice at P7 (*n* = 3, mean ± s.d.; as analyzed in **a**). Statistical significance was determined using the Wilcoxon rank-sum test. **e**, Pictures of WT and *Zbtb16*-null testes at P14. **f**, Box-and-whisker plot showing testes weights of WT and *Zbtb16*-null testes at P14 (*n* = 5 per group). The center line represents the median, the box extends

from first to the third quartiles (IQR) and the whiskers extend to 1.5× the IQR. A diamond indicates the mean. Each dot represents an individual biological replicate. Statistical significance was determined using the Wilcoxon rank-sum test. **g**, Representative IF images showing accumulation of SPG and deficiency of spermatocytes in testis sections from *Zbtb16*-null mice at P14. SYCP3, green; LIN28A, red; pH2AX, magenta; DNA, cyan. White arrowheads indicate SYCP3⁻/LIN28A⁺ uSPG. Yellow arrowheads indicate SYCP3⁻/LIN28A⁺ dSPG. Arrows indicate SYCP3⁺/pH2AX^XY body PSs. Scale bars, 10 μm. The experiment was repeated independently three times with similar results, using three biological replicates. **h**, Violin plots quantifying the number of SPG, PS, total spermatocytes and total cells per seminiferous tubule in WT and *Zbtb16*-null mice at P14 in **g**. A total of 300 circular tubules were counted for each genotype (*n* = 3 biological replicates). The center line represents the median, the black diamond represents the mean and the width reflects the data distribution. Statistical significance was determined using the Wilcoxon rank-sum test. **i**, Fewer tubules with PSs were present in *Zbtb16*-null mice at P14, as analyzed in **g**. Statistical significance was determined using the Wilcoxon rank-sum test.

We first validated prior work[41,42] showing that uSPG express all three TFs (Fig. 4a). We then performed ChIP-seq on SOX3 and SALL4 in whole testis at P7. SOX3 ChIP-seq yielded SOX3 at 19,063 genes and

revealed an impressive overlap with ZBTB16 targets (16,953 genes, 88.9% overlap) (Fig. 4b,c). However, only 1,005 of these genes overlapped with SOX3-occupied genes in neural progenitor cells (NPCs)

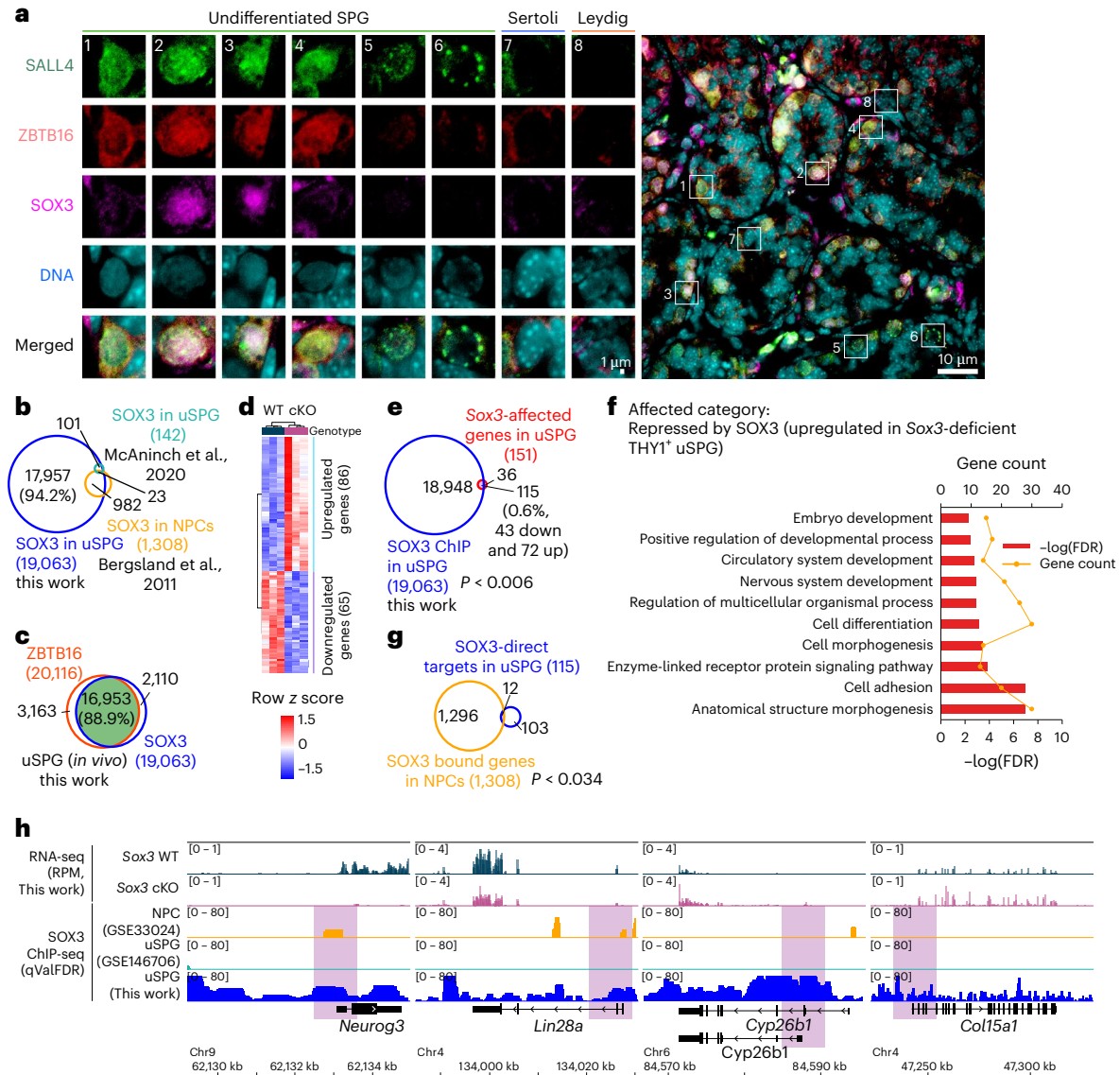

**Fig. 4 | SOX3 promotes SPG differentiation. a**, Representative IF images showing ZBTB16, SALL4 and SOX3 expression patterns in testis sections from WT mice at P7. ZBTB16, green; SALL4, red; SOX3, magenta; DNA, cyan. Scale bars, 10 μm. The experiment was repeated independently three times with similar results, using three biological replicates. **b**, Venn diagram depicting the overlap of SOX3-bound genes in uSPG (this work) with SOX3-bound genes in uSPG[44] and NPCs[71]. **c**, Venn diagram showing the overlap of SOX3-bound genes with ZBTB16. **d**, RNA-seq heat map depicting differential gene expression (fold change cutoff = 1.5, $P < 0.05$, derived from DESeq2 analysis) in *Sox3*-deficient THY1+ uSPG compared to WT. Adjusted $P$ values account for multiple comparisons using the Benjamini–Hochberg method. **e**, Venn diagram showing the overlap of SOX3-bound genes with affected genes by

*Sox3* deficiency (as analyzed in **b**,**d**). Statistical analysis was performed using a hypergeometric test. **f**, GO terms for functional clustering of SOX3-bound genes upregulated in *Sox3*-deficient THY1+ uSPG. **g**, Venn diagram showing the overlap of SOX3-bound genes and *Sox3*-affected genes in uSPG (this work) and SOX3-bound genes in NPCs[71]. Statistical analysis was performed using a hypergeometric test. **h**, Gene targets of SOX3. Browser snapshots displaying genes downregulated (*Neurog3* and *Lin28a*) or upregulated (*Cyp26b1* and *Col15a1*) in *Sox3*-deficient THY1+ uSPG. Tracks include RNA-seq from purified THY1+ uSPG in WT and *Sox3*-cKO mouse testes at P7 and SOX3 ChIP-seq data from uSPG (this work and GSE146706 (ref. 44)) and NPCs (GSE33024 (ref. 71)) (qValFDR).

(Fig. 4b), demonstrating cell-type-specific SOX3 occupancy. To define SOX3 impact, we derived germline-specific *Sox3*-conditional-KO (cKO) mice using *Ddx4*–Cre mice and conducted RNA-seq analysis from *Sox3*-deficient THY1+ uSPG, which identified 151 differentially expressed genes (DEGs): 65 downregulated and 86 upregulated (Fig. 4d). SOX3 occupied 115 of these genes affected by *Sox3*, comprising 43 downregulated and 72 upregulated genes in *Sox3*-deficient THY1+ uSPG (Fig. 4e). These findings suggest a dual role of SOX3 in gene regulation of uSPG, akin to ZBTB16. Analysis of upregulated genes in *Sox3*-null cells enriched GO terms related to cell adhesion and cell differentiation, while there was no significant GO term enriched in downregulated genes (Fig. 4f).

To identify shared functions of SOX3 in uSPG and NPCs, we conducted an intersection analysis of ChIP-seq data in uSPG with data from NPCs, revealing only 12 co-occupied genes (Fig. 4g). Examples include *Cyp26b1*, a retinoic acid (RA)-metabolizing enzyme[45] that is attenuated by SOX3, suggesting a candidate mechanism to enhance RA responsiveness, leading to uSPG differentiation. Additionally, SOX3 directly activates *Lin28a* expression, which promotes uSPG differentiation[46] (Fig. 4g,h). Together, our results raise the possibility that SOX3 promotes cell differentiation through RA signaling and *Lin28a*, while also tempering cell migration, potentially linked to SPG differentiation.

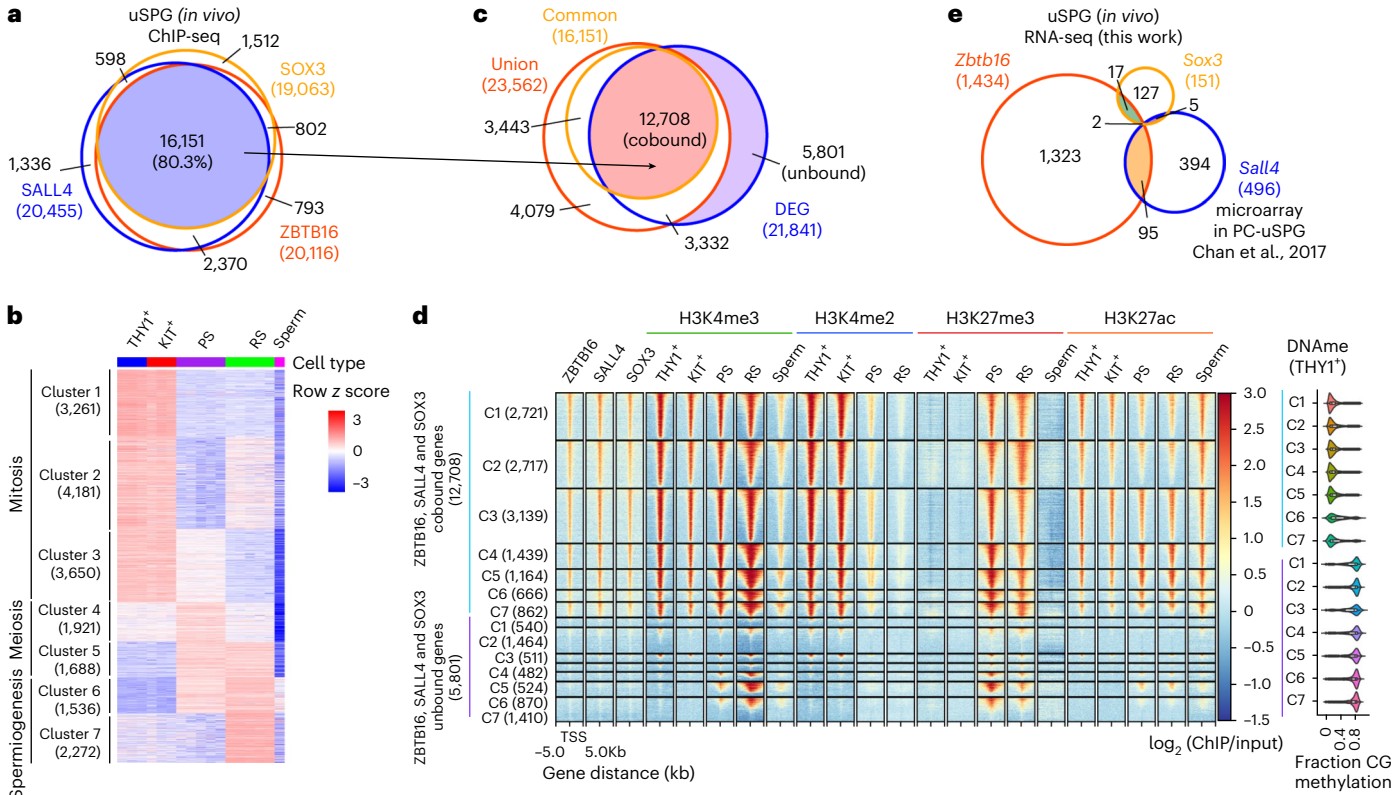

**Fig. 5 | A ZBTB16, SALL4 and SOX3 TF and chromatin network in uSPG.**
**a**, Venn diagram showing the overlap of ZBTB16-bound genes with SALL4 and SOX3. **b**, RNA-seq heat map showing differential gene expression across spermatogenesis[19] ($k = 7$). **c**, Venn diagram showing the overlap of genes cobound by ZBTB16, SALL4 and SOX3 with DEGs during spermatogenesis as analyzed in **a**,**b**. **d**, Left, ChIP-seq heat map showing differential enrichment at promoters of genes cobound by ZBTB16, SALL4 and SOX3 (typical promoter) and unbound genes (atypical promoter), along with histone modifications[19,23,51] across spermatogenesis as analyzed in **c**. Right, violin plot showing fraction of DNAme in THY1[+] uSPG[1]. **e**, Venn diagram showing the overlap of *Zbtb16*-affected genes (this work) with *Sox3*-affected genes (this work) and *Sall4*-affected genes[42].

## SALL4 coactivates a subset of ZBTB16 target genes

We next examined the roles of SALL4 in uSPG. First, our data do not support the prior model that SALL4 sequesters or antagonizes ZBTB16 to prevent ZBTB16 activation of *Kit* (ref. 34). This is because ZBTB16 and SALL4 were coexpressed only in uSPG and early KIT[+] dSPG, while ZBTB16 was absent in late KIT[+] dSPG (Fig. 4a and Extended Data Fig. 8a). SALL4 occupied 20,455 genes but only directly regulated 351 genes, constituting only 1.7% of bound genes (Extended Data Fig. 8b). There was remarkable overlap in the genes bound by ZBTB16 and SALL4, encompassing 18,521 genes (90.5% overlap in uSPG) and 5,510 genes (66.7% in PC-uSPG) (Extended Data Fig. 8c,d). However, SALL4 targets yielded certain unique GO terms, suggesting unique functions for SALL4 in uSPG (Fig. 1d,e and Extended Data Fig. 8e,f). SALL4 occupancy was lost at 55.8% of its in vivo target genes in PC-uSPG despite higher *Sall4* expression in PC-uSPG compared to THY1[+] uSPG (Extended Data Fig. 8b,g,h). However, this difference in *Sall4* expression was not statistically significant, underscoring variability and potential differences between in vivo and in vitro conditions. Notably, common direct target genes of ZBTB16 and SALL4 showed a positive correlation (Extended Data Fig. 8i–l). Thus, SALL4 and ZBTB16 collaboratively regulate certain targets, challenging prior antagonism models[34]. Collectively, loss of ZBTB16, SOX3 or SALL4 impacts a distinct and limited subset of genes, indicating limited independent functions.

## ZBTB16, SALL4 and SOX3 co-occupy mitotic and meiotic genes

We next explored the contributions of ZBTB16, SALL4 and SOX3 in transcriptional regulation of uSPG. IF analyses showed that the nuclear localization of SALL4 and SOX3 in uSPG was not affected by the absence of *Zbtb16* (Extended Data Figs. 6f and 8a). Furthermore, ChIP-seq analysis using testis from WT and *Zbtb16*-null mice at P7 revealed only a slight reduction in SALL4 occupancy, while SOX3 occupancy remained unchanged in the absence of ZBTB16 (Extended Data Fig. 9a). These findings suggest that SALL4 and SOX3 bind their targets independently of ZBTB16.

Remarkably, ZBTB16, SALL4 and SOX3 co-occupied 16,151 gene promoters (Fig. 5a) and their co-occupancy aligned with active chromatin marks, especially H3K4me3 and H3K27ac in uSPG (Fig. 2c and Extended Data Fig. 9b–e), prompting a deeper exploration for a possible collective role for these three factors in chromatin opening or poising at 16,000 genes. We investigated connections between these three TFs and the temporal stages of gene expression of their occupied genes by examining THY1[+] uSPG, KIT[+] dSPG, PSs, RSs and mature sperm. This revealed 21,841 DEGs that were divided into clusters representing the developmental stage of peak expression: mitosis (clusters 1–3), meiosis (clusters 4 and 5) and spermiogenesis (clusters 6 and 7) (Fig. 5b).

Among these DEGs, 12,708 genes overlapped with promoter binding by the three TFs, while 5,801 genes were unbound (Fig. 5c). Notably, in THY1[+] uSPG and KIT[+] dSPG, these factors occupied most genes activated in mitosis and meiosis (clusters 1–5) but only a minority of genes activated in spermiogenesis (clusters 6 and 7) (Fig. 5d). Here, although the fraction of genes bound within these two classes was vastly different, their levels of occupancy at typical promoters across clusters 1–7 were similar (Extended Data Fig. 9f).

Whereas most cobound genes were protein-coding genes, ~50% of unbound genes were noncoding RNAs (Extended Data Fig. 9g). Interestingly, the limited numbers of genes co-occupied by the three TFs in

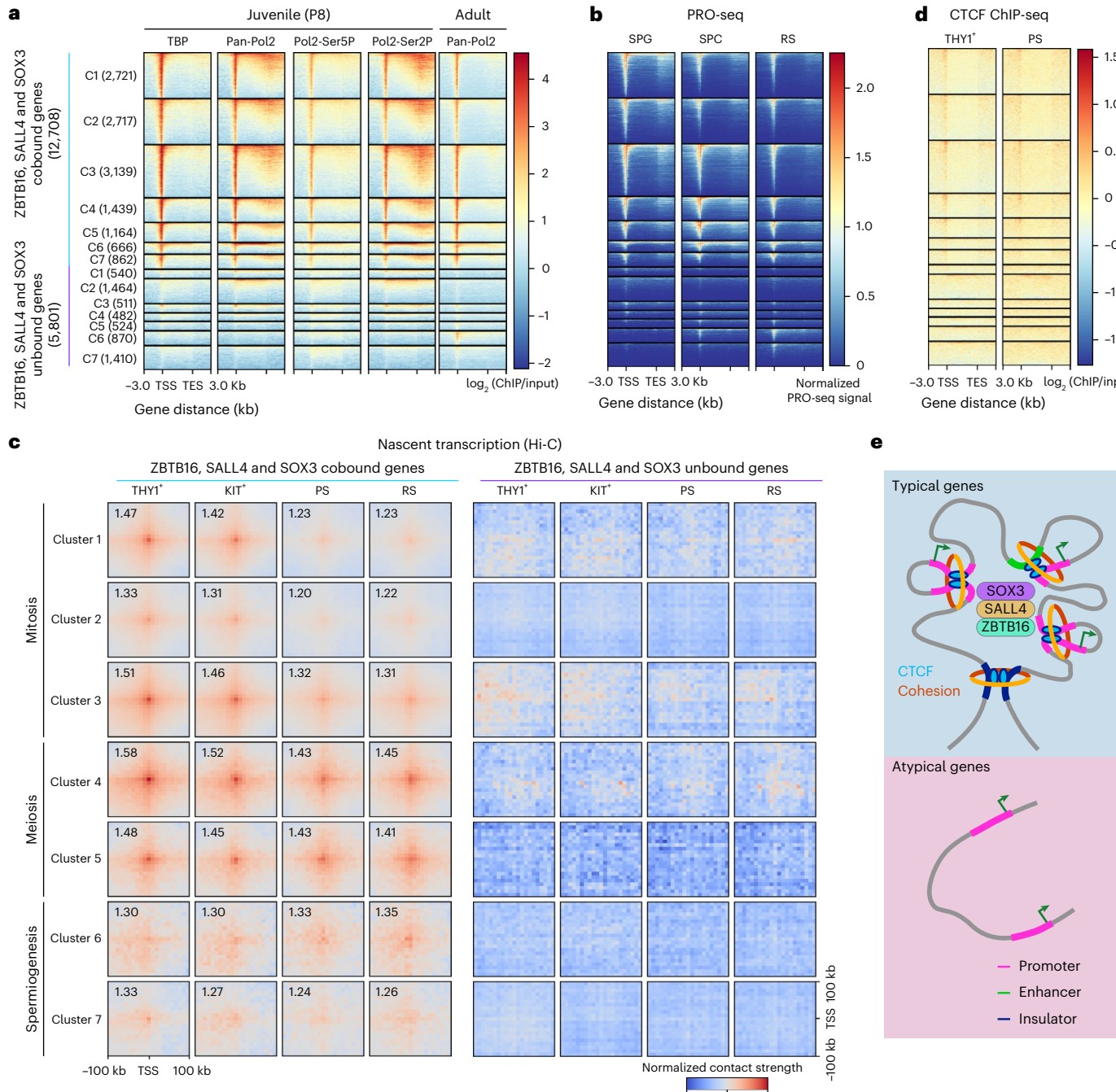

**Fig. 6 | Differential occupancy of RNAPol2 and higher-order chromatin interactions at typical and atypical promoters during spermatogenesis.**
**a**, Metagene heat map showing occupancy of TBP, RNAPol2 and RNAPol2 active states marked by CTD Ser5 phosphorylation (transcription initiation) and Ser2 phosphorylation (transcription elongation) at typical and atypical promoters in testes from juveniles (P8; only SPG) or adults (all germ cell types)[53]. TES, transcription end site. **b**, Metagene heat map illustrating PRO-seq signal data[21], highlighting RNAPol2 pausing in SPG at meiotic genes (as analyzed in Fig. 5d) and the retention of RNAPol2 at silent typical genes in RSs. **c**, Aggregate plots of Hi-C data, depicting the P–P interactions at typical promoters or atypical

promoters across different stages of spermatogenesis (as analyzed in Fig. 5d), revealing the higher-order chromatin organization and interactions solely among typical promoters. **d**, Metagene heat map displaying CTCF enrichment in THY1⁺ uSPG and PSs at promoters of typical and atypical genes (as analyzed in Fig. 5d) showing CTCF binding focused at typical promoters. **e**, Model illustrating that ZBTB16, SALL4 and SOX3 co-occupy typical gene promoters, forming an active 3D conformation enriched with CTCF, which differs from chromatin organization observed at atypical gene promoters during different phases of spermatogenesis.

clusters 6 and 7 (spermiogenesis) were mainly associated with splicing and translation in somatic cells (for example, small nuclear RNAs (16 genes), *Eif4a2* and small nucleolar RNAs (68 genes)) and, thus, were not exclusive to spermiogenesis (Extended Data Fig. 9h). Conversely, many unbound genes are associated with histone-to-protamine exchange

(Extended Data Fig. 9i), which is exclusive to spermiogenesis. Thus, genes co-occupied by ZBTB16, SALL4 and SOX3 were primarily associated with mitotic function in SPG or meiotic processes, whereas a modest number of co-occupied genes activated during spermiogenesis had somatic expression/functions and, therefore, were not germline specific.

## ZBTB16, SALL4 and SOX3 distinguish typical from atypical promoters

Our prior work revealed an unexpected chromatin logic during spermatogenesis, where atypical promoters marked by high DNAme and H3K27me3 represent the majority of active promoters during spermiogenesis[19]. This contrasts with typical promoters, which are associated with active chromatin modifications (H3K4me3 and H3K27ac), lack DNAme and are active primarily during mitotic and meiotic phases[19]. Intriguingly, our analyses here revealed that the target promoters of ZBTB16, SALL4 and SOX3 were strongly associated with typical promoters bearing active chromatin features such as H3K4me2/3, H3K27ac and low DNAme, regardless of the spermatogenesis stage (Fig. 5d). Conversely, genes lacking these three factors exhibited atypical chromatin features, including high DNAme (Fig. 5d). Furthermore, these co-occupied typical promoters maintained consistent enrichment of H3K4me3 and H3K27ac across spermatogenesis. (Fig. 5d and Extended Data Fig. 9j). Chromatin accessibility, measured by assay for transposase-accessible chromatin using sequencing (ATAC-seq)[23,47], was established early in uSPG and remained consistently open in typical genes throughout spermatogenesis. In contrast, atypical genes showed more restricted chromatin openness (only focally open), even during expression in spermiogenesis (Extended Data Fig. 9k).

Interestingly, H3K27me3 was highly enriched at meiotic, spermiogenic and a subset of mitotic typical promoters during expression (Fig. 5d and Extended Data Fig. 9j). Enrichment of H3K27me3 has also been observed at a subset of promoters in ES cells that are undergoing active transcription[48]. These findings suggest that H3K27me3 alone may not be sufficient for strong gene repression, indicating a more complex role in gene regulation. Additionally, atypical genes such as those encoding protamine, gained H3K4me3 in PSs (along with many typical genes) and retained active histone modifications (H3K4me3 and H3K27ac) in mature sperm. In contrast, atypical genes that acquired H3K4me3 and became transcriptionally active in RSs subsequently lost these modifications in mature sperm (Extended Data Fig. 9j).

As ZBTB16 regulates enhancers in hematopoietic progenitors[49] and MS cells[50], we then extended our analyses to known super-enhancers (SEs) in spermatogenesis[51]. Notably, these three TFs cobound the majority of previously identified mitotic (81/107) and meiotic (260/425) SEs, which were characterized by H3K4me2/3 and H3K27ac marks. Conversely, unbound meiotic SEs were either low or lacking H3K4me2/3 and H3K27ac (Extended Data Fig. 9l). Taken together, uSPG appear to establish a large TF-active chromatin network that opens chromatin at typical promoters and the large majority of meiotic SEs to drive the SPG mitotic program, which may poise the meiotic program for subsequent activation.

## Transcription and occupancy interactions between ZBTB16 and reproductive homeobox 10 (RHOX10) in uSPG

To better understand the transcription network regulatory logic, we next sought a TF that might reside and impact only a minor subset of the network targets. We chose the candidate RHOX10, which facilitates the transition from gonocytes to uSPG, in part through indirect activation of *Zbtb16* (refs. 28,52). Strikingly, 99.1% of RHOX10-bound genes in PC-uSPG overlap with ZBTB16-bound genes in uSPG and 82.5% of RHOX10-bound genes overlap with ZBTB16, SALL4 and SOX3. However, only 8.9% of ZBTB16-occupied loci are co-occupied by RHOX10 (Extended Data Fig. 10a). As expected, RHOX10-occupied gene promoters align with the active chromatin marks H3K4me3 and H3K27ac and exclude H3K27me3, implying a role at active genes (Extended Data Fig. 10b,c). Functional analysis (RNA-seq) of *Zbtb16* and *Rhox10* mutant uSPG revealed only 134 commonly regulated genes (Extended Data Fig. 10d). Among these, only three genes emerged as direct cotargets (Extended Data Fig. 10e). Interestingly, genes indirectly affected by RHOX10 displayed anticorrelation with ZBTB16's direct target genes, particularly for cell migration genes such as *Zeb2* (Extended

Data Fig. 10f). Collectively, these results highlight an example of a TF, RHOX10, that binds and activates only a minor subset of the targets within the ZBTB16–SALL4–SOX3 network and specifically impacts cell migration targets in uSPG.

## RNAPol2 and TATA-binding protein (TBP) at typical promoters persist in spermatogenesis

Recent interesting work reported the presence of RNAPol2 at meiotic genes in uSPG, even before their peak expression during meiosis[21], suggesting that paused RNAPol2 primes meiotic genes for rapid activation. To investigate whether this pausing of RNAPol2 is specific to typical promoters or also occurs at atypical promoters, we conducted a series of ChIP-seq experiments focusing on TBP, RNAPol2 and its active forms, marked by C-terminal domain (CTD) Ser5 phosphorylation (transcription initiation) and Ser2 phosphorylation (transcription elongation). These experiments were dovetailed with recent precision run-on sequencing (PRO-seq) data from uSPG, spermatocytes and RSs[21].

Our ChIP-seq analysis using juvenile testis revealed that both TBP and RNAPol2 consistently occupied typical gene promoters across all stages of spermatogenesis but were notably absent from atypical gene promoters (Fig. 6a). This finding indicates that the transcriptional machinery for meiotic gene expression is poised during the SPG stage and resides within an open and active or poised chromatin landscape. Interestingly, the small number of typical promoters (consisting of genes that are not germline specific) that are active during spermiogenesis are also poised in uSPG.

Additionally, we observed that RNAPol2, marked by Ser5 or Ser2 phosphorylation, occupied the promoters of meiotic genes in SPG. The presence of both phosphorylated forms indicates that RNAPol2 was actively engaged in both the initiation and the elongation phases of transcription at these promoters (Fig. 6a). We note that meiotic genes were expressed at low levels in SPG and then highly expressed during meiosis, indicating that RNAPol2 was prerecruited and primed for elevated expression levels specifically during the meiotic phase. This strategic readiness ensures transcriptional machinery preassembly, allowing for rapid and robust activation of critical meiotic genes when the appropriate developmental signals are received.

Strikingly, our analysis of recent PRO-seq data[21] revealed a nearly complete overlap (86.9% overlap) with typical promoters and no notable overlap (1.1%) with atypical promoters (Fig. 6b). This overlap suggests a specific regulatory strategy where RNAPol2 pausing is linked to the chromatin categorization of promoters. Intriguingly, RNAPol2 remained associated with mitotic and meiotic genes even after these genes were silenced during spermiogenesis, strongly suggesting a mechanism for reimposing pausing at these promoters. In contrast, RNAPol2 ChIP-seq analysis using adult testis[53], combined with the PRO-seq data[21], revealed that atypical genes recruit RNAPol2 in tandem with active transcription during meiosis and especially spermiogenesis, most notably within clusters 6 and 7 (Fig. 6a,b).

## Extensive chromatin interactions among typical promoters

To investigate whether the shared TF-active chromatin network revealed above forms a physical higher-order chromatin network in uSPG, we used Hi-C to examine promoter–promoter (P–P) interactions across spermatogenesis. Remarkably, in uSPG, typical promoters, including those with peak expression in meiosis, displayed robust physical interaction with other nearby typical promoters (within ~20 kb), whereas atypical promoters lacked such P–P interactions. Interestingly, in meiotic cells, the interactions involving mitotic gene typical promoters were reduced while the interactions involving typical meiotic and (rarer) spermiogenic promoters remained robust (Fig. 6c).

Furthermore, in uSPG, we found that 56% of CCCTC-binding factor (CTCF)-bound promoters were typical promoters, whereas only 2.9% of CTCF-bound promoters were atypical, mirroring the pattern observed in PSs (Fig. 6d). Thus, these Hi-C results reveal the

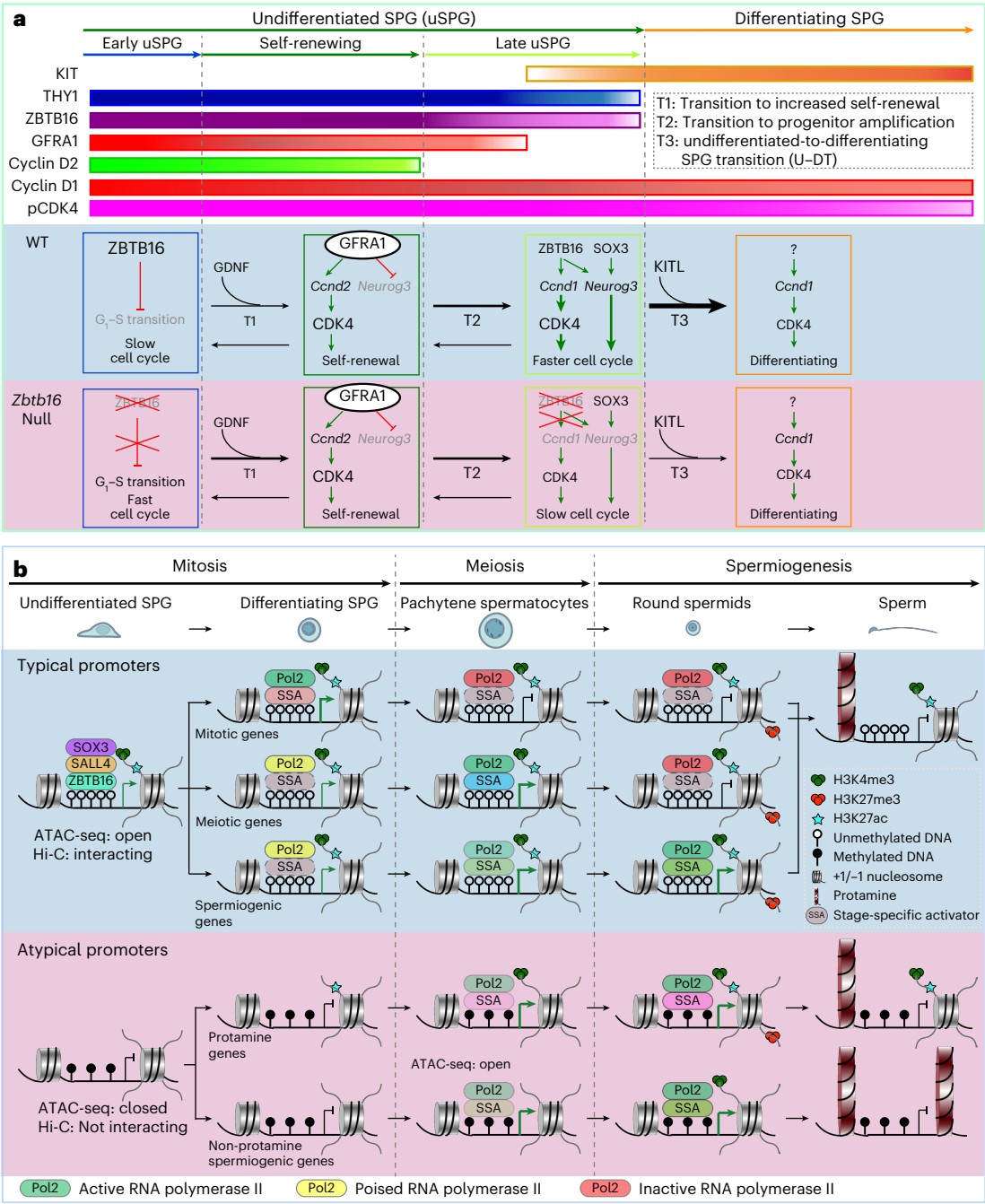

**Fig. 7 | Transcriptional and chromatin regulation during spermatogenesis.**
**a**, Schematic summary of the logic used by key factors during phases of
spermatogenesis. First, differential expression and regulation of CCND1 and
CCND2 and cognate partner CDK4 accompany early uSPG versus late uSPG. A
lack of *Zbtb16* promotes (thicker transition 1 arrow) the transition of quiescent
uSPG to more active early uSPG, which are CCND2 positive. However, as ZBTB16
normally also helps activate *Ccnd1* and *Neurog3* to promote developmental
progression, uSPG lacking *Zbtb16* are less able to progress (thinner transition 3
arrow) through U–DT. **b**, Schematic summarizing the distinctive TF and histone
dynamics at typical and atypical gene promoters during spermatogenesis
phases. In uSPG, typical promoters are actively bound by the three TFs
(ZBTB16, SALL4 and SOX3), along with poised and paused RNAPol2 (ref. 21),

and are accompanied by H3K4me3, H3K27ac, accessible chromatin (assessed
by ATAC-seq), higher-order P–P interactions (examined by Hi-C) and DNA
hypomethylation. Conversely, atypical promoters lack these factors and features
in uSPG and instead bear DNAme and recruit only RNAPol2 when activated in
spermiogenesis. Notably, despite the absence of all three TFs in spermatocytes,
the chromatin features of typical promoters persist throughout meiosis and
spermiogenesis, albeit with the addition of H3K27me3 during these phases.
Here, we speculate that SSAs activate subsets of the typical network in SPG (for
example, RHOX10, SOHLH1/2, NEUROG3 and DMRT1) and during meiotic phases
(for example, STRA8, cMyb and TCFL5), whereas a largely separate set of SSAs
(for example, RFX2, CREM and ACT) activates atypical promoters to execute
stages of spermiogenesis.

establishment of a higher-order physically interacting network involv-
ing typical promoters in uSPG, which persists at meiotic genes dur-
ing meiosis but diminishes at mitotic genes as they are silenced and
acquire H3K27me3.

Taken together, in uSPG, ZBTB16, SOX3 and SALL4 (and often
CTCF) co-occupy >15,000 promoter and enhancer locations that lack
DNAme, bear active histone modifications, exhibit open chromatin
and (for meiotic genes) contain paused RNAPol2. These factors interact

in a higher-order three-dimensional (3D) network that is largely maintained through spermatogenesis. This contrasts with the majority of genes activated in spermiogenesis, which lack these TFs, are marked by DNAme, lack 3D interactions, and recruit RNAPol2 only at the time of expression (Fig. 6e).

## Discussion

Our work started with an in-depth examination in the testis of the uSPG-specific TF ZBTB16 and its relationships to other uSPG-specific TFs and chromatin. Our findings indicate that ZBTB16 exerts multifaced roles in different uSPG processes (Fig. 7a). We then broadened the work, which led to the unexpected finding that a set of TFs participates in a higher-order 3D chromatin network in uSPG to presumably poise the subsequent meiotic program, which is separate from the mechanisms and factors that appear to promote the majority of the spermiogenesis program (Fig. 7b).

First, we addressed roles for ZBTB16 in uSPG development, showing that early uSPG maintain a slow cell cycle through high ZBTB16, impeding $G_1$–S progression (Fig. 7a). This aligns with the accumulation of S-phase eomesodermin-positive early uSPG in *Zbtb16*-null mice[11] and conceptually resembles myeloid cells, where *Zbtb16* overexpression decelerates $G_1$–S progression by suppressing *Ccna2* (refs. 54,55). Notably, the number CCND1⁺ uSPG and dSPG significantly decreased after P14 in *Zbtb16*-null mice[10], which corresponds with the reduction in testis size observed from P14 onward (Fig. 3d,e and Extended Data Fig. 8d,e). This may partly underlie ZBTB16's putative role in supporting long-term uSPG maintenance[7,10,11]. Although the exact mechanism remains unknown, ZBTB16's regulation of the cell cycle does not seem to involve S-phase cyclins (Extended Data Fig. 8b). Instead, GDNF drives cyclic expression of CCND2 to trigger early uSPG self-renewal[36,37,40,56] (Fig. 7a). However, for late uSPG, we established that ZBTB16 activates *Ccnd1*, which leads to the U–DT (Fig. 7a).

*Cdk4* promotes early uSPG self-renewal[39,57] and functions in late uSPG, dSPG and spermatocytes. uSPG self-renewal involves cell-cycle progression driven by GDNF and CCND2 (refs. 36,37,40,56). However, late uSPG use CCND1, driven in part by ZBTB16, to promote cell-cycle progression. Thus, both early and late uSPG deploy pCDK4 but use distinct CCND–CDK4 complexes to guide uSPG progression and both rely on ZBTB16 for effective regulation of CCND1 (Fig. 7a). Overall, murine ZBTB16 appears to finetune uSPG long-term maintenance by balancing self-renewal and progenitor progression. In keeping with reduced testicular size in *Zbtb16*-null mice, a prepubertal case with a *ZBTB16* mutation likewise exhibited reduced testicular size[58], hinting toward conservation of roles in humans. Furthermore, our work comparing uSPG to PC-uSPG indicates that ZBTB16 exerts roles in supporting MS cells, reinforcing uSPG identity and preventing conversion to an ES cell-like pluripotent state in vitro.

Next, our work provided several lines of evidence supporting an extensive higher-order and interacting network involving TFs and active chromatin in uSPG that encompasses the majority of the uSPG transcriptional program and poises the SPG differentiation and meiosis programs (Fig. 7b). This network involves over 12,000 promoters expressed in germ cells during spermatogenesis and a plethora of meiotic SEs (Extended Data Fig. 9k), which are co-occupied by ZBTB16, SALL4 and SOX3 and bear high and broad H3K4me3 and DNA hypomethylation in uSPG (Fig. 7b). These results may also provide a mechanistic underpinning for the recent observation of paused RNAPol2 at meiotic gene promoters in uSPG[21]. Notably, this co-occupancy and poised chromatin status distinguishes typical (co-occupied) from atypical promoters (unoccupied and DNA methylated), which instead drive most genes during spermiogenesis and recruit RNAPol2 commensurate with transcription. Our Hi-C data reinforce this notion, underscoring a higher-order physical principle aligned with our promoter classifications; typical promoters display robust physical interactions within uSPG, whereas atypical promoters lack such interactions (Fig. 7b). This

finding is consistent with the observation that CTCF binding prefers regions with low DNAme[59–64], which may contribute to the establishment of open chromatin and facilitate promoter interactions. Regarding dynamics, meiotic promoter interactions were maintained during meiotic activation, whereas mitotic promoters were reduced, correlated with their silencing during meiosis. Thus, these extensive higher-order interactions appear to be established in uSPG and to be notably decommissioned as transcription is silenced (clusters 1 and 2; Fig. 6d).

A major question is why spermatogenesis might use such an extensive TF–chromatin physical poising network. Here, we note that cytoplasmic bridges connect male germ cells during spermatogenesis, which enables the sharing of contents, including TFs[65]. This may enable synchronization of the expression of new TFs (stage-specific activators, SSAs; Fig. 7b), which then conduct synchronous activation of the promoters that contain their binding sites, which already reside in poised or open chromatin across the interconnected cells, promoting orderly entry into and progression through the stages of meiosis. Here, we speculate that particular SSAs activate particular subsets of the typical network in SPG differentiation (for example, RHOX10, spermatogenesis and oogenesis specific basic helix–loop–helix 1 and 2 (SOHLH1/2), NEUROG3 and doublesex and mab-3 related TF 1 (DMRT1)) and during meiotic phases (for example, stimulated by retinoic acid 8 (STRA8), cMyb and TF-like 5 (TCFL5)), whereas a largely separate set of SSAs (for example, regulatory factor X2 (RFX2), cyclic adenosine-monophosphate-responsive element modulator (CREM) and activator of CREM (ACT)) activates atypical promoters to execute stages of spermiogenesis (Fig. 7b).

Additional future work would likely focus on determining whether these extensive interactions between typical promoters represent chromatin loops that reside in condensates or hubs that coalesce locally-interacting typical promoters. Notably, this chromatin poising relationship bears some resemblance to the poising logic observed in the pluripotency network of ES cells[66–70], which involves hundreds of developmental genes, although the uSPG network revealed here is much larger. In counter distinction, atypical promoters are not occupied by this network and involve genes specific to male germline and spermatogenesis, including the global histone–protamine exchange (for example, *Tnp1* and *Prm1*) during spermiogenesis (third phase in Fig. 7b).

Regarding transcriptional specificity, we found that loss of one factor (*Zbtb16*, *Sall4* or *Sox3*) impacts only a small subset of the ~16,000 genes occupied by the network, consistent with partial redundancy among these factors, possibly buffered by the higher-order chromatin interaction network and the likely involvement of additional TFs contributing to the network, which were not highlighted in this study. This redundancy for poising is supported by our observation that omission of ZBTB16 only minimally impinges on the binding of SALL4 or SOX3 at network targets in uSPG. Thus, an important issue for future work is to determine whether additional TFs localize to and help define the chromatin state of the entire network versus those that specifically occupy and regulate a smaller subset of the network (for example, RHOX10). Such endeavors would inform both redundancy and specificity within the network. Here, we hypothesize that SSAs stimulate transcription on an already poised or active promoter chromatin landscape at typical promoters during meiosis, while a separate set of SSAs (which can bind methylated DNA) then activate DNA-methylated atypical promoters during spermiogenesis (Fig. 7b). Further studies would need to be directed at understanding the establishment of this TF and higher-order chromatin network in uSPG and its dynamic changes during spermatogenesis.

## Online content

Any methods, additional references, Nature Portfolio reporting summaries, source data, extended data, supplementary information, acknowledgements, peer review information; details of author contributions

and competing interests; and statements of data and code availability are available at https://doi.org/10.1038/s41594-025-01509-5.

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

## Methods

### Mouse husbandry and genotyping

The mice used in this study were derived from the C57BL/6J (B6) background obtained from Jackson Laboratory (RRID:IMSR_JAX:000664). *Zbtb16* (luxoid) mice, described previously[7], were also procured from Jackson Laboratory (RRID:IMSR_JAX:000100). Genotyping of *Zbtb16*-null mice was performed using the TaqMan single-nucleotide polymorphism genotyping assay (Thermo Fisher Scientific, 4332075, clone ID: AH7042A) after assessment of the mice for the presence of hind limb defects. Homozygous *Zbtb16*-deficient (*Zbtb16*lu/lu) mice were compared to WT (*Zbtb16*+/+) and heterozygous (*Zbtb16*+/lu) littermates in all experiments.

Sox3-floxed mice (B6) were previously generated[72]. *Ddx4*/*Vasa*–Cre[73] (B6, RRID:IMSR_JAX:018980) was purchased from Jackson Laboratory. *Sox3*-cKO mice (B6), specifically targeting male germ cells, were generated using *Ddx4*–Cre mice and the cKO mice were genotyped as described previously[72]. All mice were housed in a pathogen-free animal facility and provided with a standard rodent chow diet. The housing facility maintained a controlled temperature (20–25 °C), a 12-h light–dark cycle and a relative humidity of 30–70% to support the mice's circadian rhythm. The study was conducted in accordance with approved animal use protocols (no. 18-03004 and 00001726) by the Institutional Animal Care and Use Committee at the University of Utah and the National Institutes of Health Guide for the Care and Use of Laboratory Animals.

### Protein extracts and immunoblot

Mouse testes were homogenized in lysis buffer containing 20 mM Tris-HCl pH 8.0, 150 mM NaCl, 2 mM EDTA pH 8.0 and 1% Nonidet-P40 (NP-40) with proteinase inhibitors. After centrifugation, the supernatants were diluted with sample buffer and boiled. Then, 100 µg of total proteins per lane were separated on a 10% SDS–PAGE and subsequently transferred to PVDF membranes (GE Healthcare, RPN303F). For immunoblotting, primary antibodies were diluted in 5% nonfat dried milk in 0.1% Tween-20 TBS and incubated overnight at 4 °C. The primary antibodies used for immunoblotting were anti-rabbit ZBTB16 (Santa Cruz, sc-11146; 1:400) and anti-goat ZBTB16 (Santa Cruz, sc-22839; 1:100). After washing, membranes were probed with horseradish peroxidase-conjugated goat anti-rabbit IgG (Bio-Rad, 1706515; 1:10,000) and bovine anti-goat IgG (Santa Cruz, sc-2350; 1:5,000) secondary antibodies.

### RNA-seq sample preparation and analysis

Testes from three mice in *Zbtb16*-null and *Sox3*-cKO conditions, along with their respective WT littermates, were pooled for each cell isolation experiment. Single cells were isolated from testes at P7 using 0.25% trypsin digestion and resuspended in MACS separation buffer (Miltenyi Biotec, 130-091-221). The harvested cells were used to purify uSPG and dSPG through IMCS. This was achieved using anti-THY1/CD90.2 (Miltenyi Biotec, 130-049-101; 1:10) and anti-KIT/CD117 (Miltenyi Biotec, 130-091-224; 1:5) antibodies conjugated with magnetic microbeads, following previously described protocols[1,22]. The purified cells were lysed in TRIzol and RNA was subsequently isolated and purified using a Direct-zol RNA mini-prep kit (Zymo Research, R2061), which included DNase I treatment to remove genomic DNA.

To assess the RNA integrity, high-sensitivity R6K Screen Tapes were used with a 2200 TapeStation instrument (Agilent Technologies, G2991AA) and the RNA integrity number equivalent values were confirmed to be at least 8.1. RNA libraries were prepared using the Illumina TruSeq stranded RNA kit with Ribo-Zero gold for the mouse, which facilitated the removal of ribosomal RNA (New England Biolabs). The library sequencing was performed on an Illumina 2000/2500 platform and each library sequenced up to 74 million fragments using HiSeq 50-cycle single-read sequencing version 4.

The RNA-seq reads from each library were processed and aligned to the Ensembl annotation NCBI38/mm10 using Novoalign (version

4.04.01, http://novocraft.com). Adaptor sequences were trimmed using the following parameters: -o SAM -r all 50 -a AGATCGGAAGAG-CACACGTCTGAACTCCAGTCA. Processed reads were used to generate FPKM (fragments per kilobase of transcript per million mapped reads) values. The analysis of differential gene expression was performed using the default parameters with the addition of -x 50000000 using USeq8.9.6 packages that included DESeq2 (version 1.42.1)[74,75]. Significantly affected genes were defined as those with an adjusted *P* value ≤ 0.05 and a 1.5-fold absolute change. Adjusted *P* values were calculated using the Benjamini–Hochberg method to account for multiple comparisons. The volcano plot was generated with these thresholds to highlight significant upregulated and downregulated genes. Published RNA-seq datasets were retrieved from the National Center for Biotechnology Information Gene Expression Omnibus (GEO) and Sequence Read Archive (SRA) using GNU Wget and SRA Toolkit (version 2.10.8) with the fasterq-dump utility for sequence extraction. For consistency and comparability, all datasets were aligned to the mm10 mouse genome using uniform alignment criteria. The following datasets were used: GSE78127 for PC-uSPG, MASCs, ES cells and HSCs; GSE49624 for PSs; SRA097278 for E13.5 male PGC.

### ChIP-seq

ChIP-seq experiments were performed following a previously established protocol[19] with some modifications. For TF ChIP-seq, seminiferous tubules were isolated from three mice at P7–P8, after removing the tunica albuginea. For histone modification ChIP-seq, THY1+ uSPG were purified from a total of seventy mice, following the same procedure as described in the RNA-seq sample preparation section. The isolated tissues or THY1+ uSPG were crosslinked using 1% formaldehyde–PBS for 10 min at room temperature (RT) and then quenched with glycine (final concentration of 0.125 M) for 5 min. The crosslinked tissues and cells were washed twice with PBS and then frozen using liquid nitrogen. Subsequently, they were homogenized with 15 strokes using a KIMBLE Dounce tissue grinder (Millipore Sigma, D8938) with 10 ml of SDS buffer (100 mM NaCl, 50 mM Tris-Cl pH 8.0, 5 mM EDTA, 10% SDS and protease inhibitor cocktail tablet (Millipore Sigma, 4693159001)). The homogenized samples were transferred to 15-ml tubes and lysed at RT for 10 min. The lysates were then centrifuged at 2,400$g$ at 4 °C for 5 min to collect the nuclei, which were subsequently resuspended in IP buffer (100 mM NaCl, 66.6 mM Tris-Cl pH 8.0, 5 mM EDTA, 6.6% SDS, 1.3% Triton X-100 and protease inhibitor cocktail tablet). The nuclear extracts were sheared using a Branson sonifier, with 0.9 s on and 0.1 s off for seven cycles, resulting in an average fragment size of 300–700 bp. The sheared chromatin was then centrifuged at 18,000$g$ at 4 °C for 10 min and the supernatant was subjected to immunoclearing by incubating with 20 µl of Dynabeads Protein A/G (Thermo Fisher Scientific, 10002D and 10004D) at 4 °C for 30 min. After immunoclearance, 2 mg of the sheared chromatin was incubated with the specific antibodies, including rabbit anti-ZBTB16 (Santa Cruz, sc-22839, 5 µg (25 µl)), goat anti-ZBTB16 (Santa Cruz, sc-11146, 5 µg (25 µl)), rabbit anti-SALL4 (Abcam, ab29112, 5 µg), mouse anti-SOX3 (Santa Cruz, sc-101155, 2.5 µg (25 µl)), mouse anti-TBP (EMD Millipore, MAB3658, 1 µl), pan-RNAPol2 (Active motif, 39097, 0.2 µg (1 µl)), RNAPol2 CTD Ser5 (Active Motif, 39233, 5 µg (5 µl)), RNAPol2 CTD Ser2 (Active motif, 61083, 5 µg (5 µl)), H3K4me3 (Active Motif, 39159, 3 µl), H3K9me3 (Active Motif, 39161, 5 µl) and H3K27me3 (EMD Millipore, 07-449, 5 µl), at 4 °C overnight. To precipitate antibody-bound protein–DNA complexes, 50 µl of Dynabeads protein A/G were used and the incubation lasted for 4 h at 4 °C. The beads were then washed as follows: two times with mixed micelle buffer (150 mM NaCl, 20 mM Tris-Cl pH 8.0, 5 mM EDTA, 34% sucrose (w/v), 0.5% Triton X-100 and 0.2% SDS), two times with buffer 500 (500 mM NaCl, 50 mM HEPES pH 7.5, 1 mM EDTA, 0.1% deoxycholic acid and 1% Triton X-100), two times with LiCl detergent solution (250 mM LiCl, 10 mM Tris-Cl pH 8.0, 1 mM EDTA, 0.5% deoxycholic acid and 0.5% NP-40) and one time with Tris–EDTA buffer (10 mM Tris-Cl

pH 7.4 and 1 mM EDTA). The washed beads and input were resuspended in 130 μl of ChIP elution buffer (1% SDS, 0.1 M NaHCO$_3$) and subjected to overnight incubation at 65 °C with 900 rpm using a thermomixer to reverse-crosslink, including RNase and proteinase digestions. The DNA was purified using the MinElute PCR purification kit (Qiagen, 28006).

To identify TF-binding motifs, ChIP-nexus was conducted, following established procedures[32]. Briefly, the ChIP-exo treatment process, involving lambda exonuclease, was incorporated into the standard ChIP-seq protocol.

For ChIP-seq library preparation, we used the NEBNext ChIP-Seq library prep master mix set (New England Biolabs, E6240L) and the NEBNext Ultra II DNA library prep kit (E7634L). Library sequencing was performed on an Illumina 2000/2500 platform, with up to 33 million fragments for ZBTB16, SALL4, SOX3, H3K4me3, H3K9me3 and H3K27me3 using HiSeq 50-cycle single-read sequencing version 4. Additionally, TBP, RNAPol2, RNAPol2 CTD Ser5 and RNAPol2 CTD Ser2 libraries were sequenced was performed on a NovaSeq X platform, using 150-bp paired-read sequencing.

### ChIP-seq peak calling, replicate handing and peak annotation

For ChIP-seq data analysis, the reads were aligned to the mouse reference genome (mm10) using Novoalign (version 4.04.01, https://novocraft.com) with the following parameters: -o SAM -r random -H -a AGATCGGAAGAGCACACGTCTGAACTCCAGTCA, which includes adaptor sequence removal. To ensure fair comparison across all datasets, PCR duplicates and all unmapped reads were removed using Picard MarkDuplicates (version 2.7.1; https://broadinstitute.github.io/picard/) and SAMtools[76].

Peak calling was performed using USeq8.9.6 packages and MACS (2.1.1) with default parameters[74,77]. Peaks with a −10log$_{10}$($q$ value) adjusted by false discovery rate (qValFDR) greater than 30 ($q < 0.001$) for USeq and 20 ($q < 0.01$) for MACS were considered for downstream analyses. Replicate handing was modified on the basis of a previous approach[78]. Briefly, only peaks from the merged files of two or three replicates overlapped with at least 50% of peaks from the union of biological replicates. This was achieved using BEDTools intersect with parameters -u -f 0.5. Moreover, genomic regions blacklisted in mice were removed from the peaks using BEDTools intersect with the parameter -v (refs. 79,80).

To annotate the peaks, ChIPseeker (version 1.38.0)[81] was used with the RefSeq gene list for genome version mm10. For ZBTB16, SALL4, SOX3, TBP and RNAPol2, promoters were defined as ±2 kb from the transcription start site (TSS); for histone modification, promoters were defined as ±1 kb from the TSS. Additionally, annotated genes that were excluded from the merged replicates but overlapped with at least two biological replicates were included and bidirectional promoter genes were manually added using BEDTools. The raw ChIP-seq data and relative negative controls from previous publications were obtained and reprocessed using the same criteria. The data sources were as follows: GSE33024, GSE49624, GSE50807, GSE55060, GSE57186, GSE73390, GSE78129, GSE79230, GSE89502, GSE124190, GSE125168, GSE130652, GSE146706 and GSE165372.

### Functional classification and pathway analyses

To discern functional insights from the data, we used PANTHER (version 17.0) for GO and pathway analysis, using a statistical overrepresentation test with the default settings[82]. A corrected $P$-value threshold of <0.05 was applied to establish statistically significant correlations.

### piRNAs, CpG islands, DNAme, ATAC-seq and retrotransposons

Mouse piRNA coordinates[83] were converted from mm9 to mm10 using the LiftOver tool available on the UCSC genome browser (http://genome.ucsc.edu/cgi-bin/hgLiftOver). The CpG island database was sourced from the UCSC Genome Browser[84]. DNAme data for THY1$^+$ uSPG, KIT$^+$ dSPG, PSs, RSs and mature sperm were reprocessed

for genome version mm10, extracted from GSE49624 (ref. 19) and GSE62355 (ref. 1) and processed following a previously described procedure[1]. Similarly, ATAC-seq data were reprocessed for genome version mm10, obtained from GSE79230 and GSE102954, using established protocols[23]. For the analysis of retrotransposons, mm10 rmsk data were retrieved from the USCS Table Browser (https://genome.ucsc.edu/cgi-bin/hgTables).

### Heat map visualizations, graphs and plots

Heat maps and box-and-whisker plots for RNA-seq data were generated using R (version 4.3.2) packages, including ggplot2 (version 3.5.1) and pheatmap (version 1.0.12). Genome browser visualization was prepared using normalized ChIP signals, which were first adjusted to their corresponding inputs using USeq after peak calling[74]. Genome browser snapshots were captured using the Integrative Genomics Viewer browser (version 2.16.2)[85]. For further ChIP-seq data analysis, the deepTools suite was used. The bamCompare module was used to normalize ChIP signals against their input. Heat maps and clustering were generated using the computeMatrix and plotHeatmap modules, using a reference point and scale regions. Spearman correlation between ChIP-seq samples was computed using the plotCorrelation module[86]. For the area-proportional Venn diagram, the BxToolBox (http://apps.bioinforx.com/bxaf7c/app/venn/app_overlap.php) was used, with statistical significances assessed using a hypergeometric test (http://nemates.org/MA/progs/overlap_stats.html).

### Statistics and reproducibility

Statistical analyses were conducted to ensure data robustness and reproducibility. For box-and-whisker plots and line graphs, statistical significance was determined using a two-sided Wilcoxon rank-sum test. Exact $P$ values are reported in the figure legends and $n$ values (sample sizes) are explicitly defined for each dataset. Data distribution was assessed using the Shapiro–Wilk normality test.

All data are presented as either the mean ± s.d. or median with interquartile range (IQR), as specified in the legends. Box plots illustrate the minima, maxima, center (median), bounds (first and third quartiles) and whiskers (1.5× the IQR). Individual data points are overlaid for visualization of data distribution and clarity.

Experiments were independently repeated and biological replicates are explicitly stated in the figure legends. Representative images and data reflect consistent outcomes across experiments.

### DNA-binding motif analysis

Analysis of DNA-binding motifs was conducted using the GEM (version 2.6) and MEME suite 5.1.1 programs[30,31]. For GEM, both ChIP and input SAM files were used to analyze the DNA-binding motifs, applying the default parameters --f SAM --k_min 6 --k_max 13. A fragment size of 100 bp was selected from the summits within all peaks, top 500 peaks and gene promoter regions derived from genes either downregulated or upregulated in *Zbtb16*-null cells, as identified using MACS2. For further motif analysis, all known motifs were investigated through MEME-ChIP using the HOCOMOCO Mouse (version 11 FULL) motif database. The regions encompassing binding motifs were scanned using Find Individual Motif Occurrences.

### Flow cytometry analysis

To perform flow cytometry analysis, single cells were isolated from testes at P7 and resuspended in MACS separation buffer (Miltenyi Biotec, 130-091-221), following the procedure outlined for RNA-seq sample preparation. The collected cells were subsequently stained with anti-THY1/CD90.2–PE–Cy7 (Thermo Fisher Scientific, 25-0902-81; 0.06 μg per test) and anti-KIT/CD117–PE (Thermo Fisher Scientific, 12-1171-81; 0.125 μg per test) antibodies, according to the manufacturer's instructions, for a duration of 15 min at 4 °C. Following this, the cells were fixed in a solution of 4% formaldehyde–PBS for 20 min

on ice. After appropriate washes, the cells were subjected to staining with DAPI solution (0.1% Triton X-100 and 10 µg ml⁻¹ of DAPI in PBS) and incubated overnight at 4 °C. Subsequently, the samples were analyzed using fluorescence-activated cell sorting (FACS) with a FACS Canto Scan (Becton Dickinson) using FACSDiva (version 8.01, BD BioSciences). The acquired data were analyzed for cell population and cell cycle using the FlowJo software (version 9.9, BD BioSciences).

## IF and whole-mount staining

For IF analysis, testes from both WT and *Zbtb16*-null mice were weighed and then fixed using 4% formaldehyde, incubated overnight at 4 °C. Following fixation, tissues were processed for paraffin embedding using standard procedures. The paraffinized testes were sectioned into 5-µm slices, followed by deparaffinization using CitriSolv (1601, Decon Labs). Subsequently, rehydration was performed through a graded ethanol series (2 × 100%, 2 × 95%, 1 × 80%, 1 × 70%) for 2 min each, followed by rinsing in 1× PBS.

To enable antigen retrieval, sections were subjected to boiling in 10 mM sodium citrate (pH 6.0) or Tris–EDTA buffer (pH 9.0) for 20 min and then allowed to cool for 20 min. Afterward, the sections were blocked for 1 h at RT using a blocking buffer containing 3% BSA and 5% normal donkey serum in PBS.

Primary antibodies specific to target proteins were diluted in the blocking buffer and added to the sections for overnight incubation at 4 °C. The following primary antibodies were used: rabbit anti-ADAMTS5 (Thermo Fisher Scientific, PA5-27165; 1:100 (10 µg ml⁻¹)), mouse anti-CCND1 (Santa Cruz Biotechnology, sc-8396; 1:50 (4 µg ml⁻¹)), rat anti-CCND2 (Santa Cruz Biotechnology, sc-452; 1:50 (4 µg ml⁻¹)), rabbit anti-pCDK4 (Thr172) (Thermo Fisher Scientific, 702556; 1:100 (5.0 µg ml⁻¹)), rabbit anti-pCDK6 (Thermo Fisher Scientific, PA537517; 1:100 (10.0 µg ml⁻¹)), rabbit anti-pH2AX (Cell Signaling, 9718; 1:400), goat anti-LIN28A (R&D systems, AF3757; 1:400), rabbit anti-SALL4 (Abcam, ab29112; 1:1,000 (1 µg ml⁻¹)), mouse anti-SOX3 (Santa Cruz, sc-101155; 1:100 (2 µg ml⁻¹)), mouse anti-SYCP3 (Abcam, ab97672; 1:200 (5.0 µg ml⁻¹)), mouse anti-SYCP3 (Santa Cruz Biotechnology, sc-74569; 1:400 (0.5 µg µl⁻¹)), mouse anti-UTF1 (EMD Millipore, MAB4337; 1:100 (10 µg ml⁻¹)), goat anti-ZBTB16 (Santa Cruz Biotechnology, sc-11146; 1:200 (1 µg ml⁻¹)) and rabbit anti-ZBTB16 (Santa Cruz, sc-22839; 1:100 (2 µg ml⁻¹)).

After washing the slides, the sections were then incubated for 1 h at RT with the fluorescent-conjugated secondary antibodies diluted in PBS. Following another wash, the sections were stained with DAPI (1:300 dilution) for 3 min at RT. Subsequently, ProLong gold antifade mountant with DAPI (Thermo Fisher Scientific, P36931) was applied to preserve the fluorescence.

In addition, whole-mount staining of seminiferous tubules was carried out following a previous described protocol[87]. Briefly, seminiferous tubules were dissociated and fixed with 1% formaldehyde for 2 h at 4 °C. Primary antibodies, including goat anti-GATA4 (Santa Cruz, sc-1237; 1:200 (1 µg ml⁻¹)), mouse anti-SOX3 (Santa Cruz, sc-101155; 1:100 (2 µg ml⁻¹)) and rabbit anti-Versican V0, V1 Neo (Thermo Fisher Scientific, PA1-1748A 1:1,000 (2 µg ml⁻¹)), were diluted in the blocking buffer and incubated tubules. Like the previous steps, tubules were subjected to secondary antibody incubation, DAPI staining and mounting using ProLong gold antifade mountant with DAPI.

For visualization, Nikon A1R HF25 confocal and ECLIPSE Ti microscopes were used and the obtained images were using Nikon Elements AR software (www.microscope.healthcare.nikon.com). Quantification was conducted by evaluating a minimum of 100 circular tubule cross-sections per animal from at least three animals, using ImageJ software (version 1.53t)[88].

## Hi-C: pile-up analysis and CTCF ChIP-seq

THY1⁺ and KIT⁺ SPG were purified as describe above. The BSA gravity sedimentation method[23,24] was used to isolate PSs and RSs from the testis of adult male C57BL/6 mice. Hi-C libraries were generated using the Arima Hi-C kit (A410079) and Accel-NGS 2S plus DNA library kit (21024) following the manufacturer's instructions. Cells used 4 × 10⁵ to 1 × 10⁶ for THY1⁺ uSPG and KIT⁺ dSPG, 3 × 10⁶ for PSs and 4 × 10⁶ for RSs. Paired-end Hi-C libraries were sequenced on Illumina HiSeq4000 and processed with Juicer (version 1.5)[89] using BWA (version 0.7.3a)[90] for alignment with *Mus musculus* mm10. A .hic file, a highly compressed binary file, was created by Juicer Tools Pre. A .hic file was used to create a balanced cool file using cooler (version 0.8.11)[91] that was used for pile-up analysis. Using coolup.py (version 0.9.5)[92], we visualized the average interaction between TSS positions. For the pile-up analysis depicting interactions among TSS positions, each TSS position underwent analysis using cool files binned at a 10-kb resolution and a BED file displaying TSS positions, with an additional 100 kb of padding. The resulting piled-up data were visualized through plotup.py. CTCF ChIP-seq was performed in THY1⁺ uSPG and PSs using previously described methods[51].

## Reporting summary

Further information on research design is available in the Nature Portfolio Reporting Summary linked to this article.

## Data availability

The datasets generated and/or analyzed during this study are publicly available on the GEO database under the following accession numbers: bulk RNA-seq and ChIP-seq data, GSE202819; Hi-C and CTCF ChIP-seq data, GSE244681. All data sources, including the mm10 genome assembly, are appropriately cited and referenced. Source data are provided with this paper.

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

## Acknowledgements

We thank B. Dalley for sequencing, T. Mosbruger and T. Parnell for bioinformatics assistance, N. Verma, X. Nie, K. Aston, Y. Guo, Q. Chen, T. Mulvihill and H. Tran for the manuscript feedback, B.R.C. laboratory members for valuable scientific discussions and suggestions, C. Payne for anti-ZBTB16 antibody, J, Weiss for *Sox3*-floxed embryos and Y. DeRose, C. Wike and Y. Green for microscopy assistance. We acknowledge the HSC Cell Imaging Core (Nikon A1 and widefield microscopes, 1S10RR024761-01) and HSC Flow Cytometry Core (BD FACS Canto Scan, S10OD026959 for core facilities) at the University of Utah for use of equipment. We thank C. Rodesch and M. Bridge for their imaging assistance, J. Marvin for flow cytometry analysis assistance and J. Song for *Zbtb16* genotyping assistance. B.R.C. is an investigator with the Howard Hughes Medical Institute. This work was supported by the National Institute on Aging (R01 AG069725), by the Howard Hughes Medical Institute (genomics and biologicals), CA24014 to the Huntsman Cancer Institute for the support of core facilities used in this work and GM141085 to S.H.N.).

## Author contributions

C.Y., conceptualization, data curation, formal analysis, validation, investigation, visualization, methodology and writing—original draft; Y.K. and S.H.N., Hi-C analysis; S.M., CTCF ChIP-seq; B.R.C., conceptualization, resources, supervision, funding acquisition and writing—original draft.

## Competing interests

B.R.C. is a cofounder and chair of the scientific advisors of Paterna Biosciences. The other authors declare no competing interests.

## Additional information

**Extended data** is available for this paper at https://doi.org/10.1038/s41594-025-01509-5.

**Correspondence and requests for materials** should be addressed to Bradley R. Cairns.

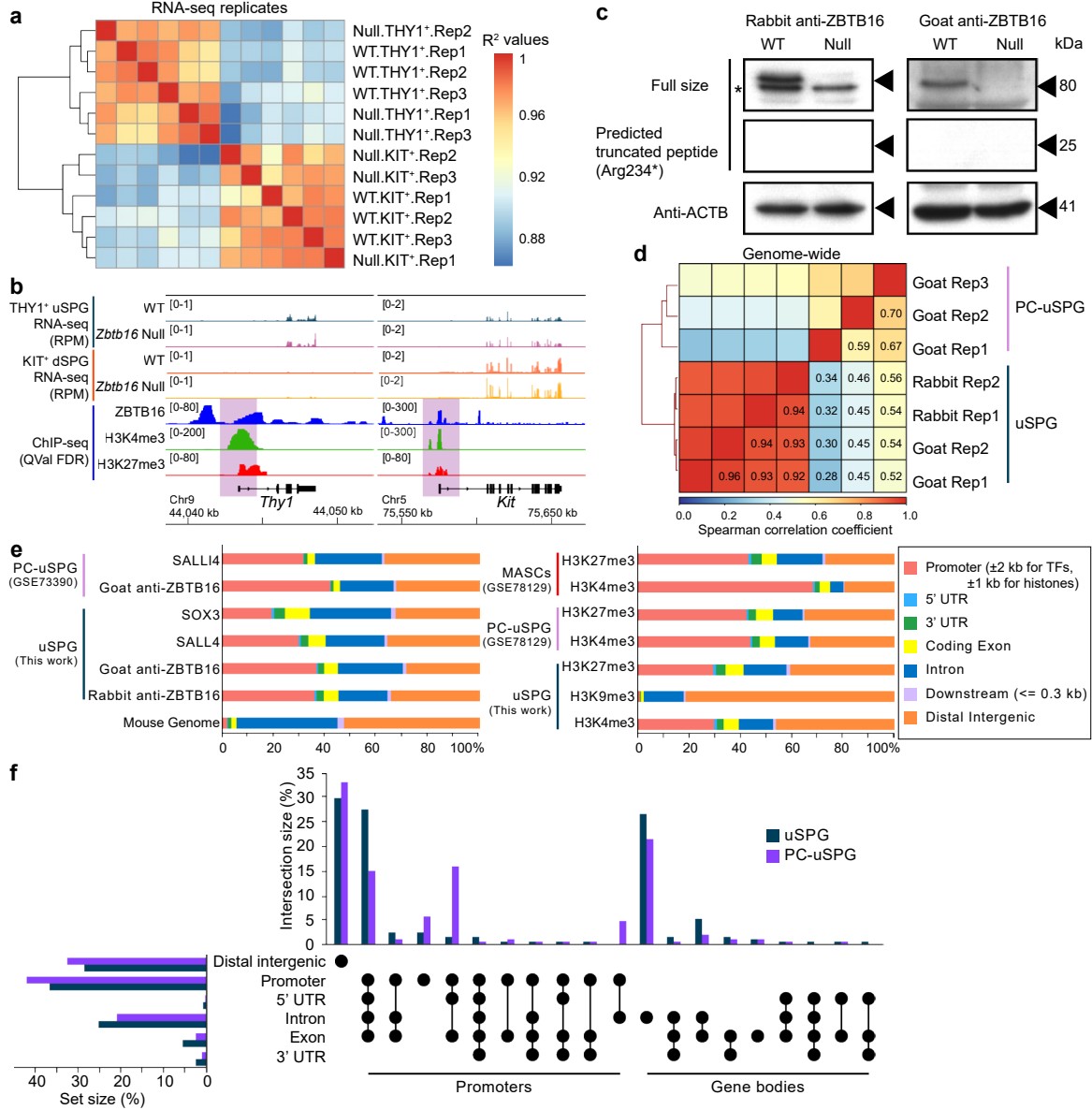

**Extended Data Fig. 1 | RNA-seq and ChIP-seq quality control, Related to Fig. 1. a**, Heatmap showing pairwise R² values of three biological replicates of RNA-seq in THY1⁺ uSPG and KIT⁺ dSPG from WT or null. **b**, Genome browser shots displaying cell-specific marker genes, including *Thy1* and *Kit*. **c**, Immunoblot demonstrating the specificity of two ZBTB16 antibodies raised in rabbits (left) and goats (right), confirming the loss of ZBTB16 expression in *Zbtb16* null samples. Arrowheads indicate the denoted protein sizes, and an asterisk (*) shows a nonspecific band. The 25 kDa band indicates the predicted truncated peptide of ZBTB16. This experiment was repeated independently twice with three biological

replicates, yielding similar results. **d**, Heatmap showing clustering of pairwise Spearman correlation of ZBTB16 ChIP-seq in uSPG and PC-uSPG. In uSPG, rabbit anti-ZBTB16 ChIP-seq data are correlated with goat anti-ZBTB16 ChIP-seq data, which are less correlated with goat anti-ZBTB16 ChIP-seq in PC-uSPG. **e**, Distribution of ChIP-seq peaks in uSPG, PC-uSPG, and MASCs. Left: ZBTB16, SALL4 and SOX3. Right: H3K4me3, H3K9me3, and H3K27me3. Promoters are defined within ±2 kb of the TSS for transcription factors and within ±1 kb of the TSS for histone modifications. **f**, Upset plot displaying multiple binding sites of ZBTB16 in uSPG and PC-uSPG.

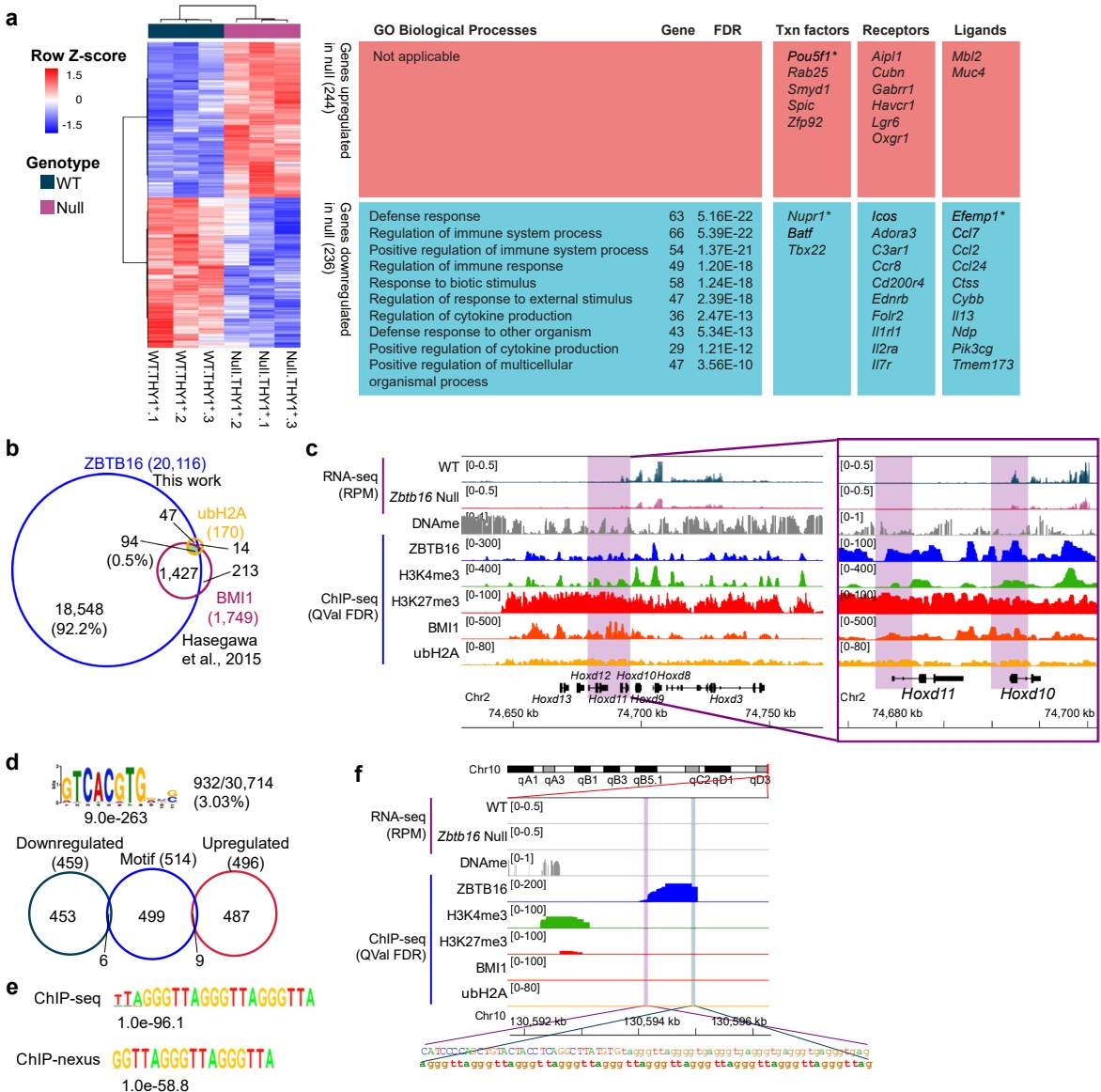

**Extended Data Fig. 2 | Indirect gene targets of ZBTB16 and ZBTB16 binding motifs, related Figs. 1–2. a**, RNA-seq heatmap depicting indirect gene targets of ZBTB16 differentially expressed in WT and *Zbtb16* null THY1⁺ uSPG (left), alongside top enriched GO terms, transcription factors, receptors, and ligands from upregulated and downregulated genes (right). Asterisks (*) denote genes related to male infertility. **b**, Venn diagram showing the overlap of ZBTB16-bound genes with ubH2AK119 and BMI1 in THY1⁺ uSPG reprocessed from GSE55060 ref. 93. **c**, Genome browser view of *Hoxd* clusters (left) with an enlarged view

of the purple boxed region showing *Hoxd10* and *Hoxd11* (right). **d**, Putative ZBTB16 binding motifs identified using MEME-ChIP[30] (top), with a Venn diagram showing the motif intersections with *Zbtb16*-affected genes (bottom). **e**, ZBTB16 binding motifs identified from regular ChIP-seq peaks (top) and ChIP-nexus data (bottom) using GEM[31]. **f**, Genome browser view of a ZBTB16-bound site at Chromosome 10 telomere (top), with an enlarged view of the purple boxed region showing the DNA sequences (bottom).

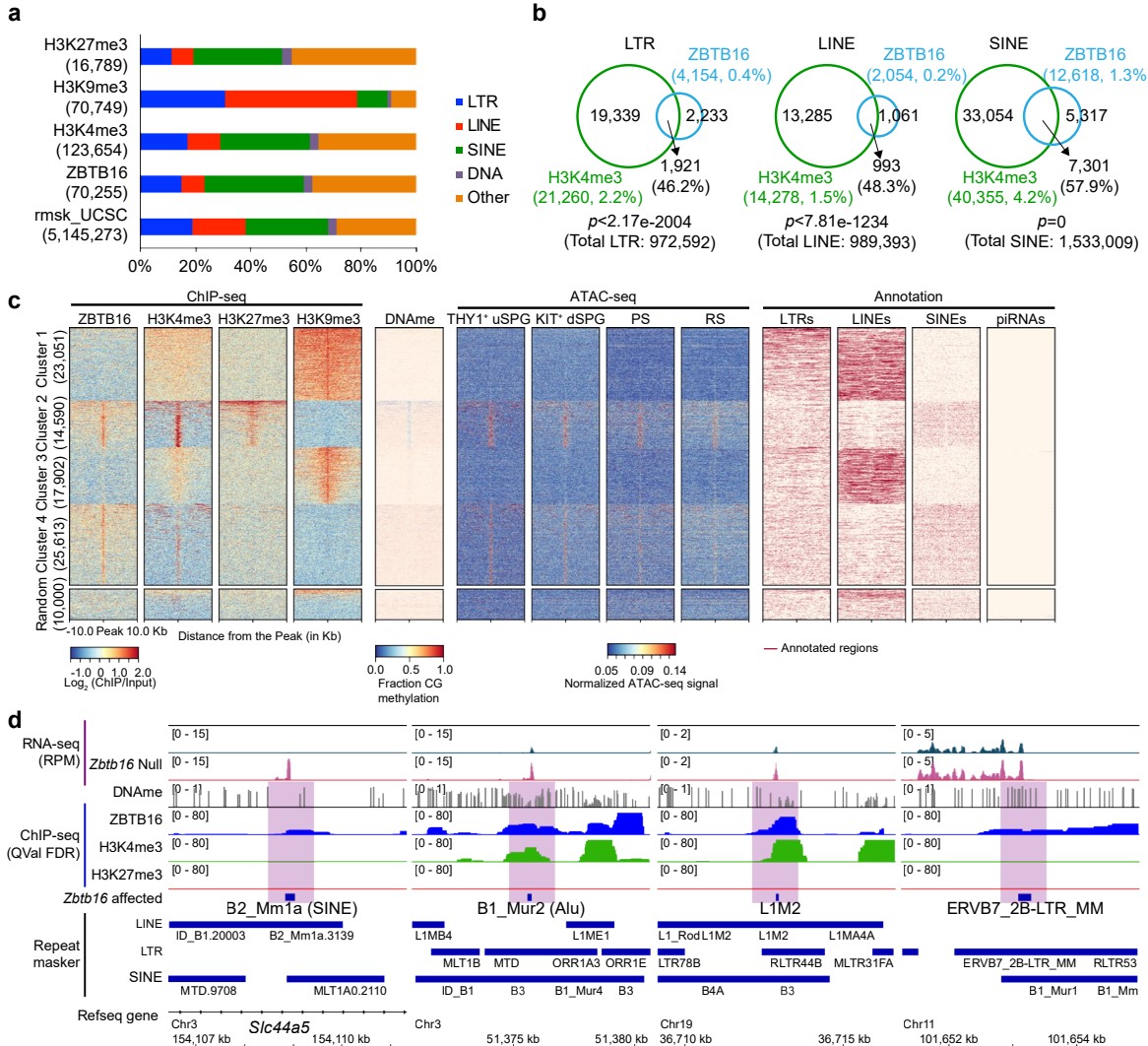

**Extended Data Fig. 3 | ZBTB16 binds and suppresses particular retrotransposons, Related to Fig. 2. a**, Distribution of ChIP-seq peaks from ZBTB16, H3K4me3, H3K9me3, and H3K27me3. Promoters are defined within ±1 kb of retrotransposons. **b**, Venn diagram showing ZBTB16-bound LTR, LINE, and SINE overlapping with H3K4me3 in THY1[+] uSPG. Statistical analysis was performed using a hypergeometric test. **c**, Heatmap showing *k*-means clustering (*k* = 4) of ZBTB16, H3K4me3, H3K9me3, and H3K27me3 at the center (±10 kb) of binding sites from the union of all ChIP-seq data sets with DNA methylation[19], ATAC-seq[23], and annotation of LTRs, LINEs, SINEs, and piRNAs. **d**, Genome browser panels showing B2_Mm1a and B1_Mur2 (SINE), L1M2 (LINE), and ERVB7_2B (LTR) suppressed by ZBTB16 with tracks for chromatin landscape marks, including H3K4me3, H3K27me3, and DNAme.

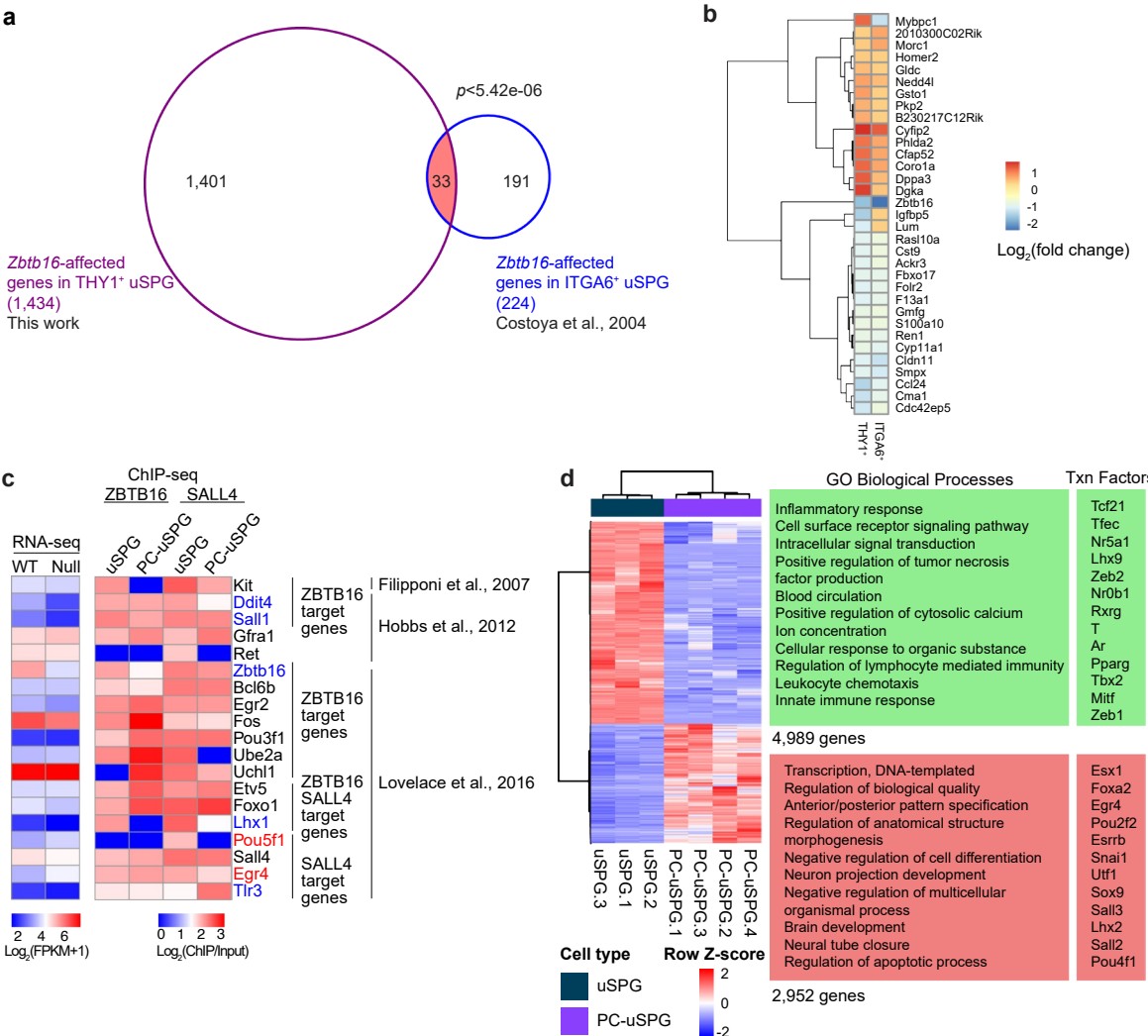

**Extended Data Fig. 4 | Comparison of ZBTB16 target genes in uSPG and PC-uSPG, Related to Fig. 3. a**, Venn diagram showing *Zbtb16*-affected genes in THY1⁺ uSPG using RNA-seq (this work) overlapping with those in ITGA6⁺ uSPG using microarray[10]. Statistical analysis was performed using a hypergeometric test. **b**, Heatmap showing 33 common ZBTB16 target genes in THY1⁺ uSPG and ITGA6⁺ uSPG, as analyzed in **a**. **c**, Heatmap showing ZBTB16 target genes compared with prior works[12,13,34]. RNA-seq heatmap showing gene expression in WT and *Zbtb16*-deficient THY1⁺ uSPG (left) with ChIP-seq heatmap (right) showing ZBTB16 and SALL4 enrichments in uSPG (this work) and PC-uSPG[12]. **d**, RNA-seq heatmap depicting differential expression in WT THY1⁺ uSPG and PC-uSPG[5] (left) with enriched GO term and transcription factors (right).

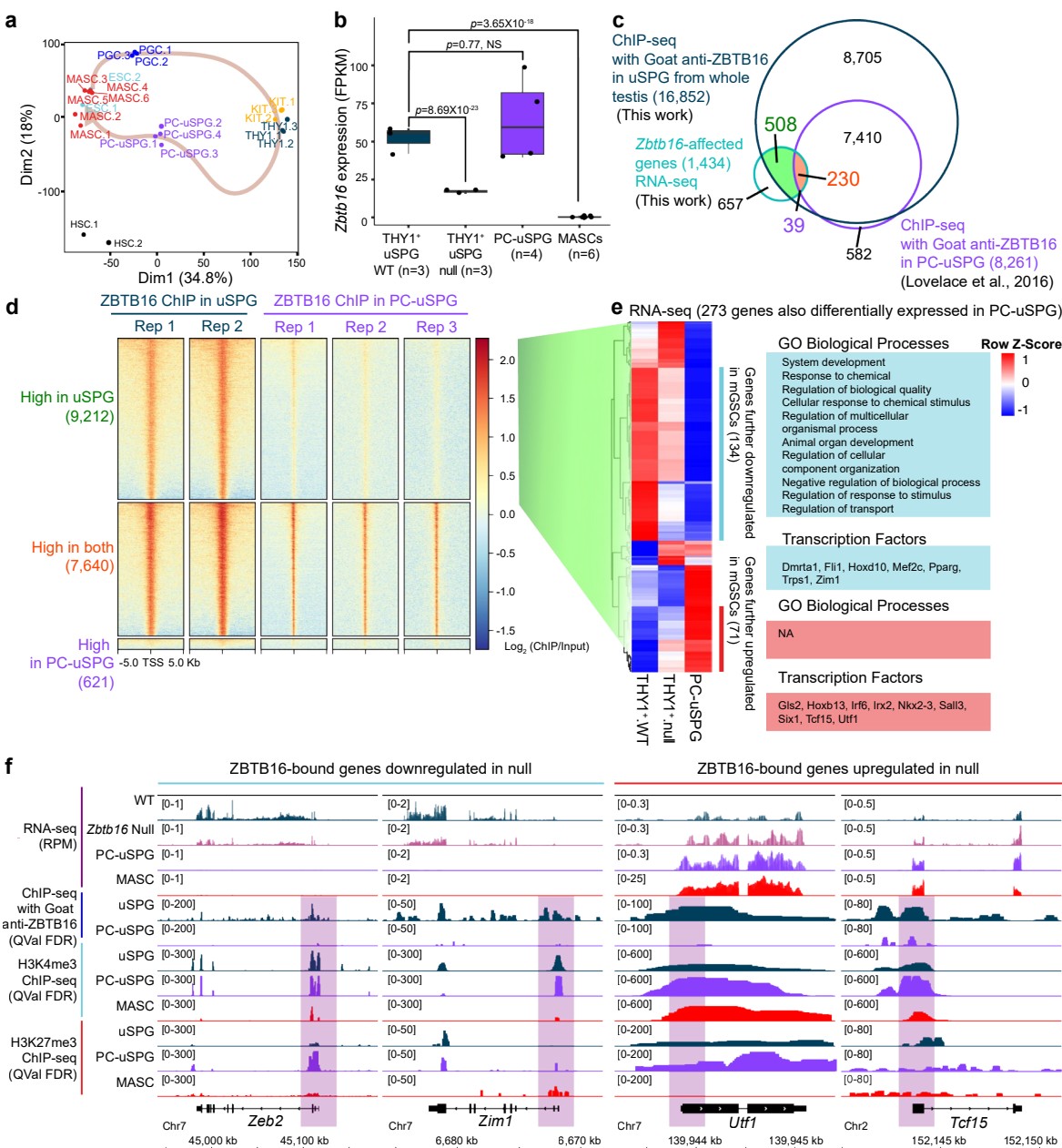

**Extended Data Fig. 5 | Distinctive gene regulation by ZBTB16 in uSPG versus PC-uSPG, Related to Fig. 3. a**, PCA plot showing transcriptome relationships among ESCs, PGCs, THY1⁺ uSPG, KIT⁺ dSPG, PC-uSPG, and MASCs. **b**, Box-and-whiskers plot showing *Zbtb16* expression level (FPKM) in WT THY1⁺ uSPG, *Zbtb16*-deficient THY1⁺ uSPG, PC-uSPG, and MASCs. The boxes represent the IQR with the central line indicating the median. Whiskers extend to 1.5 times the IQR. Each dot indicates a biological replicate. **c**, Venn diagram showing ZBTB16 target genes in uSPG (this work) overlapped with those in PC-uSPG reprocessed from GSE73390 ref. 12, intersected with differentially expressed genes in *Zbtb16*-deficient THY1⁺ uSPG compared with WT using RNA-seq (this work).

**d**, ChIP-seq heatmap showing differential enrichment at promoters of ZBTB16 *in vivo* (n = 2, this work) and PC-uSPG (n = 3). **e**, RNA-seq heatmap across the same clusters as analyzed in **d** displaying loss of ZBTB16 binding and differentially expressed genes in PC-uSPG compared to WT THY1⁺ uSPG and *Zbtb16* null THY1⁺ uSPG (left, shown in **c**) with enriched GO term and transcription factors (right). **f**, Representative genome browser shots displaying loss of ZBTB16 occupancy and differentially expressed genes in PC-uSPG: two downregulated genes (*Zeb2* and *Zim1*) and two upregulated genes in null (*Utf1* and *Tcf15*) that are further affected in PC-uSPG and MACSs.

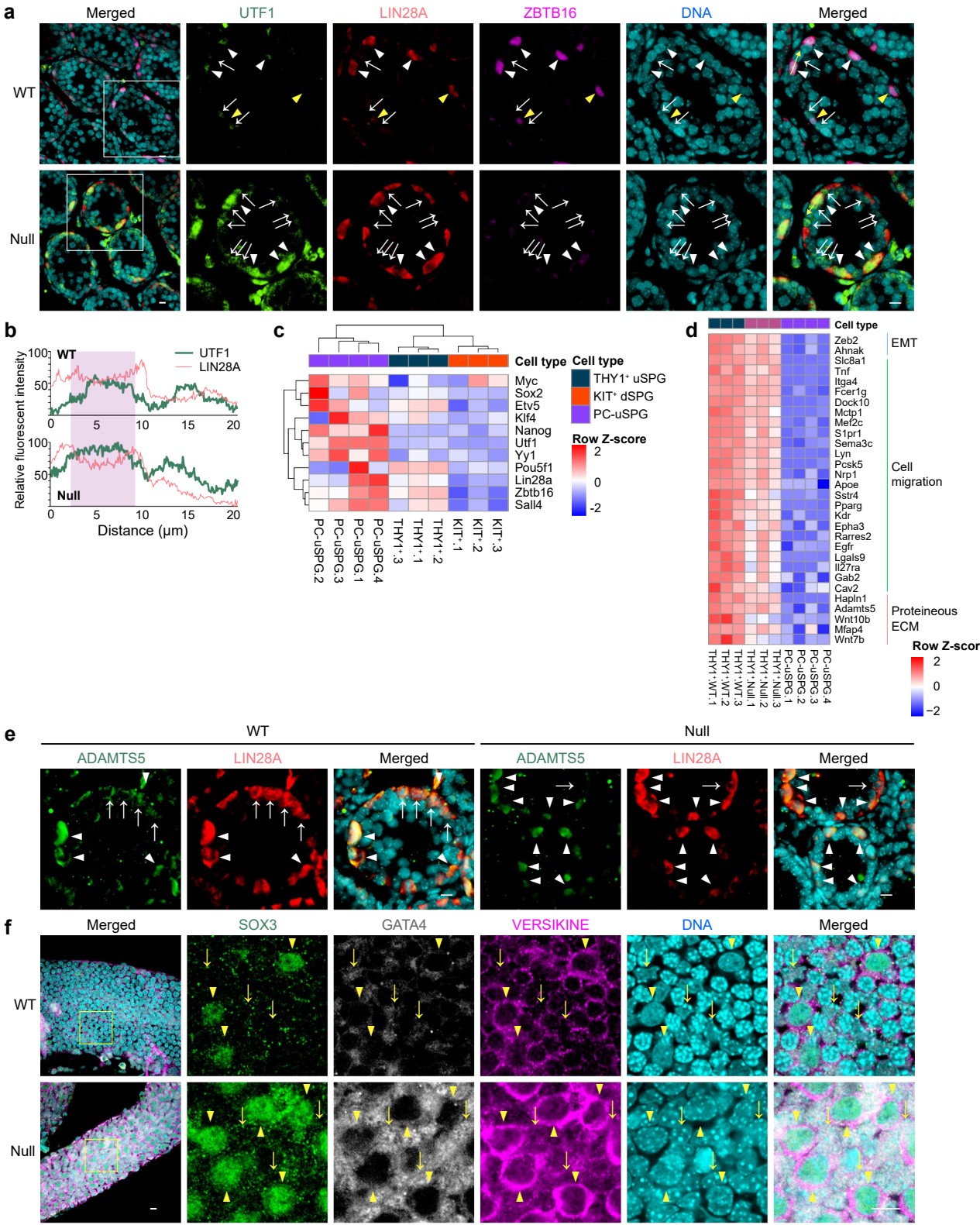

**Extended Data Fig. 6 | See next page for caption.**

**Extended Data Fig. 6 | PC-uSPG lose a majority of ZBTB16 occupancy, with moderate effects on gene expression, Related to Fig. 3. a**, Representative immunofluorescence images showing UTF1, LIN28A, and ZBTB16 expression pattern in testis sections from *Zbtb16* null mice and littermate control (WT) at P14: UTF1 (green); LIN28A (red); ZBTB16 (magenta); and DNA (cyan). White arrowheads indicate UTF1⁺/LIN28A⁺/ZBTB16⁺ uSPG. Yellow arrowheads indicate UTF1⁻/LIN28A⁺/ZBTB16⁺ uSPG. White arrows indicate UTF1⁺/LIN28A⁻/ZBTB16⁻ dSPG. Scale bars = 10 µm. The experiment was repeated independently three times with similar results, using three biological replicates. **b**, Line plot analysis on yellow-arrowed area showing the relative fluorescent intensity of UTF1 (green) and LIN28A (red) in **a**. Data are representative of three biological replicates. **c**, Heatmap showing differential gene expression of pluripotency transcription factors in THY1⁺ uSPG, KIT⁺ dSPG, and PC-uSPG. **d**, Heatmap showing differential gene expression of genes involved in cell migration in THY1⁺ uSPG versus PC-uSPG, as analyzed in Extended Data Fig. 5e. **e**, Representative immunofluorescence images showing ADAMTS5 and LIN28A expression patterns in testis sections from *Zbtb16* null mice and littermate controls (WT) at P14: ADAMTS5 (green); LIN28A (red); and DNA (cyan). Arrowheads indicate ADAMTS5⁺/LIN28A⁺ uSPG. Arrows indicate ADAMTS5⁻/LIN28A⁺ dSPG. Scale bars = 10 µm. The experiment was repeated independently three times with similar results, using three biological replicates. **f**, Representative whole-mount staining images showing SOX3, GATA4, and VERSIKINE expression pattern in seminiferous tubules from *Zbtb16* null mice and littermate controls (WT) at P14: SOX3 (green); GATA4 (grey); VERSIKINE (magenta); and DNA (cyan). Arrowheads indicate SOX3⁺/GATA4⁻/VERSIKINE⁺ uSPG. Arrows indicate SOX3⁻/GATA4⁺/VERSIKINE⁻ Sertoli cells. Scale bars = 10 µm. The experiment was repeated independently three times with similar results, using three biological replicates.

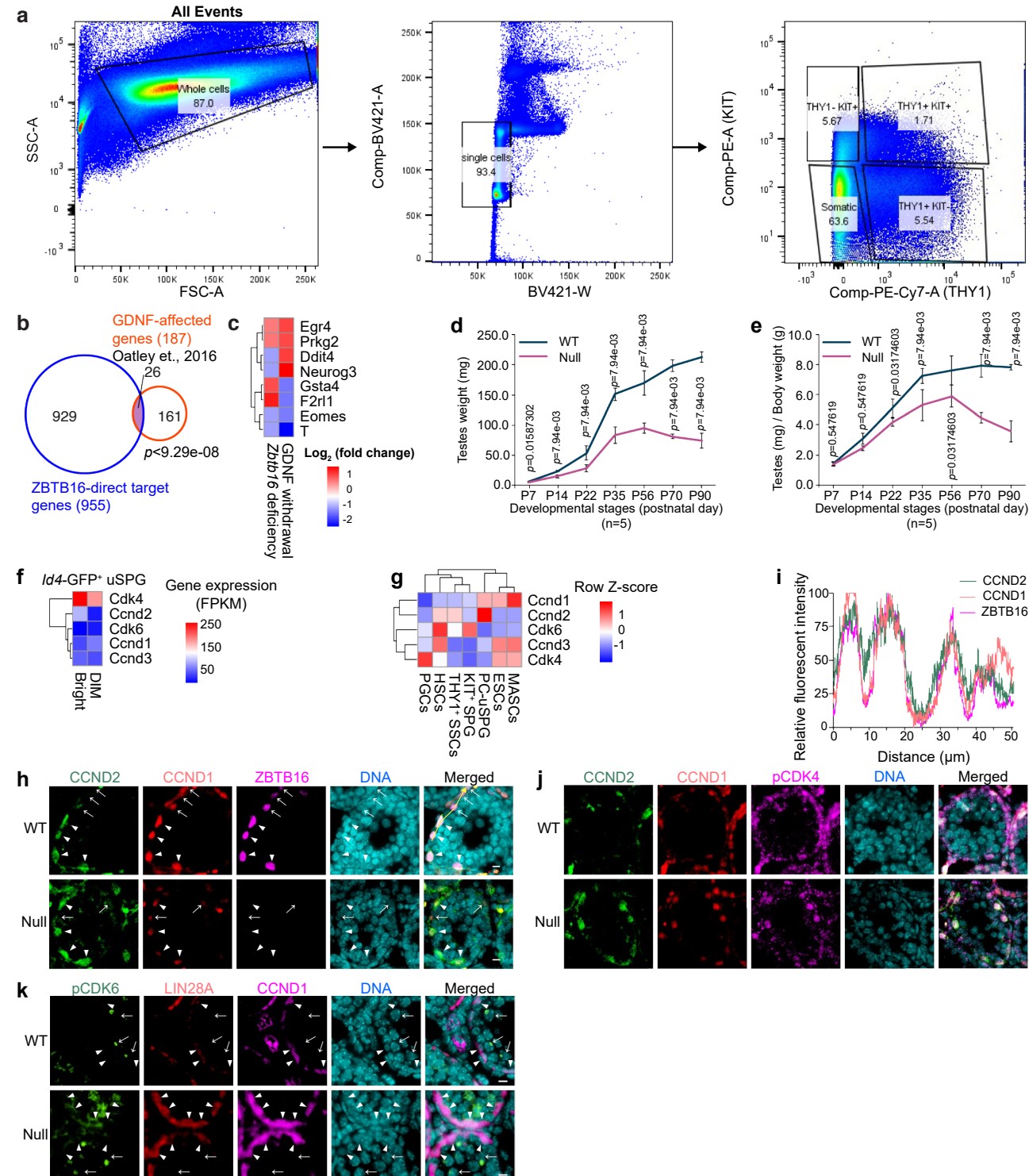

**Extended Data Fig. 7 | See next page for caption.**

**Extended Data Fig. 7 | ZBTB16 target genes, testis development and gene expression of cyclin Ds and CDKs in uSPG, Related to Fig. 3. a**, Flow cytometry plots from mouse SPG at P7 illustrating the gating strategy. Cells were identified by forward scatter (FSC) and back scatter (BSC), followed by doublets exclusion and separation of uSPG (THY1[+]) and dSPG (KIT[+]). DNA contents analysis assessed cell cycle stages. **b**, Venn diagram showing *Zbtb16*-affected genes (this study) and GDNF-affected genes[56]. Statistical significance was assessed using a hypergeometric test. **c**, Heatmap of selected *Zbtb16*-affected genes overlapping with GDNF-affected genes, as analyzed in **a**. **d**, Testis weights of *Zbtb16*-null mice and WT littermates at indicated ages (*n* = 5 biological replicates per genotype). Statistical significance was determined using the Wilcoxon rank-sum test. **e**, Testis weights normalized to body weights (*n* = 5 biological replicates per genotype). Statistical significance was determined using the Wilcoxon rank-sum test. **f**, Heatmap of cyclin D and CDK gene expression in *Id4*-GFP[+] Bright uSPG versus Dim uSPG at P7[38]. **g**, Heatmap of cyclin D and CDK gene expression across developmental stage (ESCs, PGCs, WT uSPG, *Zbtb16*-null uSPG, dSPG, PC-uSPG,

and MASCs). **h**, Immunofluorescence images showing CCND1 and CCND2 expression in WT and *Zbtb16* null mice at P14: CCND2 (green), CCND1 (red), ZBTB16 (magenta), and DNA (cyan). Arrowheads indicate CCND2[+]/CCND1[+]/ZBTB16[+] self-renewing uSPG. Arrows indicate CCND2[−]/CCND1[+]/ZBTB16[+] late uSPG. Scale bars = 10 μm. **i**, Line plot of CCND2 (green), CCND1 (red), and ZBTB16 (magenta) levels from yellow arrows in **h**. **j**, Immunofluorescence of pCDK4 and CCND1/2 in WT and *Zbtb16*-null at P14: CCND2 (green), CCND1 (red), pCDK4 (magenta), and DNA (cyan). Arrowheads indicate CCND2[+]/CCND1[+]/pCDK4[+] early uSPG. Arrows indicate CCND2[−]/CCND1[+]/pCDK4[+] progenitors. Scale bars = 10 μm. **k**, Immunofluorescence showing pCDK6 localization in leptotene and pachytene spermatocytes but not uSPG at P14: pCDK6 (green), LIN28A (red), CCND1 (magenta), and DNA (cyan). Arrowheads: CDK6[+] leptotene spermatocytes; arrows: pCDK6[+] pachytene spermatocytes. Scale bars = 10 μm. The experiment was repeated independently three times with similar results, using three biological replicates.

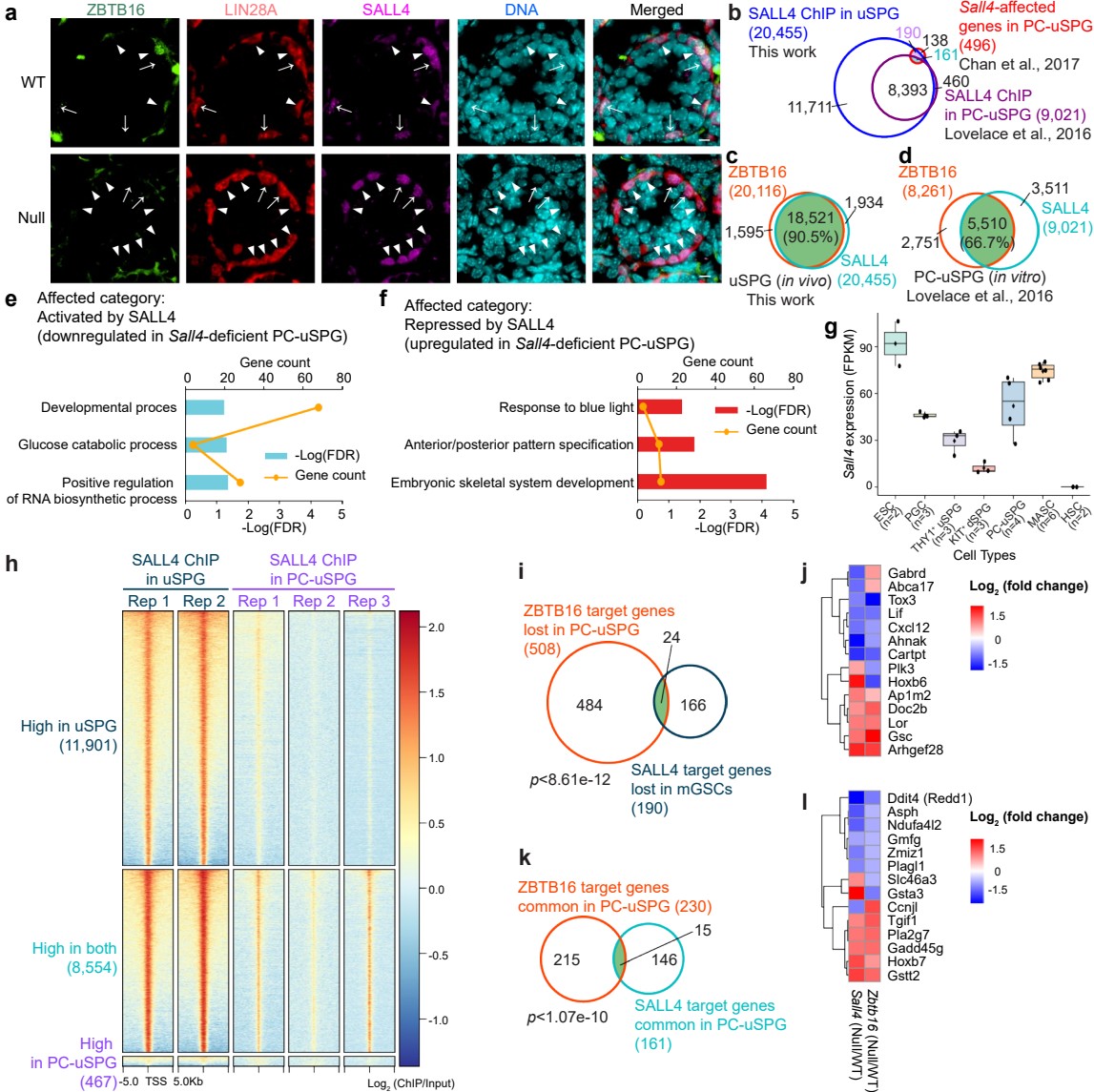

**Extended Data Fig. 8 | SALL4 expression and GO enrichment analysis in SALL4 target genes, Related to Fig. 4. a**, Representative immunofluorescence images showing differential expression of ZBTB16, LIN28A, and SALL4 expression pattern in testis sections from *Zbtb16* null mice and littermate control (WT) at P14: ZBTB16 (green); LIN28A (red); SALL4 (magenta); and DNA (cyan). Arrowhead indicates ZBTB16⁺/LIN28A⁺/SALL4⁺ uSPG. Arrow indicates ZBTB16⁻/LIN28A⁺/SALL4⁺ differentiating SPG. Scale bars = 10 μm. The experiment was repeated independently three times with similar results, using three biological replicates. **b**, Venn diagram showing the extent of overlap of SALL4 comparing *in vivo* uSPG (this work) to PC-uSPG reprocessed from GSE73390 ref. 12 and *Sall4*-affected genes in PC-uSPG[42]. **c**, Venn diagram showing ZBTB16-bound genes overlapping with SALL4-bound genes in uSPG, as analyzed in Fig. 1c and Extended Data Fig. 8b. **d**, Venn diagram showing ZBTB16-bound genes overlapping with SALL4-bound genes in PC-uSPG, as analyzed in Extended Data Fig. 5c and Fig. 8b. **e**, GO terms for functional clustering of genes downregulated in *Sall4*-deficient PC-uSPG[42] associated with *in vivo* SALL4 ChIP-seq peaks (this work). **f**, GO terms for functional clustering of genes upregulated in *Sall4*-

deficient PC-uSPG[42] associated with *in vivo* SALL4 ChIP-seq peaks (this work). **g**, Box-and-whiskers plot showing *Sall4* expression level (FPKM) in ESCs[5], PGCs (SRA097278), THY1⁺ uSPG, KIT⁺ dSPG, PC-uSPG[5], MASCs[5], and HSCs[5]. The boxes represent the IQR with the central line indicating the median. Whiskers extend to 1.5 times the IQR. Each dot indicates a biological replicate. **h**, ChIP-seq heatmap showing differential enrichment at promoters of SALL4 *in vivo* (n = 2, this work) and PC-uSPG (n = 3), as analyzed in **b**. **i**, Venn diagram showing ZBTB16-bound genes overlapping with SALL4-bound genes that are lost in PC-uSPG, as analyzed in Extended Data Fig. 5c and Fig. 8b. Statistical analysis was performed using a hypergeometric test. **j**, Heatmap showing the positive correlation between selected ZBTB16 target genes and SALL4 target genes, as analyzed in **i**. **k**, Venn diagram showing the overlap of SALL4 direct target genes in PC-uSPG with ZBTB16 direct target genes in PC-uSPG as analyzed in Extended Data Fig. 5c and Fig. 8b. Statistical analysis was performed using a hypergeometric test. **l**, Heatmap showing the positive correlation between ZBTB16 target genes and SALL4 target genes, as analyzed in **k**.

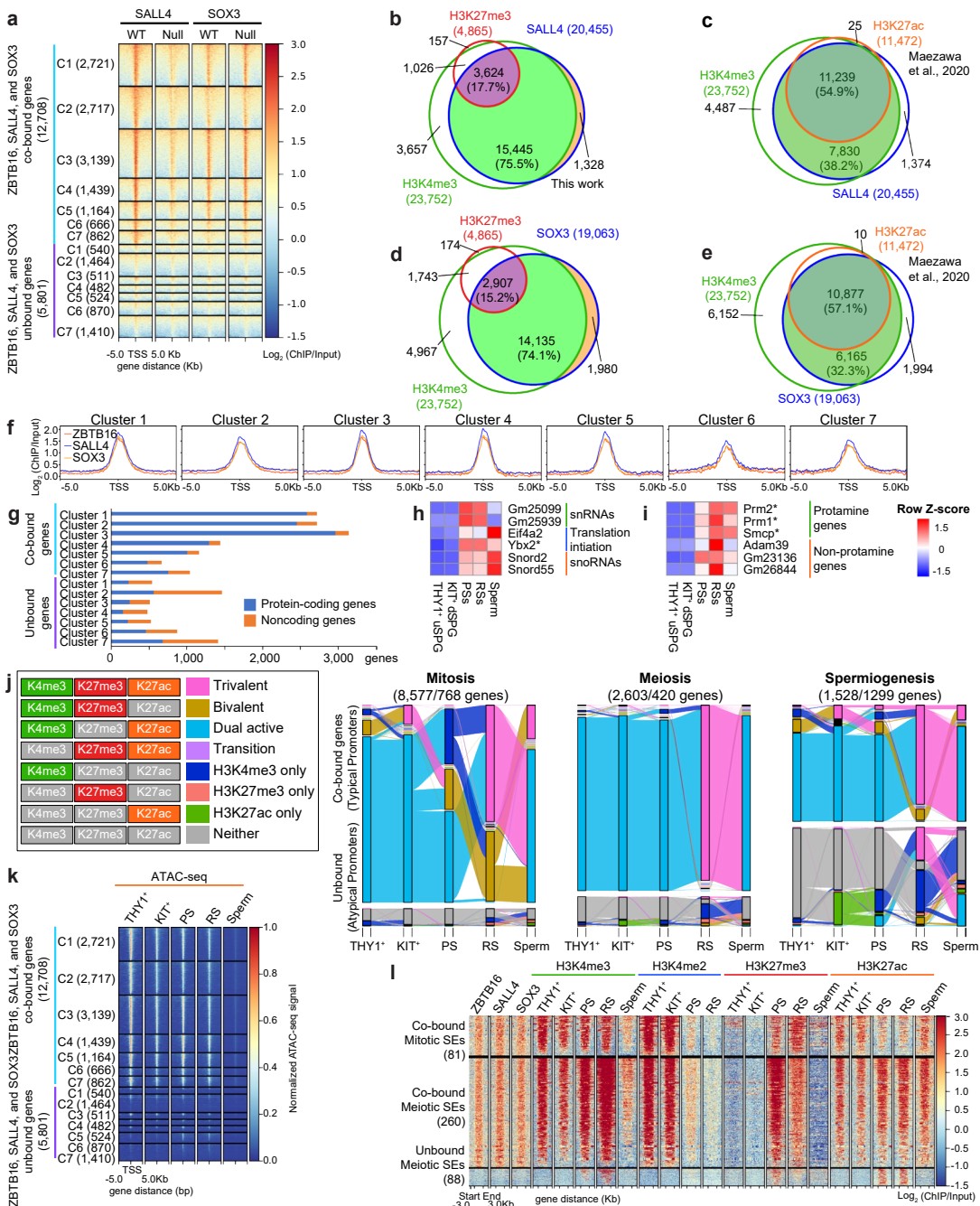

**Extended Data Fig. 9 | ZBTB16, SALL4 and SOX3-bound genes with histone modifications and chromatin accessibility in Typical and Atypical genes, Related to Fig. 5. a**, ChIP-seq heatmap showing enrichment of Typical gene promoters in WT and *Zbtb16* null mice, highlighting similar SALL4 and SOX3 occupancy, as analyzed in Fig. 5d. **b**, Venn diagram showing the overlap of SALL4-bound genes with H3K4me3 and H3K27me3 in THY1+ uSPG (this work). **c**, Venn diagram showing the overlap of SALL4-bound genes with H3K4me3 (this work) and H3K27ac[51] in THY1+ uSPG. **d**, Venn diagram showing the overlap of SOX3-bound genes with H3K4me3 and H3K27me3 in THY1+ uSPG (this work). **e**, Venn diagram showing the overlap of SOX3-bound genes with H3K4me3 (this work) and H3K27ac[51] in THY1+ uSPG. **f**, Line graph depicting the co-occupancy of ZBTB16, SALL4, and SOX3 at TSS within 5 Kb across clusters 1–7, as analyzed

in Fig. 5d. **g**, Distribution of protein-coding and non-coding genes in Typical and Atypical genes as analyzed in Fig. 5d. **h**, Heatamp showing the relative average expression for selected snRNA, translation initiation factors and snoRNA genes across the Typical genes in RSs. **i**, Heatamp displaying the relative average expression for spermiogenic genes across the Atypical genes in RSs. Asterisks (*) indicate genes related to male infertility. **j**, Alluvial plots showing histone dynamics during mitosis (clusters 1–3), meiosis (clusters 4 and 5), and spermiogenesis (clusters 6 and 7) in Typical and Atypical genes, as analyzed in Fig. 5d. **k**, ATAC-seq[23,47] heatmap showing chromatin accessibility in Typical and Atypical genes, as analyzed in Fig. 5d. **l**, ChIP-seq heatmap showing differential enrichment at mitotic and meiotic super-enhancers[51] of Typical and Atypical genes, along with histone modifications across spermatogenesis.

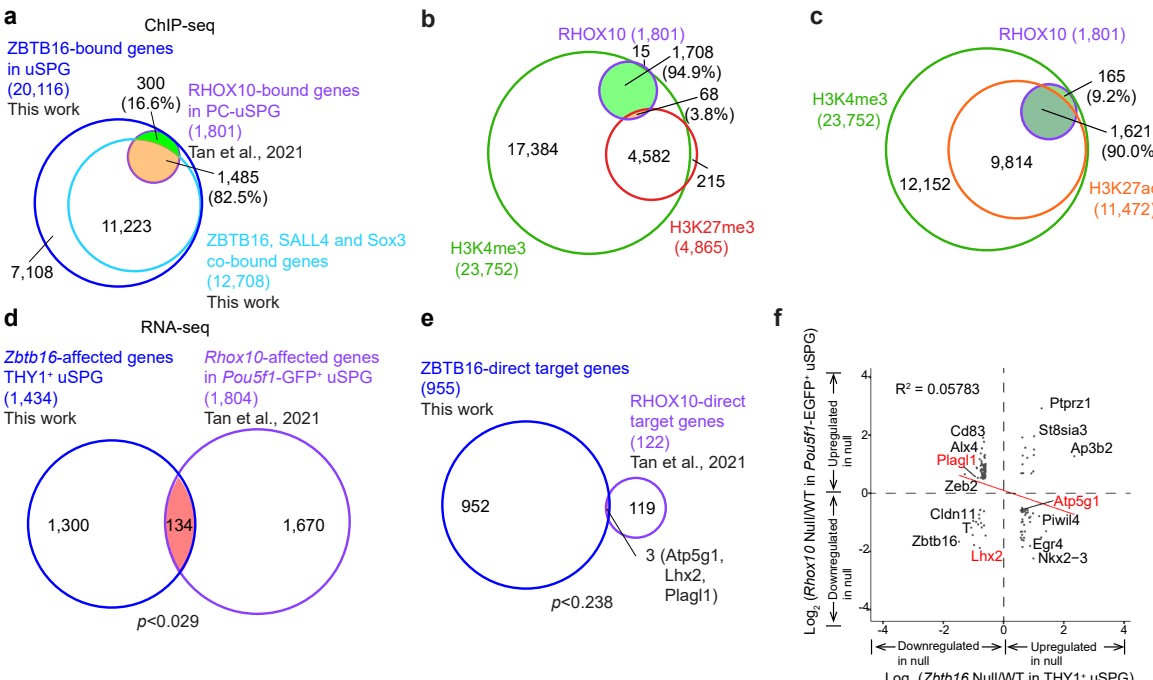

**Extended Data Fig. 10 | Target genes of RHOX10 overlap with those of ZBTB16 targets, Related to Fig. 5. a**, Venn diagram showing the overlap of ZBTB16-bound genes (this work) and ZBTB16, SALL4 and SOX3 co-bound genes with RHOX10-bound genes[52]. **b**, Venn diagram showing the overlap of RHOX10-bound genes with H3K4me3 and H3K27me3 in THY1[+] uSPG (this work). **c**, Venn diagram showing the overlap of RHOX10-bound genes[52] with H3K4me3 (this work) and H3K27ac[51] in THY1[+] uSPG. **d**, Venn diagram showing the overlap of *Zbtb16*-affected genes (this work) with *Rhox10*-affected genes[52]. Statistical analysis was performed using a hypergeometric test. **e**, Venn diagram showing the overlap of ZBTB16 direct target genes (this work) with RHOX10 direct target genes[52], as analyzed in **a** and **d**. Statistical analysis was performed using a hypergeometric test. **f**, Scatter plot comparing the log₂ fold change between differentially expressed genes in both *Zbtb16*-deficient uSPG and *Rhox10*-deficient *Pou5f1*-EGFP[+] uSPG[52] compared with WT, as analyzed in **d**. Red genes indicate direct target genes co-bound with ZBTB16 and RHOX10.

# Reporting Summary

## Statistics

For all statistical analyses, confirm that the following items are present in the figure legend, table legend, main text, or Methods section.

| n/a | Confirmed | |
|---|---|---|
| ☐ | ☒ | The exact sample size (*n*) for each experimental group/condition, given as a discrete number and unit of measurement |
| ☐ | ☒ | A statement on whether measurements were taken from distinct samples or whether the same sample was measured repeatedly |
| ☐ | ☒ | The statistical test(s) used AND whether they are one- or two-sided<br>*Only common tests should be described solely by name; describe more complex techniques in the Methods section.* |
| ☒ | ☐ | A description of all covariates tested |
| ☐ | ☒ | A description of any assumptions or corrections, such as tests of normality and adjustment for multiple comparisons |
| ☐ | ☒ | A full description of the statistical parameters including central tendency (e.g. means) or other basic estimates (e.g. regression coefficient) AND variation (e.g. standard deviation) or associated estimates of uncertainty (e.g. confidence intervals) |
| ☐ | ☒ | For null hypothesis testing, the test statistic (e.g. *F*, *t*, *r*) with confidence intervals, effect sizes, degrees of freedom and *P* value noted<br>*Give P values as exact values whenever suitable.* |
| ☒ | ☐ | For Bayesian analysis, information on the choice of priors and Markov chain Monte Carlo settings |
| ☒ | ☐ | For hierarchical and complex designs, identification of the appropriate level for tests and full reporting of outcomes |
| ☒ | ☐ | Estimates of effect sizes (e.g. Cohen's *d*, Pearson's *r*), indicating how they were calculated |

*Our web collection on statistics for biologists contains articles on many of the points above.*

## Software and code

Policy information about availability of computer code

| | |
|---|---|
| Data collection | RNA-seq and ChIP-seq reads were aligned to the mm10 mouse genome using Novoalign (v4.04.01) with default parameters. Differential gene expression analysis was performed using USeq (v8.9.6), incorporating DESeq2 (v1.42.1). ChIP-seq peaks were identified with USeq (v8.9.6) and MACS2 (v2.1.1) using default settings. Genome browser visualizations were generated with IGV (v2.16.2).<br>Published data were retrieved from the NCBI Gene Expression Omnibus (GEO) Sequence Read Archive (SRA) using GNU Wget and SRA Toolkit (v2.10.8), with the fasterq-dump utility for sequence extraction.<br>Hi-C data were processed using Juicer (v1.5) with BWA (v0.7.3a) for alignment. The resulting .hic file was generated using Cooler (v0.8.11) and visualized with Cooler tools (coolup.py, v0.9.5). |
| Data analysis | http://novocraft.com (Novoalign v4.04.01)<br>https://www.gnu.org/software/wget/ (GNU Wget)<br>https://hpc.nih.gov/apps/sratoolkit.html (SRA Toolkit v2.10.8)<br>https://useq.sourceforge.net (USeq v8.9.6)<br>https://hbctraining.github.io/Intro-to-ChIPseq/lessons/05_peak_calling_macs.html (MACS2 v2.1.1)<br>http://www.htslib.org (Samtools v1.19)<br>https://bioconductor.org/packages/release/bioc/html/DESeq2.html (DESeq2 v1.42.1)<br>https://broadinstitute.github.io/picard/ (Picard v2.26.3)<br>https://deeptools.readthedocs.io/en/latest/ (deepTools v3.5.4)<br>https://igv.org (IGV v2.16.2)<br>https://bedtools.readthedocs.io/en/latest/ (BEDTools v2.29.0)<br>https://pantherdb.org (PANTHER v17.0)<br>https://bioinforx.com/apps/venn_overlap.php(BxToolBox: Venn diagram) |

```
https://github.com/tjparnell/biotoolbox (biotoolbox v1.69)
https://meme-suite.org/meme/tools/meme-chip (MEME-ChIP v11)
https://groups.csail.mit.edu/cgs/gem/ (GEM v2.6)
https://ggplot2.tidyverse.org (ggplot2 v3.5.1)
https://cran.r-project.org/web/packages/pheatmap/index.html (pheatmap v1.0.12)
https://www.bdbiosciences.com/en-us/products/software/instrument-software/bd-facsdiva-software (FACSDiva v8.01)
https://www.flowjo.com (FlowJo v9.9)
https://imagej.net/software/fiji/ (Image J-Fiji v1.53t)
http://nemates.org/MA/progs/overlap_stats.html (hypergeometric probability tests)
https://github.com/aidenlab/juicer/releases (juicer v1.5)
https://pypi.org/project/cooler/ (cooler v0.8.11)
https://pypi.org/project/coolpuppy/ (coolup.py v0.9.5)
```

For manuscripts utilizing custom algorithms or software that are central to the research but not yet described in published literature, software must be made available to editors and reviewers. We strongly encourage code deposition in a community repository (e.g. GitHub). See the Nature Portfolio guidelines for submitting code & software for further information.

## Data

Policy information about availability of data

All manuscripts must include a data availability statement. This statement should provide the following information, where applicable:
- Accession codes, unique identifiers, or web links for publicly available datasets
- A description of any restrictions on data availability
- For clinical datasets or third party data, please ensure that the statement adheres to our policy

The datasets generated and/or analyzed during this study are publicly available on the GEO database under the following accession numbers: bulk RNA-seq and ChIP-seq data (GSE202819) and Hi-C data (GSE244681). All data sources, including the mm10 genome assembly, are appropriately cited and referenced in the manuscript.

## Research involving human participants, their data, or biological material

Policy information about studies with human participants or human data. See also policy information about sex, gender (identity/presentation), and sexual orientation and race, ethnicity and racism.

| | |
|---|---|
| Reporting on sex and gender | N/A |
| Reporting on race, ethnicity, or other socially relevant groupings | N/A |
| Population characteristics | N/A |
| Recruitment | N/A |
| Ethics oversight | N/A |

Note that full information on the approval of the study protocol must also be provided in the manuscript.

# Field-specific reporting

Please select the one below that is the best fit for your research. If you are not sure, read the appropriate sections before making your selection.

☒ Life sciences          ☐ Behavioural & social sciences          ☐ Ecological, evolutionary & environmental sciences

For a reference copy of the document with all sections, see nature.com/documents/nr-reporting-summary-flat.pdf

# Life sciences study design

All studies must disclose on these points even when the disclosure is negative.

| | |
|---|---|
| Sample size | Sample sizes were determined based on standard practices in the field and the feasibility of the experimental design. For each experiment, we ensured that the number of biological replicates was sufficient to capture variability and allow robust statistical analyses. Although no formal sample size calculation was performed, the chosen sample sizes are consistent with similar studies and provide adequate statistical power to support the conclusions. The sufficiency of these sample sizes was confirmed through statistical tests that showed clear differentiation between experimental groups. For all genomics experiments, ChIP-seq and Hi-C were performed in duplicate, while bulk RNA-seq experiments were performed in triplicate to ensure reproducibility. Flow cytometry experiments were conducted in three biological replicates. Microscopy experiments were conducted in three biological replicates, with quantifications based on at least 300 seminiferous tubules per measurement to ensure accurate and reliable assessment of trends. |
| Data exclusions | No data were excluded from analyses. |

| | |
|---|---|
| Replication | We verified consistent results across three independent biological replicates for flow cytometry and immunofluorescence analyses. Additionally, the ChIP-seq experiments for ZBTB16, SALL4, and SOX3 were conducted with two independent biological replicates, confirming consistency through Spearman correlation analysis using plotCorrelation (deepTools) and pairwise R square values among the three independent biological replicates of RNA-seq. |
| Randomization | Randomization was not applicable to this study as the experimental groups (e.g., genotypes or developmental stages) were predefined by the study design and biological context. Covariates were controlled by using consistent experimental conditions, including identical protocols for sample preparation, data collection, and analysis across all groups. Biological replicates were derived independently to account for natural variability, ensuring the robustness of the findings. |
| Blinding | Blinding was not performed in this study because the experimental groups (e.g., genotypes or developmental stages) were explicitly predefined and inherently distinguishable, such as through genetic markers or developmental time points. The nature of the study required precise identification of these groups during sample processing and analysis. To minimize bias, consistent protocols were applied across all groups, and data analysis was conducted using automated and objective computational tools wherever possible. |

# Reporting for specific materials, systems and methods

We require information from authors about some types of materials, experimental systems and methods used in many studies. Here, indicate whether each material, system or method listed is relevant to your study. If you are not sure if a list item applies to your research, read the appropriate section before selecting a response.

## Materials & experimental systems

| n/a | Involved in the study |
|---|---|
| ☐ | ☒ Antibodies |
| ☒ | ☐ Eukaryotic cell lines |
| ☒ | ☐ Palaeontology and archaeology |
| ☐ | ☒ Animals and other organisms |
| ☒ | ☐ Clinical data |
| ☒ | ☐ Dual use research of concern |
| ☒ | ☐ Plants |

## Methods

| n/a | Involved in the study |
|---|---|
| ☐ | ☒ ChIP-seq |
| ☐ | ☒ Flow cytometry |
| ☒ | ☐ MRI-based neuroimaging |

## Antibodies

| | |
|---|---|
| Antibodies used | We provide a comprehensive list of all antibodies used in this study in Extended Data Table 1, including details on the manufacturer, catalog number, and specific applications. |
| Validation | All antibodies employed in this study are commercially available and were validated by the manufacturer. Validation of rabbit anti-ZBTB16, goat anti-ZBTB16 and mouse anti-SOX3 antibodies included the use of relevant mutant animals in this study. In details:<br><br>Host organism Target Species Manufacturer Catalog number RRID Validation statement<br>mouse ACTB mouse Proteintech Cat# 66009-1-lg AB_2782959 https://www.ptglab.com/products/Pan-Actin-Antibody-66009-1-Ig.htm<br>rabbit ADAMTS5 mouse Thermo Fisher Scientific Cat# PA5-27165 AB_2544641 https://www.thermofisher.com/antibody/product/ADAMTS5-Antibody-Polyclonal/PA5-27165<br>mouse CCND1 mouse Santa Cruz Biotechnology Cat# sc-8396 AB_627344 https://www.scbt.com/p/cyclin-d1-antibody-a-12?gad_source=1&gclid=Cj0KCQjwqP2pBhDMARIsAJQ0CzrS5XHH5sEjTc8G84h58NdC2Ec1dlLl89WhA0bEHpE9c9fwdiHLTboaAp5UEALw_wcB<br>rat CCND2 mouse Santa Cruz Biotechnology Cat# sc-452 AB_627350 https://www.scbt.com/p/cyclin-d2-antibody-34b1-3<br>goat GATA4 mouse Santa Cruz Biotechnology Cat# sc-1237 AB_2108747 https://www.scbt.com/p/gata-4-antibody-c-20<br>rabbit H3K27me3 mouse EMD Millipore Cat# 07-449 AB_310624 https://www.emdmillipore.com/US/en/product/Anti-trimethyl-Histone-H3-Lys27-Antibody,MM_NF-07-449?ReferrerURL=https%3A%2F%2Fwww.google.com%2F<br>rabbit H3K4me3 mouse Active Motif Cat# 39159 AB_2615077 https://www.activemotif.com/catalog/details/39159/histone-h3-trimethyl-lys4-antibody-pab<br>rabbit H3K9me3 mouse Active Motif Cat# 39161 AB_2532132 https://www.activemotif.com/catalog/details/39161/histone-h3-trimethyl-lys9-antibody-pab<br>rat KIT/CD117 mouse Miltenyi Biotec Cat# 130-091-224 AB_2753213 https://www.miltenyibiotec.com/US-en/products/cd117-microbeads-mouse.html#130-091-224<br>rat KIT/CD117-PE mouse Thermo Fisher Scientific Cat# 12-1171-81 AB_465812 https://www.thermofisher.com/antibody/product/CD117-c-Kit-Antibody-clone-2B8-Monoclonal/12-1171-81<br>goat LIN28A mouse R&D systems Cat# AF3757 AB_2234537 https://www.rndsystems.com/products/human-lin-28a-antibody_af3757<br>rabbit Phospho-CDK4 (Thr172) mouse Thermo Fisher Scientific Cat# 702556 AB_2632989 https://www.thermofisher.com/antibody/product/Phospho-CDK4-Thr172-Antibody-clone-9H2L7-Recombinant-Monoclonal/702556<br>rabbit Phospho-CDK6 mouse Thermo Fisher Scientific Cat# PA537517 AB_2554126 https://www.thermofisher.com/antibody/product/Phospho-CDK6-Tyr13-Antibody-Polyclonal/PA5-37517<br>mouse Phospho-H2AX mouse Abcam Cat# ab26350 AB_470861 https://www.abcam.com/products/primary-antibodies/gamma-h2ax-phospho-s139-antibody-9f3-ab26350.html<br>rabbit Phospho-H2AX (S136, 20E3) mouse Cell Signaling Cat# 9718 AB_2118009 https://www.cellsignal.com/products/primary- |

antibodies/phospho-histone-h2a-x-ser139-20e3-rabbit-mab/9718

mouse Pan-RNAPol2 mouse "Active Motif" Cat# 39097 AB_2732926 https://www.activemotif.com/catalog/details/39097/rna- pol-ii-antibody-mab

rabbit RNAPol2 CTD phospho Ser5 mouse "Active Motif" Cat# 39233 AB_2793198 https://www.activemotif.com/catalog/details/39233/rna- pol-ii-ctd-phospho-ser5-antibody-pab

rat  RNAPol2 CTD phospho Ser2 mouse "Active Motif" Cat# 61083 AB_2687450 https://www.activemotif.com/catalog/61083/rna- pol-ii-ctd-phospho-ser2-antibody-mab

rabbit  SALL4  mouse Abcam  Cat# ab29112 AB_777810 https://www.abcam.com/products/primary-antibodies/sall4-antibody-ab29112.html

rabbit SOX3 mouse EMD Millipore Cat# AB5772 AB_2302597 https://www.sigmaaldrich.co.th/th_en/ab5772-th

mouse  SOX3 mouse Santa Cruz Biotechnology Cat# sc-101155 AB_2195961 https://www.scbt.com/p/sox-3-antibody-16-c2

mouse SYCP3  mouse Abcam Cat# ab97672 AB_10678841 https://www.abcam.com/products/primary-antibodies/scp3-antibody-cor-10g117-ab97672.html

mouse SYCP3  mouse Santa Cruz Biotechnology Cat# sc-74569 AB_2197353 https://www.scbt.com/p/scp-3-antibody-d-1

mouse TBP mouse EMD Millipore Cat# MAB3658 AB_2200056 https://www.emdmillipore.com/US/en/product/Anti-TATA-Binding-Protein-Antibody,MM_NF-MAB3658

rat  THY1/CD90.2  mouse Miltenyi Biotec Cat# 130-049-101 AB_3073748 https://www.miltenyibiotec.com/US-en/products/cd90-2-microbeads-mouse.html

rat  THY1/CD90.2-PE-Cy7 (53-2-1) mouse Thermo Fisher Scientific Cat# 25-0902-81 AB_469641 https://www.thermofisher.com/antibody/product/CD90-2-Thy-1-2-Antibody-clone-53-2-1-Monoclonal/25-0902-81

mouse UTF1  mouse EMD Millipore Cat# MAB4337 AB_827541 https://www.emdmillipore.com/US/en/product/Anti-UTF-1-Antibody-clone-5G10.2,MM_NF-MAB4337

rabbit VERSICAN V0, V1 Neo  mouse Thermo Fisher Scientific Cat# PA1-1748A AB_2304324 https://www.thermofisher.com/antibody/product/PA1-1748A.html?ef_id=Cj0KCQjwqP2pBhDMARIsAJQ0CzqyYcJEbrSNQY5iQofd3Z0IO3xyge1G_agd3S9qnkk-xTCRdJBGndwaAhF_EALw_wcB:G:s&s_kwcid=AL!3652!3!459737518508!!!g!!!10950825775!106531320406&cid=bid_pca_aup_r01_co_cp1359_pjt0000_bid00000_0se_gaw_dy_pur_con&gad_source=1&gclid=Cj0KCQjwqP2pBhDMARIsAJQ0CzqyYcJEbrSNQY5iQofd3Z0IO3xyge1G_agd3S9qnkk-xTCRdJBGndwaAhF_EALw_wcB

goat  ZBTB16  mouse Santa Cruz Biotechnology Cat# sc-11146 AB_2218938 https://www.scbt.com/p/plzf-antibody-n-21

rabbit  ZBTB16  mouse Santa Cruz Biotechnology Cat# sc-22839 AB_2304760 https://www.scbt.com/p/plzf-antibody-h-300

goat  Rabbit IgG-HRP mouse Bio-Rad Cat# 170-6518 AB_11125338 https://www.bio-rad.com/en-us/sku/1706518-goat-anti-rabbit-igg-ap-conjugate?ID=1706518

bovine Goat IgG-HRP mouse Santa Cruz Biotechnology Cat# sc-2354  https://www.scbt.com/p/mouse-anti-goat-igg-hrp?srsltid=AfmBOoqyWfLHmZRs1kYZJXmqq1_4Dmit850qpbpwc0aDZljGZFxrPo7m

donkey Alex Fluor 488 donkey anti-mouse IgG mouse Thermo Fisher Scientific Cat# A21202 AB_141607 https://www.thermofisher.com/antibody/product/Donkey-anti-Mouse-IgG-H-L-Highly-Cross-Adsorbed-Secondary-Antibody-Polyclonal/A-21202

donkey Alex Fluor 594 donkey anti-mouse IgG mouse Thermo Fisher Scientific Cat# A21203 AB_2535789 https://www.thermofisher.com/antibody/product/Donkey-anti-Mouse-IgG-H-L-Highly-Cross-Adsorbed-Secondary-Antibody-Polyclonal/A-21203

donkey Alex Fluor 647 donkey anti-mouse IgG mouse Thermo Fisher Scientific Cat# A31571 AB_162542 https://www.thermofisher.com/antibody/product/Donkey-anti-Mouse-IgG-H-L-Highly-Cross-Adsorbed-Secondary-Antibody-Polyclonal/A31571

donkey Alex Fluor 488 donkey anti-rabbit IgG mouse Thermo Fisher Scientific Cat# A21206 AB_2535792 https://www.thermofisher.com/antibody/product/Donkey-anti-Rabbit-IgG-H-L-Highly-Cross-Adsorbed-Secondary-Antibody-Polyclonal/A-21206

donkey Alex Fluor 594donkey anti-rabbit IgG mouse Thermo Fisher Scientific Cat# A21207 AB_141637 https://www.thermofisher.com/antibody/product/Donkey-anti-Rabbit-IgG-H-L-Highly-Cross-Adsorbed-Secondary-Antibody-Polyclonal/A-21207

donkey Alex Fluor 647 donkey anti-rabbit IgG mouse Thermo Fisher Scientific Cat# A31573 AB_2536183 https://www.thermofisher.com/antibody/product/Donkey-anti-Rabbit-IgG-H-L-Highly-Cross-Adsorbed-Secondary-Antibody-Polyclonal/A-31573

donkey Alex Fluor 594 donkey anti-goat IgG mouse Thermo Fisher Scientific Cat# A11058 AB_2534105 https://www.thermofisher.com/antibody/product/Donkey-anti-Goat-IgG-H-L-Cross-Adsorbed-Secondary-Antibody-Polyclonal/A-11058

donkey Alex Fluor 488 donkey anti-rat IgG mouse Thermo Fisher Scientific Cat# A21208 AB_2535794 https://www.thermofisher.com/antibody/product/Donkey-anti-Rat-IgG-H-L-Highly-Cross-Adsorbed-Secondary-Antibody-Polyclonal/A-21208

# Animals and other research organisms

Policy information about studies involving animals; ARRIVE guidelines recommended for reporting animal research, and Sex and Gender in Research

| Laboratory animals | The study was conducted in accordance with the approved animal use protocols (no. 18-03004 and 00001726) by the Institutional Animal Care and Use Committee (IACUC) at the University of Utah and the National Institute of Health Guide for the Care and Use of Laboratory Animals. All mice were kept in a pathogen-free animal facility and provided with a standard rodent chow diet. The housing facility maintained a controlled temperature (20-25 °C), a 12-hour light/dark cycle and a relative humidity of 30-70% to support the mice's circadian rhythm. The mice used in this study were derived from the C57BL/6J (B6) background obtained from Jackson Laboratory (RRID:IMSR_JAX:000664). Zbtb16 (luxoid) mice were also procured from Jackson Laboratory (RRID:IMSR_JAX:000100). Ddx4/Vasa-Cre (B6, RRID:IMSR_JAX:018980) was purchased from Jackson Laboratory. Sox3 floxed embryos (B6) were generously provided by Dr. Jeffrey Weiss. Sox3 conditional knockout (cKO) mice (B6), specifically targeting male germ cells, were generated using Ddx4-Cre mice. Male mice were randomly selected at the age of postnatal 7-90, and all the experiments used littermate controls or mice of the same ages. |
|---|---|
| Wild animals | No wild animals were used in this study. |

| Reporting on sex | All biological replicates employed in this study for male germ cell studies were derived from male C57BL/6J mice. |
| --- | --- |
| Field-collected samples | No field-collected samples were used in the study. |
| Ethics oversight | The authors confirm that all animal experiments were conducted in accordance with the approved animal use protocols (no. 18-03004 and 00001726) by the Institutional Animal Care and Use Committee (IACUC) at the University of Utah and the National Institute of Health Guide for the Care and Use of Laboratory Animals. |

Note that full information on the approval of the study protocol must also be provided in the manuscript.

## Plants

| Seed stocks | N/A |
| --- | --- |
| Novel plant genotypes | N/A |
| Authentication | N/A |

## ChIP-seq

### Data deposition

☒ Confirm that both raw and final processed data have been deposited in a public database such as GEO.

☒ Confirm that you have deposited or provided access to graph files (e.g. BED files) for the called peaks.

| Data access links<br>*May remain private before publication.* | The datasets generated and/or analyzed during this study are publicly available on the GEO database under the following accession numbers: bulk RNA-seq and ChIP-seq data (GSE202819) and Hi-C data (GSE244681). All data sources, including the mm10 genome assembly, are appropriately cited and referenced. |
| --- | --- |
| Files in database submission | ChIPSeq_P7_Testis_ZBTB16_rabbit,<br>ChIPSeq_P7_Testis_ZBTB16_goat,<br>ChIPSeq_P7_Testis_SOX3,<br>ChIPSeq_P7_Testis_SALL4,<br>ChIPSeq_P7_Testis_ZBTB16_ChIPNexus,<br>ChIPSeq_P7_Testis_SOX3_ChIPNexus,<br>ChIPSeq_P7_Testis_SALL4_ChIPNexus,<br>ChIPSeq_P7_THY1+_uSPG_H3K4me3,<br>ChIPSeq_P7_THY1+_uSPG_H3K9me3,<br>ChIPSeq_P7_THY1+_uSPG_H3K27me3,<br>ChIPSeq_P7_Testis_WT_SALL4,<br>ChIPSeq_P7_Testis_WT_SOX3,<br>ChIPSeq_P7_Testis_WT_Pan-RNAPol2,<br>ChIPSeq_P7_Testis_WT_RNAPol2Ser5P,<br>ChIPSeq_P7_Testis_WT_RNAPol2Ser2P,<br>ChIPSeq_P7_Testis_Null_SALL4 and<br>ChIPSeq_P7_Testis_Null_SOX3 |
| Genome browser session<br>(e.g. UCSC) | https://genome.ucsc.edu/s/jaruvy/ZBTB16_SOX3_SALL4 |

### Methodology

| Replicates | The ChIP-seq and ChIP-nexsus experiments for ZBTB16, SALL4, and SOX3 were conducted with two independent biological replicates. |
| --- | --- |
| Sequencing depth | Title Total reads Unique reads Sequence Length single or paired-end<br>ChIPSeq_P7_Testis_Input_rep1 37,594,101 25,526,801 50 single-end<br>ChIPSeq_P7_Testis_ZBTB16_rabbit_rep1 36,050,671 19,284,944 50 single-end<br>ChIPSeq_P7_Testis_ZBTB16_goat_rep1 28,832,621 18,973,252 50 single-end<br>ChIPSeq_P7_Testis_SOX3_rep1 32,153,478 16,132,382 50 single-end<br>ChIPSeq_P7_Testis_SALL4_rep1 39,298,208 20,173,212 50 single-end<br>ChIPSeq_P7_Testis_ZBTB16_ChIPNexus_rep1 30,444,948 78,929 50 single-end<br>ChIPSeq_P7_Testis_SOX3_ChIPNexus_rep1 27,835,503 37,308 50 single-end<br>ChIPSeq_P7_Testis_SALL4_ChIPNexus_rep1 27,824,153 87,230 50 single-end<br>ChIPSeq_P7_Testis_Input_rep2 41,302,839 27,994,685 50 single-end |

ChIPSeq_P7_Testis_ZBTB16_rabbit_rep2 34,616,014 18,543,562 50 single-end
ChIPSeq_P7_Testis_ZBTB16_goat_rep2 34,186,714 20,233,831 50 single-end
ChIPSeq_P7_Testis_SOX3_rep2 38,718,666 21,940,574 50 single-end
ChIPSeq_P7_Testis_SALL4_rep2 36,166,684 20,280,165 50 single-end
ChIPSeq_P7_Testis_ZBTB16_ChIPNexus_rep2 27,091,147 54,720 50 single-end
ChIPSeq_P7_Testis_SOX3_ChIPNexus_rep2 27,867,097 23,303 50 single-end
ChIPSeq_P7_Testis_SALL4_ChIPNexus_rep2 27,867,628 380,585 50 single-end
ChIPSeq_P7_THY1+_uSPG_Input_rep1 36,682,065 26,816,768 50 single-end
ChIPSeq_P7_THY1+_uSPG_H3K4me3_rep1 35,641,326 19,701,827 50 single-end
ChIPSeq_P7_THY1+_uSPG_H3K9me3_rep1 36,129,820 12,844,452 50 single-end
ChIPSeq_P7_THY1+_uSPG_H3K27me3_rep1 36,620,517 26,547,448 50 single-end
ChIPSeq_P7_Testis_WT_SALL4_rep1 39,614,022 33,732,649 150  paired-end
ChIPSeq_P7_Testis_WT_SOX3_rep1 29,288,746 12,735,078 150  paired-end
ChIPSeq_P7_Testis_WT_TBP_rep1 32,049,918 25,715,535 150  paired-end
ChIPSeq_P7_Testis_WT_Pan-RNAPol2_rep1 30,628,382 27,873,006 150  paired-end
ChIPSeq_P7_Testis_WT_RNAPol2Ser5P_rep1 29,699,194 25,660,891 150  paired-end
ChIPSeq_P7_Testis_WT_RNAPol2Ser2P_rep1 37,513,828 33,793,427 150  paired-end
ChIPSeq_P7_Testis_WT_SALL4_rep2 37,565,462 32,063,498 150  paired-end
ChIPSeq_P7_Testis_WT_SOX3_rep2 39,382,316 33,015,219 150  paired-end
ChIPSeq_P7_Testis_WT_TBP_rep2 30,208,066 24,086,142 150  paired-end
ChIPSeq_P7_Testis_WT_Pan-RNAPol2_rep2 31,942,220 29,224,982 150  paired-end
ChIPSeq_P7_Testis_WT_RNAPol2Ser5P_rep2 35,961,962 32,402,681 150  paired-end
ChIPSeq_P7_Testis_WT_RNAPol2Ser2P_rep2 34,290,226 31,209,312 150  paired-end
ChIPSeq_P7_Testis_Null_SALL4_rep1 34,628,924 28,034,309 150  paired-end
ChIPSeq_P7_Testis_Null_SOX3_rep1 40,160,082 32,331,132 150  paired-end
ChIPSeq_P7_Testis_Null_SALL4_rep2 33,824,290 27,715,009 150  paired-end
ChIPSeq_P7_Testis_Null_SOX3_rep2 37,332,202 31,367,842 150  paired-end
ChIPSeq_P7_Testis_WT_Input_rep1 24,357,664 17,036,717 150  paired-end
ChIPSeq_P7_Testis_WT_Input_rep2 24,914,202 17,185,627 150  paired-end
ChIPSeq_P7_Testis_Null_Input_rep1 23,151,738 16,418,484 150  paired-end
ChIPSeq_P7_Testis_Null_Input_rep2 25,414,338 16,155,747 150  paired-end

**Antibodies**

Host organism Target Manufacturer Catalog number Application
rabbit H3K27me3  EMD Millipore  Cat# 07-449 5  µl
rabbit H3K4me3  Active Motif  Cat# 39159 3  µl
rabbit H3K9me3  Active Motif  Cat# 39161 5  µl
mouse RNAPol2 Active Motif Cat# 39097 0.2 µg (1 µl)
rabbit RNAPol2 CTD phospho Ser5 Active Motif Cat# 39233 ChIP 5 µg (5 µl)
rabbit RNA Pol2 CTD phospho Ser2  Active Motif Cat# 61083 ChIP 5 µg (5 µl)
rabbit  SALL4  Abcam  Cat# ab29112 5 µg
mouse  SOX3 Santa Cruz  Cat# sc-101155 2.5 µg (25 µl)
rabbit SOX3 EMD Millipore Cat# AB5772 ChIP 2.5 µg  (5  µl)
mouse TBP EMD Millipore Cat# MAB3658 AB_2200056 ChIP 1 µl
goat  ZBTB16  Santa Cruz  Cat# sc-11146 5 µg (25 µl)
rabbit  ZBTB16  Santa Cruz  Cat# sc-22839 5 µg (25 µl)

**Peak calling parameters**
Peak calling was performed using USeq8.9.6 packages and MACS (2.1.1) with the default parameters.

**Data quality**
Peaks with -10log10(q-Value) (QValFDR) greater than 30 (q<0.001) for USeq and 20 (q<0.01) for MACS were considered for downstream analyses.

**Software**
For ChIP-seq data analysis, the reads were aligned to the mouse reference genome (mm10) using Novoalign (http://novocraf.com) with the following parameters: -o SAM -r Random -H -a AGATCGGAAGAGCACACGTCTGAACTCCAGTCA, which includes adapter sequence removal. To ensure fair comparison across all datasets, PCR duplicates and all unmapped reads were removed using Picard MarkDuplicates (version 2.7.1; https://broadinstitute.github.io/picard/) and Samtools. Peak calling was performed using USeq8.9.6 packages and MACS (2.1.1) with the default parameters. Peaks with -10log10(q-Value) (QValFDR) greater than 30 (q<0.001) for USeq and 20 (q<0.01) for MACS were considered for downstream analyses. Replicate handing was modified based on a previous approach. Briefly, only peaks from the merged files of two/three replicates overlapped with at least 50% of peaks from the union of biological replicates. This was achieved using BEDTools intersect with parameters: -u -f 0.5. Moreover, genomic regions blacklisted in mice were removed from the peaks using BEDTools intersect with the parameter: -v. To annotate the peaks, ChIPseeker was used with the RefSeq gene list for genome version mm10. For ZBTB16, SALL4 and SOX3, promoters were defined as ±2 Kb from the transcription start site (TSS), and for histone modification, promoters were defined as ±1 Kb from the TSS. Additionally, annotated genes that were excluded from the merged replicates but overlapped with at least two biological replicates were included, and bidirectional promoter genes were manually added using BEDTools. For genome browser visualization, ChIP signals were first normalized to their corresponding inputs using USeq after peak calling. The snapshots for genome browser were captured using the IGV browser. To facilitate the analysis, we employed the deepTools suite, using the bamCompare module to ChIP signals against their input. The computeMatrix and plotHeatmap modules were used for ChIP-seq heatmaps and clustering, employing a reference-point approach. The spearman correlation between ChIP-seq samples was computed using the plotCorrelation module.

# Flow Cytometry

## Plots

Confirm that:

☒ The axis labels state the marker and fluorochrome used (e.g. CD4-FITC).

☒ The axis scales are clearly visible. Include numbers along axes only for bottom left plot of group (a 'group' is an analysis of identical markers).

☒ All plots are contour plots with outliers or pseudocolor plots.

☒ A numerical value for number of cells or percentage (with statistics) is provided.

## Methodology

| | |
|---|---|
| Sample preparation | To perform flow cytometry analysis, single cells were isolated from testes at P7 and resuspended in MACS separation buffer (Miltenyi Biotec, 130-091-221). The collected cells were subsequently stained with anti-THY1/CD90.2-PE-Cy7 (Thermo Fisher Scientific, 25-0902-81) and anti-KIT/CD117-PE (Thermo Fisher Scientific, 12-1171-81) antibodies, according to the manufacturer's instructions, for a duration of 15 minutes at 4 °C. Following this, the cells were fixed in a solution of 4% formaldehyde/PBS for 20 minutes on ice. After appropriate washes, the cells were subjected to staining with DAPI solution (0.1% Triton X-100 and 10 µg/ml of DAPI in PBS) and incubated overnight at 4 °C. |
| Instrument | FACS Canto Scan (BD Biosciences) |
| Software | DATA collection: FACSDiva version 8.01 (BD Biosciences)<br>Data analysis: FlowJo (v9.9, BD Biosciences) |
| Cell population abundance | Populations were sorted as depicted in Extended Data Fig.8c. The FACS Canto Scan is not suitable for analyzing cell population abundance from post-sort fractions. Sort purity was not validated through reanalysis, but antibody specificity was confirmed using unstained cells and single-stained cells. Sort efficiency was typically 67% (actual number of cells vs. sorter counts). The markers used for sorting were differentially expressed by RNA, as illustrated in Extended Data Fig. 1c, indicating that the sorted populations were pure. |
| Gating strategy | Initially, potential cells were identified using forward scatter (FSC) area and back scatter (BSC) area. Subsequently, potential doublets were eliminated based on BSC and FSC signal width. Finally, undifferentiated spermatogonia and differentiating spermatogonia were separated using THY1 and KIT antibodies, followed by the analysis of DNA contents for cell cycle assessment. |

☒ Tick this box to confirm that a figure exemplifying the gating strategy is provided in the Supplementary Information.

