## [Peer Review File · Nature Structural & Molecular Biology]

ZBTB16/PLZF regulates juvenile spermatogonial stem cell development via an extensive transcription factor poisoning network

Corresponding Author: Professor Bradley Cairns

Version 0:

Decision Letter:

Nature Structural & Molecular Biology NSMB-PI48299

10th Oct 2023

Dear Dr. Cairns,

Thank you for the presubmission inquiry regarding your manuscript "Roles for ZBTB16 in mouse spermatogonial stem cell identity, cell cycle control and an extensive transcription factor-chromatin poisoning network". Based on the summary of your study, we are interested in reading the entire manuscript and further considering your report for possible publication in Nature Structural & Molecular Biology. Please note that it is difficult to accurately assess a manuscript in the absence of the entirety of the data. Therefore, while we very much look forward to receiving your paper and reading what sounds like an interesting story, we cannot guarantee that we would necessarily send the work out for review until we have read, assessed and discussed the entire manuscript.

In order to submit your complete manuscript, please use the link below:

Link Redacted

Sincerely,

Dimitris Typas
Associate Editor
Nature Structural & Molecular Biology
ORCID: 0000-0002-8737-1319

Version 1:

Decision Letter:

13th Dec 2023

Dear Dr. Cairns,

Thank you again for submitting your manuscript "ZBTB16/PLZF regulates self-renewal and differentiation of spermatogonial stem cells through an extensive transcription factor-chromatin poisoning network". I apologise for the delay in responding, which resulted from the difficulty in timely obtaining suitable referee reports. Nevertheless, we now have comments (below) from the 3 reviewers who evaluated your paper. In light of these reports, we remain interested in your study and would like to see your response to the comments of the referees, in the form of a revised manuscript.

You will see that though the experts appreciate the experimental approaches used and note that certain findings are

interesting, they raise serious concerns on the causality and extent of the phenotypes observed, as well as the applicability of certain cellular systems used, which would need to be unequivocally addressed in a revised manuscript. More specifically, reviewer #1 assesses that the data do not convincingly support a mechanistic model and do not clearly establish direct/causal effects, therefore requesting extensive additional data that support the reached conclusions. The editorial team agrees that these experiments would be needed to strengthen the study for publication in NSMB. Reviewer #3 notes that in the absence of using pure germ cell populations, or clearly excluding contaminating cellular populations, there are potential caveats in the interpretation of the data - please address this concern either experimentally and/or with discussion of caveats in the experimental approach. Finally, multiple reviewers assess that the story would benefit from additional clarifications, controls, reorganisation, new analyses and further expanding in certain areas. We editorially note that addressing all technical and functional requests of the referees, as well as clearly demonstrating or rationalising the validity of the cell systems used, will be paramount for the success of the story and a prerequisite for returning the revised manuscript to the experts for their assessment.

Please be sure to address/respond to all concerns of the referees in full in a point-by-point response and highlight all changes in the revised manuscript text file.

We appreciate the requested revisions are extensive. We thus expect to see your revised manuscript within 6 months. If you cannot send it within this time, please let us know. We will be happy to consider your revision as long as nothing similar has been accepted for publication at NSMB or published elsewhere. Should your manuscript be substantially delayed without notifying us in advance and your article is eventually published, the received date would be that of the revised, not the original, version.

Reporting Summary:

We require deposition of coordinates (and, in the case of crystal structures, structure factors) into the Protein Data Bank with the designation of immediate release upon publication (HPUB). Electron microscopy-derived density maps and coordinate data must be deposited in EMDB and released upon publication. Deposition and immediate release of NMR chemical shift assignments are highly encouraged. Deposition of deep sequencing and microarray data is mandatory, and the datasets must be released prior to or upon publication. To avoid delays in publication, dataset accession numbers must be supplied with the final accepted manuscript and appropriate release dates must be indicated at the galley proof stage. Please find the complete NRG policies on data availability at <http://www.nature.com/authors/policies/availability.html>.

Nature Structural & Molecular Biology is committed to improving transparency in authorship. As part of our efforts in this direction, we are now requesting that all authors identified as 'corresponding author' on published papers create and link their Open Researcher and Contributor Identifier (ORCID) with their account on the Manuscript Tracking System (MTS), prior to acceptance. This applies to primary research papers only. ORCID helps the scientific community achieve unambiguous attribution of all scholarly contributions. You can create and link your ORCID from the home page of the MTS by clicking on

'Modify my Springer Nature account'. For more information please visit www.springernature.com/orcid.

Link Redacted

Sincerely,

Dimitris Typas
Associate Editor
Nature Structural & Molecular Biology
ORCID: 0000-0002-8737-1319

Reviewers' Comments:

Reviewer #1:

Remarks to the Author:

In this manuscript, Yi et al evaluate the genomic distribution and regulatory functions of three transcription factors involved in mouse spermatogonial development, ZBTB16, SALL4, and SOX3. Focusing on undifferentiated spermatogonia (uSPG) isolated from postnatal day 7 (P7) testis, the paper combines new and published ChIP-seq, RNA-seq, Hi-C, cell cycle, and immunofluorescence data along with mouse mutant and in vitro models to assess binding and transcriptional effects of each of these transcription factors at different points during spermatogenesis. In addition to uSPG, they also examine data from differentiating spermatogonia (KIT+, dSPG), pachytene spermatocytes and round spermatids in vivo, as well as germline stem cells (GSCs), multipotent adult spermatogonial-derived stem cells (MASCs) and ESCs in vitro. They compare wild type data to knockout and depletion models for each transcription factor including Zbtb16 KO mice and Sall4 and Sox3 knockdown GSCs. They perform ChIP-seq for ZBTB16, SOX3, and SALL4, RNA-seq in knockout (KO) cells for each factor, cell cycle analysis by flow cytometry, immunofluorescence in gonad sections, and Hi-C for analysis of long-range interactions. Their major conclusion is that a transcription factor network including ZBTB16, SALL4, and SOX3 pre-establishes an activation-ready chromatin state in spermatogonia at many genes that will be activated later in spermatogenesis, especially those important for meiosis.

The central finding of the study is that ZBTB16, SOX3, and SALL4 co-occupy thousands of genomic in uSPGs, and that the occupancy of all three transcription factors is highly correlated. This finding is well supported by the data shown. In addition, the compilation of datasets is a useful resource for mapping the functional interactions between these three factors. However, the study does not effectively integrate the impressive collection of datasets to provide a coherent picture for how this co-occupancy contributes to gene regulation. There is little relationship between occupancy based on ChIP-seq and gene expression based on the knockout/depletion models, nor is there a strong relationship between the genes found to be misregulated following depletion of each factor independently. Overall, the study relies heavily on correlations among ChIP-seq datasets and does not provide convincing support for a mechanistic model whereby the proposed transcription factor network regulates developmentally important gene activity during spermatogenesis. Several claims, including the overarching model introduced in the Discussion that are not well supported by the data. As a result, the study is of limited impact.

Major concerns

1) The model described in the discussion is an over-claim based on the essentially descriptive and correlative data. The major finding of this study is that sites of ZBTB16, SALL4, and SOX3 enrichment are highly correlated in undifferentiated spermatogonia, and that this enrichment marks a set of loci that are in an active chromatin state in pre-meiotic and meiotic stages, and appear to cluster together in three-dimensional space. This is a striking and intriguing observation, but it does not provide evidence that ZBTB16, SALL4, and SOX3 coordinate expression of these genes or functional interactions between them. In fact, very few genes are co-regulated by any combination of these factors (Fig 5e), and the correlations and anti-correlations among these rare co-regulated genes are weak (Fig 5f-h).

2) Similarly, the discussion suggests that stage-specific activators are superimposed on the open/poised network marked by ZBTB16/SOX3/SALL4 and can stimulate transcription at appropriate developmental intervals (lines 431-432). This is an intriguing model but not supported by the data in this study. The best support is provided by analysis of RHOX10 binding, shown in Extended Data Fig. 11, which indicates that RHOX10 binds only a subset of the gene set held open by ZBTB16/SOX3/SALL4. However, this analysis still does not provide evidence that this enrichment has a functional effect on gene expression as suggested by the model.

3) The comparison between uSPG and MASCs/mGSCs indicates that the in vitro models are a poor approximation for in

vivo uSPG regulatory state (Extended Data Figure 5). This is an interesting and important conclusion by itself, but does not really help with interpreting the in vivo function of ZBTB16. In addition, this argues that the comparison of data from SALL4-depleted mGSCs with data from uSPGs in Extended Data Figure 9 is not informative; it is not clear what conclusions can be reliably drawn from the mGSC data. Likewise, the statement that “Our data does not support a prior model that SALL4 sequesters and antagonizes ZBTB16 to repress Kit... as Kit expression remained unchanged in Sall4-deficient mGSCs” (lines 258-259) is based on data from mGSCs (Extended Data Fig. 9i-l) and cannot really be used to draw conclusions about interactions in vivo.

4) Line 195-197: “These results imply that ZBTB16 ensures a slow cell cycle in self-renewing early uSPG (Thy1+/Kit-) but helps accelerate the cell cycle in late uSPG (Thy1+/Kit+).” The results could also be consistent a model where ZBTB16 does not directly regulate the cell cycle, but rather uncouples expression of differentiation markers from the cell cycle. In the subsequent section on cell cycle control and cyclin/Cdk expression (lines 209-228), it is also not clear how the correlations between CCND1/CCND2 or CDK6/CDK4 and ZBTB16 support a role for ZBTB16 in cell cycle control.

5) In Extended Data Fig. 10a and 10c, a very high overlap is shown between SOX3/SALL4 peaks and activating marks H3K4me3 and H3K27ac, as discussed in the text. However, there is also substantial overlap with the repressive mark H3K27me3, which is not discussed.

6) The paper overall is somewhat confusingly written and poorly organized. It is often hard to keep track of which cell types and models are being compared in a given analysis. The figures rely heavily on Venn diagrams that are difficult to read and interpret: it is frequently hard to tell which labels correspond to which regions of the diagram and sometimes unclear which gene set is indicated by which label. Showing the relationships between datasets in a clearer way would substantially improve the presentation. In addition, there are cases where data is shown in the figures but is not discussed or explained in the text. For example, Fig 2e-i includes a set of genes annotated as bivalent, but the relationship between ZBTB16 and bivalency is not referenced at all in the main text.

Minor concerns

7) Methods for isolation of pachytene spermatocytes and round spermatids are not described in either the main text or methods. This information should be provided.

8) Extended Data Fig. 1c: Based on the RNA-seq browser tracks, it looks like Kit+ cells are also positive for Thy1 expression. Can contamination with the Thy1+ cell population be excluded?

9) Extended Data Fig. 1d: The text referring to this figure says that “genes occupied by ZBTB16 were... nearly identical”, but the figure appears to show the correlations between ChIP signal, which this is not the same thing as the set of occupied genes.

10) A minor inconsistency is that in Fig. 1d-e, the enriched functional terms cited in text don't quite correspond to those shown in the figure. For example, no terms for cell migration are shown in Fig. 1d.

11) Fig. 3h: The abbreviations used in labeling the x-axis should be defined in the legend.

12) Line 249: Cyp26b1 and Lin28a are identified as two of the 12 genes that are “co-occupied [by ZBTB16 and SOX3] in both uSPG and NPCs”, but the cited figure 4g does not show data for occupancy of ZBTB16 in NPCs. It is not clear what the relevance of the NPC data is for ZBTB16 function.

13) Line 211: typo: Id4-GFP(bight) -> Id4-GFP(bright)

14) Line 211-214: this sentence is a fragment.

Reviewer #2:

Remarks to the Author:

In the present study Yi and co-authors describe the role of ZBTB16 in regulating self-renewal and differentiation of spermatogonial stem cells. Authors combine a plethora of high-throughput genomic analysis including RNAseq, ChIPseq and HiC to sustain their results. They first define ZBTB16 binding sites and transcription activity in vivo to then correlate with histone modification profiles. Then authors explore the contributions of the network members ZBTB16, SALL4, and SOX3 in transcriptional regulation of uSPG.

The manuscript is well written, and results are presented in a clear way. I find the authors provide compelling evidence of the multifaceted role of ZBTB16 in key uSPG processes in mice meiosis. I have, however, some general comments that need further clarification.

Lines 89-94: It is relevant that nearly 40% of ZBTB16 binding sites reside in distal intergenic regions. Moreover, only 4.7% of ZBTB16-bound genes were affected in Zbtb16-deficient THY1+ uSPG. Can the authors expand in these results?

Figure 2g show differences in H3K27ac enrichment at the TSS of downregulated or upregulated genes in Zbtb16 null. Since

differences are also seen in regions that have neither bivalent nor H3K4me3 signal, how relevant are such differences? Have the authors considered to conduct Cut&Run histone validations in Zbtb16 null mice?

It seems that different strategies were used to isolate uSPG and meiotic cells depending on the downstream analysis conducted (ChIPseq & RNAseq vs HiC). This should be clarified in the text and extended in the M&M description. While the ChIPseq methodology is well described, the HiC procedures are not. For example, it is not clear how many cells per type were used for the HiC analysis, neither how enrichment analyses were conducted. Also, it will be good to describe the HiC protocol in detail (i.e., cells analyzed, number of replicates, etc...). Figure 1 should include the method used for the isolation of uSPG and meiotic cells.

Related to the HiC results, can the authors expand on the CTCF results? Not sure 26% is a notable fraction.

Minor comments:

Lines 304-306: The statement 'Our prior work revealed an unexpected chromatin logic during spermatogenesis, where 'Atypical promoters' marked by DNAm and often H3K27me3 represent the majority of active promoters during spermiogenesis' needs a reference.

Reviewer #3:

Remarks to the Author:

In this study, Yi and colleagues aimed to uncover a molecular network that drives the biology of undifferentiated spermatogonia in mammalian testes. Because the undifferentiated component of the spermatogonial population provides foundation for continual spermatogenesis, clearly defining the mechanisms that underpin fate decisions is important. While several sophisticated analyses were conducted to generate potentially novel information about mammalian spermatogonia, the experimental approaches used have several major limitations that have not been properly considered or addressed. Thus, I find that much of the data cannot be validly interpreted in regard to a molecular network regulating spermatogonial functions.

Introduction

Lines 38-41: These statements are not fully supported by primary research and should be reconsidered. First, the referenced studies only examined mouse spermatogonia which may not fully reflect the biology of spermatogonia across Mammalia. Second, the undifferentiated spermatogonial population in mouse testes encompasses all A_{single}, A_{paired}, and A_{aligned} cohorts and in total this population is more abundant than 0.03% of the total testis cell population (see PMID: 21521798). Third, the notion of undifferentiated spermatogonia being highly mobile is based on live imaging of select seminiferous tubules in mice that are not under normal physiological conditions including being under anesthesia and the testes exposed outside of the body cavity for long periods of time. The impacts of this non normal condition on behaviors for the cells has not been explored. Thus, I encourage the authors to revise the statement to consider limitations of the experimental approaches that were used to generate data suggesting that undifferentiated spermatogonia are highly mobile.

Lines 41-44: I suggest that the authors not use the terminology of male germline stem cells (mGSCs). These in vitro derivatives are primary cultures of spermatogonia in which the actual functional stem cell content is low (1-20% as reported in the peer-reviewed literature). Referring to the cultures as mGSCs implies that all the cells harbor stem cell capacity and is misleading. It is more accurate to refer to the cells as primary cultures of undifferentiated spermatogonia.

Lines 44-47: These statements should be reconsidered for accuracy. First, previous studies have shown that ZBTB16 expression in the mouse male germline extends to at least a portion of A₁ differentiating spermatogonia (see PMID: 30026551). Second, expression of Zbtb16 is not germline specific (see any number of testis scRNA-seq datasets). Third, the phenotype of Zbtb16 KO mice does not fully support the statement that gradual undifferentiated spermatogonial exhaustion occurs. In the models that have been described in the peer-reviewed scientific literature, only a minor percentage of seminiferous tubules have loss of the germline due to exhaustion of the spermatogonial population. In fact, most of the seminiferous tubule mass has ongoing spermatogenesis well into adult life. Thus, ZBTB16 is not required or essential for either maintenance of the undifferentiated spermatogonial population or spermatogonial differentiation.

Results

A major limitation of the study is that the authors designed the experiments and drew interpretations from the findings based on an understanding that ZBTB16 is essential/required for spermatogonial biology. Unfortunately, I do not agree that findings presented in the peer-reviewed scientific literature supports this understanding. The phenotype of ZBTB16 deficient mice is not consistent with a notion of the molecule being essential or required for normal spermatogonial biology. In adult life, most of the seminiferous tubule mass has complete spermatogenesis in both the luxoid mutant and Zbtb16 genetic knockout mouse models. Thus, although some regions lose germline due to defects in spermatogonial maintenance, the majority of the tissue mass does not. In the regions of normal spermatogenesis, the spermatogonial population is functionally intact in the absence of ZBTB16. The authors should take this into account for interpreting the results and be careful in making claims of the essentiality of a molecular network that involves ZBTB16. The persistence of a spermatogonial population and complete spermatogenic lineage in large portions of seminiferous tubules of ZBTB16 deficient mice indicates that spermatogonial stem cell self-renewal and differentiation occurs normally in the absence of the proposed network.

A major concern of the study that has not been properly considered is that the RNA-seq, ChIP-seq, flow cytometric, and Hi-C data were generated from isolated cell populations that are not pure germ cells. Neither the THY1+ nor KIT+ fractions of mouse testes are pure germ cells. While the molecules are expressed by subsets of spermatogonia, both are also expressed by populations of testicular somatic cells. THY1 is also expressed by Leydig cells, macrophages, and potentially other somatic cells such as peritubular myoid cells and monocytes (see PMIDs: 19270176, 34001436, 28780938, and multiple previous scRNA-seq atlases of mouse or human testicular tissue). KIT is also expressed by Leydig cells and potentially other somatic cells (see PMIDs: 1712701, 11076685, 12773427, and multiple previous scRNA-seq atlases of mouse or human testicular tissue including the senior author's). Indeed, a major portion of the THY1+ testis population in mouse testes is macrophages (PMID: 19270176). Considering that ZBTB16 is also expressed by macrophages (see multiple previous scRNA-seq atlases of mouse or human testicular tissue including the senior author's), it is highly possible that some, potentially a lot, of the RNA-seq, ChIP-seq, and Hi-C data represents differential gene expression due to ZBTB16 deficiency and ZBTB16 binding sites in macrophages and other somatic cells rather than undifferentiated spermatogonia. Similarly, ZBTB16 is expressed by Leydig cells, thus a portion of the gene expression profiling and DNA binding analysis likely does not represent what is occurring in differentiating spermatogonia. Without being able to parse the results by germ cell vs somatic cell, I believe that most of the data presented cannot be validly interpreted.

Another aspect of the study that makes valid interpretations challenging is that cells from mice at the development age point of P7 were used. At this age, both the somatic and germline populations are in a state of development that is very different compared to the adult stage of life. The complete spermatogenic lineage is not established until P30-40 in mice and many of the somatic support cell populations are not mature until P10-12. Thus, the data generated in this study are relevant to cell lineages that are in an immature state and valid interpretations about networks underpinning their biology in a steady-state condition should not be made. The authors should either investigate whether some of the key findings made with cells at P7 hold true in the same cells in adult mice or revise their interpretations and conclusions to reflect that the findings are for cells in an immature developmental state. This includes revising the title and abstract to reflect the fact that spermatogonia in steady-state conditions of adult spermatogenesis were not investigated.

Version 2:

Decision Letter:

10th Oct 2024

Dear Professor Cairns,

Thank you again for submitting your manuscript "ZBTB16/PLZF regulates self-renewal and differentiation of juvenile spermatogonial stem cells through an extensive transcription factor-chromatin poisoning network". I apologise for the delay in responding, which resulted from the difficulty in timely obtaining suitable referee reports and discussing them amongst the entire editorial team. Nevertheless, we now have comments (below) from the 3 reviewers who evaluated your paper. In light of those reports, we remain interested in your study and would like to see your response to the comments of the referees, in the form of a revised manuscript.

You will see that though the experts assess that the manuscript is significantly improved and now constitutes a likely candidate for publication, they still raise some textual concerns that must be addressed in a revised version before the manuscript can be accepted in principle. More specifically, Reviewer #1 requests that the lack of definitive connecting data to support the functional conclusions needs to be adequately acknowledged and discussed, Reviewer #2 notes the need for the addition of certain technical details to help the reader fully understand the methods used, and Reviewer #3 notes the discrepancies in the use of the term mGSC and in the definitiveness of previous studies with respect to functional roles of ZBTB16. We urge you to please textually address all remaining points raised by the experts, by explaining limitations and caveating where necessary. If the latter is not possible, we request that you at the very least explicitly and collegiately acknowledge the difference in views with respect to certain findings and experimental systems.

We expect to see your revised manuscript within 6 weeks. If you cannot send it within this time, please contact us to discuss an extension; we would still consider your revision, provided that no similar work has been accepted for publication at NSMB or published elsewhere.

Reporting Summary:

Data availability: this journal strongly supports public availability of data. All data used in accepted papers should be available via a public data repository, or alternatively, as Supplementary Information. If data can only be shared on request, please explain why in your Data Availability Statement, and also in the correspondence with your editor. Please note that for some data types, deposition in a public repository is mandatory - more information on our data deposition policies and available repositories can be found below:

<https://www.nature.com/nature-research/editorial-policies/reporting-standards#availability-of-data>

Link Redacted

Sincerely,

Dimitris Typas
Senior Editor
Nature Structural & Molecular Biology
ORCID: 0000-0002-8737-1319

Reviewers' Comments:

Reviewer #1 (Remarks to the Author):

The authors have thoughtfully addressed most of my concerns and have substantially improved both the clarity and content of the paper. Most notably, they have incorporated new analysis of the PRO-seq data from Kaye et al. 2024 as well as new

ChIP-seq data for TBP and PolII that shows paused PolII at most Typical genes and supports the model of a pre-established open chromatin network for pre-meiotic and meiotic genes. This is a significant addition that improves the study, and provides further support for functional segregation of Typical and Atypical promoters related to a pre-activation state in spermatogonia. In addition, they have appropriately adjusted claims that were based on in vitro MASC and mGSC data and rewritten the results to better clarify where each dataset comes from. This section is now much clearer and more scientifically accurate. They have also toned down the language for the claim about SSAs promoting transcription on the poised network, allowing for a better supported model. All my minor concerns have been adequately addressed. Overall, the paper now provides a generally well-supported conceptual advance. I have a few remaining concerns:

1) The major remaining issue is that all the ZBTB16 KO data is still largely negative: the phenotype is mild, there is little effect on SALL4/SOX3 binding, and the effect on gene expression is much less than the widespread ZBTB16 binding would imply and not closely linked to binding sites at the functional level. The authors explain this as being based on redundancy between ZBTB16, SALL4, and SOX3. This is plausible, but is not directly tested (e.g. by double or triple KO), and the fact remains that there is just not much in this study to support a functional role of ZBTB16. In addition, the data remains fundamentally correlative. However, I agree that the very high correlation in binding profiles and correlation with PolII states is striking, and the study overall offers a conceptual advance. I suggest that the issue of experimental support for a functional role for ZBTB16 should be acknowledged and addressed in the Discussion section.

2) The authors indicated that the relationship to H3K27me3 is not discussed because it is complicated, and was therefore excluded due to space considerations. However, since the data is included in the paper it should at least be briefly discussed, or else the data should be left out. Low overlap with H3K27me3 is mentioned briefly in line 117, but the relationship should also be mentioned in the results section related to Fig 5d.

3) In Figure 4h, why are the SOX3 ChIP-seq tracks from published uSPG dataset (GSE146706) and this paper's dataset so different? There appears to be no signal from the published dataset.

4) In the model diagram (Figure 7b), the right-hand side with spermatozoa should be removed, since there is no data nor analysis of spermatozoa in this study.

5) Typo in line 474: promoters promoters.

Reviewer #2 (Remarks to the Author):

Authors have addressed most of my initial concerns in the revised version. However, the information provided in lines 734-740 does not fully suffice to assess the validity of the data. The rebuttal letter and the methodological section mention an accompanying paper for the description of the HiC methods, but each paper should be able to stand on its own. Therefore, I recommend the authors to extend their description of the methods section.

Reviewer #3 (Remarks to the Author):

In this revised version the authors have addressed most but not all the concerns raised in the first round of review. I would like to again suggest that the authors reconsider accuracy of some of the statements that have been made.

Please reconsider dropping mGSC from the manuscript and refer to the cells as primary cultures of spermatogonia; this does not require coming up with a new acronym, just describe them for what they are. It is appreciated that the authors did not start the misleading trend of labeling the cultured cells as mGSCs but sticking with it perpetuates the problem.

I still question statements about the role of ZBTB16 in ensuring long-term maintenance of undifferentiated spermatogonia and that its absence results in gradual depletion of the population in a significant subset of seminiferous tubules. Experimental evidence in previous studies referenced by the authors that a significant portion of seminiferous tubules are depleted of the undifferentiated spermatogonial population in Zbtb16 knockout mice is not very convincing. I reiterate that those studies showed that only a minor portion of seminiferous tubule cross-sections seem to have a depletion of the spermatogonial population. Neither Costoya et al., 2004 nor Buaas et al., 2004 quantified seminiferous tubules lacking spermatogonia, rather images of testis cross-sections were provided, but Sharma et al., 2019 (PMID: 31149899) did and reported ~10-15% of cross-sections lacked spermatogonia at 6 months of age. The significance of a degree of depletion in advanced age that is <25% of the seminiferous tubules is debatable. Note that disrupted spermatogenesis may or may not be due to problems with the undifferentiated spermatogonial population and is therefore not a definitive indicator of impaired long-term maintenance. In addition, the transplantation analysis performed in previous studies with Zbtb16 knockout cells are fraught with experimental deficiencies including use of cross-sectional histology to assess recipient seminiferous tubules which cannot accurately account for potential donor-derived spermatogonial colonization throughout the tissue mass. For these reasons, I again encourage the authors to temper statements about the functional importance of ZBTB16 in maintenance of the undifferentiated spermatogonial population.

Version 3:

Decision Letter:

Our ref: NSMB-A48299C

19th Nov 2024

Dear Professor Cairns,

Thank you for submitting your revised manuscript "ZBTB16/PLZF regulates self-renewal and differentiation of juvenile spermatogonial stem cells through an extensive transcription factor-chromatin poisoning network" (NSMB-A48299C). I apologise for the slight delay in returning our decision which came to be due to editorial absences precluding us from discussing the changes in the manuscript. Nevertheless, we have now editorially assessed the responses to the final points raised by the reviewers and we find that no major issues remain. We are therefore happy to accept the manuscript in principle in Nature Structural & Molecular Biology, pending minor revisions to tone down certain statements in accordance to the referees' final requests and to comply with our editorial and formatting guidelines.

We are now performing detailed checks on your paper and will send you a checklist detailing our editorial and formatting requirements, as well as a line-edited text with the aforementioned toning down incorporated, in about 2-3 weeks. Please do not upload the final materials and make any revisions until you receive this additional information from us.

To facilitate our work at this stage, it is important that we have a copy of the main text as a word file. If you could please send along a word version of this file as soon as possible, we would greatly appreciate it; please make sure to copy the NSMB account (cc'ed above).

Sincerely,

Dimitris Typas
Senior Editor
Nature Structural & Molecular Biology
ORCID: 0000-0002-8737-1319

Version 4:

Decision Letter:

28th Jan 2025

Dear Professor Cairns,

We are now happy to accept your revised paper "ZBTB16/PLZF regulates juvenile spermatogonial stem cell development via an extensive transcription factor poisoning network" for publication as an Article in Nature Structural & Molecular Biology.

Your paper will be published online soon after we receive proof corrections and will appear in print in the next available issue. You can find out your date of online publication by contacting the production team shortly after sending your proof corrections.

Authors may need to take specific actions to achieve <https://www.springernature.com/gp/open-research/funding/policy-compliance-faqs> compliance with funder and institutional open access mandates. If your research is supported by a funder that requires immediate open access (e.g. according to <https://www.springernature.com/gp/open-research/plan-s-compliance> Plan S principles) then you should select the gold OA route, and we will direct you to the compliant route where possible. For authors selecting the subscription publication route, the journal's standard licensing terms will need to be accepted, including <https://www.springernature.com/gp/open-research/policies/journal-policies> self-archiving policies. Those licensing terms will supersede any other terms that the author or any third party may assert apply to any version of the manuscript.

Sincerely,

Dimitris Typas
Senior Editor
Nature Structural & Molecular Biology
ORCID: 0000-0002-8737-1319

Author's Response to Reviewers' comments: Yi et al., and Cairns

We would like to express our gratitude to the reviewers for their detailed and constructive comments on our manuscript. Your feedback has been very helpful in guiding the extensive revisions we have made, which has improved the clarity and depth of our study. Below, we provide a point-by-point response to each of the reviewers' comments and highlight the three major areas where we have made substantial revisions:

1. Clarifications and Improvements: We have implemented numerous clarifications throughout the manuscript to improve accuracy and clarity. These changes include a more precise description of the *Zbtb16* null phenotype, clearer explanations of the primary and secondary functions of ZBTB16, SALL4, and SOX3, and confirmation that the cell populations used in our genomic analyses are indeed enriched for spermatogonia (SPG).

2. Exploring Cooperativity and Redundancy: To address the reviewers' questions regarding the potential cooperativity or redundancy among ZBTB16, SALL4 and SOX3, we conducted additional ChIP-seq experiments focusing on SALL4 and SOX3 binding in *Zbtb16* null spermatogonia. Our results show that the binding sites for SALL4 and SOX3 are largely preserved in the absence of ZBTB16, indicating that these transcription factors do not depend on ZBTB16 and supporting the concept of redundancy within this transcription factor network.

3. Integration of RNA Polymerase II (RNAPol2) and Transcriptional Pausing Data: During our revisions, we incorporated recent findings by Kaye et al., 2024 (PMID: 38287033) concerning RNAPol2 occupancy at meiotic genes in uSPG, which suggested that paused RNAPol2 primes meiotic genes for rapid activation. To explore this further, we performed ChIP-seq experiments targeting TBP (TATA-binding protein) and the two primary phosphorylation states of RNAPol2, and compared to published PRO-seq data from uSPG, spermatocytes, and round spermatids. Our results demonstrate that both TBP and RNAPol2 consistently occupy Typical gene promoters across all stages of spermatogenesis, while they are absent from Atypical promoters. This finding underscores a pivotal role of RNAPol2 pausing in regulating transcription at meiotic Typical genes in spermatogonia, thus reinforcing our proposed model. Further analysis reveals that paused RNAPol2 is predominantly associated with meiotic genes in uSPG, which closely align with Typical Promoters, and shows no significant overlap with Atypical promoters. Notably, we find that RNAPol2 continues to associate with SPG/mitotic and meiotic genes even after these genes are silenced during spermiogenesis, suggesting a mechanism for imposing transcriptional pausing at these genes. In contrast, Atypical genes recruit RNAPol2 concurrently with active transcription during spermiogenesis. We view this as a significant addition to our manuscript and the overall model.

Taken together, these revisions (and the others described below) have substantially strengthened our manuscript.

Authors' response to Reviewers' Comments:

Reviewer #1: *In this manuscript, Yi et al evaluate the genomic distribution and regulatory functions of three transcription factors involved in mouse spermatogonial development, ZBTB16, SALL4, and SOX3. Focusing on undifferentiated spermatogonia (uSPG) isolated from postnatal day 7 (P7) testis, the paper combines new and published ChIP-seq, RNA-seq, Hi-C, cell cycle, and immunofluorescence data along with mouse mutant and in vitro models to assess binding and transcriptional effects of each of these transcription factors at different points during spermatogenesis. In addition to uSPG, they also examine data from differentiating spermatogonia (KIT⁺, dSPG), pachytene spermatocytes and round spermatids in vivo, as well as germline stem*

cells (GSCs), multipotent adult spermatogonial-derived stem cells (MASCs) and ESCs in vitro. They compare wild type data to knockout and depletion models for each transcription factor including *Zbtb16* KO mice and *Sall4* and *Sox3* knockdown GSCs. They perform ChIP-seq for ZBTB16, SOX3, and SALL4, RNA-seq in knockout (KO) cells for each factor, cell cycle analysis by flow cytometry, immunofluorescence in gonad sections, and Hi-C for analysis of long-range interactions. Their major conclusion is that a transcription factor network including ZBTB16, SALL4, and SOX3 pre-establishes an activation-ready chromatin state in spermatogonia at many genes that will be activated later in spermatogenesis, especially those important for meiosis.

The central finding of the study is that ZBTB16, SOX3, and SALL4 co-occupy thousands of genomic in uSPG, and that the occupancy of all three transcription factors is highly correlated. This finding is well supported by the data shown. In addition, the compilation of datasets is a useful resource for mapping the functional interactions between these three factors. However, the study does not effectively integrate the impressive collection of datasets to provide a coherent picture for how this co-occupancy contributes to gene regulation. There is little relationship between occupancy based on ChIP-seq and gene expression based on the knockout/depletion models, nor is there a strong relationship between the genes found to be misregulated following depletion of each factor independently. Overall, the study relies heavily on correlations among ChIP-seq datasets and does not provide convincing support for a mechanistic model whereby the proposed transcription factor network regulates developmentally important gene activity during spermatogenesis. Several claims, including the overarching model introduced in the Discussion that are not well supported by the data. As a result, the study is of limited impact.

First, we thank the reviewer for their detailed and constructive comments, which helped structure our revision. This review includes positive comments regarding the quality and scope of the genomics work and its usefulness to the field. Indeed, a major finding of the work involves our discovery of the co-occupancy of transcription factors and the 3D spatial clustering of their binding sites in undifferentiated spermatogonia at >12K genes active during mitotic and subsequent pre-meiotic and meiotic stages – which we interpret as a TF-chromatin network in spermatogonia that poises thousands of genes for spermatogonial differentiation and meiosis, but not spermiogenesis. The reviewer has the expectation that the knockout of *Zbtb16* should have a strong effect on ZBTB16-occupied genes but does not. We also had that expectation initially – until we showed that there are other TFs, including SALL4 and SOX3, that bind to the same >12K sites. Our model then considered TF redundancy for ZBTB16, SALL4 and SOX3, which may be redundant in helping to establish the initial poised chromatin network – and may be coupled to the subsequent involvement of stage-specific TFs (like RHOX10 and others) to conduct activation on poised chromatin. During revision, we provide additional support for this model by showing that SALL4 and SOX3 partially-largely retain their binding sites in the *Zbtb16* null. ZBTB16, SALL4 and SOX3 also help activate a small number of genes within the network, in addition to a possible redundant role in network poising.

Second, we are excited to report a major addition during revision which involves the dovetailing the work of Kaye et al., 2024, (PMID: 38287033) showing that our poised TF and 3D chromatin network at Typical Promoters also contains bound, paused RNAPol2. We also show TBP occupancy at the entire Typical promoter network. This contrasts with Atypical promoters, which are not poised by chromatin, lack paused RNAPol2, and simply recruit RNAPol2 during activation. We consider this is a major addition to the manuscript and model (Figures 6-7).

Major concerns

1) *The model described in the discussion is an over-claim based on the essentially descriptive and correlative data. The major finding of this study is that sites of ZBTB16, SALL4, and SOX3 enrichment are highly correlated in undifferentiated spermatogonia, and that this*

enrichment marks a set of loci that are in an active chromatin state in pre-meiotic and meiotic stages, and appear to cluster together in three-dimensional space. This is a striking and intriguing observation, but it does not provide evidence that ZBTB16, SALL4, and SOX3 coordinate expression of these genes or functional interactions between them. In fact, very few genes are co-regulated by any combination of these factors (Fig 5e), and the correlations and anti-correlations among these rare co-regulated genes are weak (Fig 5f-h).

We regret that we did not communicate our model clearly, which involves two separate roles for these three transcription factors: a redundant role in uSPG as part of a large poised chromatin network, and a specific role in affecting the expression of a moderate number of targets within this network. As previewed above, we do not claim that the main/sole function of ZBTB16, SALL4 and SOX3 is to activate specific individual genes within this network. Instead, we believe the main function of these three TFs is (in a redundant manner) to help establish in uSPG the poised chromatin-TF network at >12K genes for SPG mitotic growth, differentiation, and meiosis (termed Typical Promoters). This function contrasts with that of the majority of spermiogenesis genes, which are not poised, lack these factors, and involve transcription from DNA-methylated promoters (Atypical Promoters). We concede that the occupancy data involves correlation, but we hope that the reviewer will see how striking the correlations are between many factors and modifications – especially with the new data on the near-perfect overlap of paused/poised RNAPol2 with the Typical network and not the Atypical promoters.

Next, the reviewer correctly notes that omission of single TFs, such as ZBTB16, only affected a limited number of genes in the network. We agree, and emphasize that this is consistent with our model, involving redundancy among ZBTB16, SALL4 and SOX3 for their main function of ‘poising’ permissive chromatin within the network. This is reinforced by our experiments done during revision which show that loss of ZBTB16 has only a modest effect on the ability of SALL4 and SOX3 to bind to the >12K sites. The secondary role of these three TFs is to additionally help regulate a modest subset of the genes within the network at specific stages. Indeed, prior examinations of transcriptional networks have revealed that certain transcription factors occupy 10-100X more binding sites or genes than the number of genes they directly regulate in particular cellular contexts (Biggin 2011, PMID: 22014521; Revilla-i-Domingo et al. 2012, PMID: 22669466).

- 2) *Similarly, the discussion suggests that stage-specific activators are superimposed on the open/poised network marked by ZBTB16/SOX3/SALL4 and can stimulate transcription at appropriate developmental intervals (lines 431-432). This is an intriguing model but not supported by the data in this study. The best support is provided by analysis of RHOX10 binding, shown in Extended Data Fig. 11, which indicates that RHOX10 binds only a subset of the gene set held open by ZBTB16/SOX3/SALL4. However, this analysis still does not provide evidence that this enrichment has a functional effect on gene expression as suggested by the model. “Here, we propose that additional stage-specific activators (SSA) stimulate transcription on an already poised/active promoter chromatin landscape at Typical promoters during meiosis, while a separate set of SSAs then activate DNA-methylated Atypical promoters during spermiogenesis (Fig.7b).”*

We agree with the reviewer that our model of SSAs acting upon the open/poised network established by ZBTB16/SOX3/SALL4 is intriguing. As the reviewer points out, our RHOX10 data provides initial support for this model. Specifically, we combined ChIP occupancy datasets with gene expression results to identify directly-occupied target genes where gene expression is reduced when RHOX10 is absent, which provides evidence for a direct functional impact.

We acknowledge that our evidence for this concept is limited to the RHOX10 example. Therefore, we will state in the manuscript (and in the figure legend for the model) that the role of

additional SSAs in promoting transcription on the open chromatin landscape a hypothesis that has initial support but warrants future investigation. The notion of SSAs acting at particular stages during spermatogenesis has been proposed by others. Here, our study contributes by identifying the specific chromatin landscapes/promoters (Typical/poised or Atypical/DNA methylated) upon which these SSAs may function (Line 506).

During revision, we also integrated our findings with data from Kaye et al., 2024, PMID: 38287033) to speculate that these SSAs may play a role in the release of paused RNA Polymerase II (RNAPol2), given our new insights of the co-presence of paused Pol II at the Typical poised gene network, but not at Atypical genes in SPG (Line 396).

- 3) *The comparison between uSPG and MASCs/mGSCs indicates that the in vitro models are a poor approximation for in vivo uSPG regulatory state (Extended Data Figure 5). This is an interesting and important conclusion by itself but does not really help with interpreting the in vivo function of ZBTB16. In addition, this argues that the comparison of data from SALL4-depleted mGSCs with data from uSPG in Extended Data Figure 9 is not informative; it is not clear what conclusions can be reliably drawn from the mGSC data. Likewise, the statement that “Our data does not support a prior model that SALL4 sequesters and antagonizes ZBTB16 to repress Kit...as Kit expression remained unchanged in Sall4-deficient mGSCs” (lines 258-259) is based on data from mGSCs (Extended Data Fig. 9i-l) and cannot really be used to draw conclusions about interactions in vivo.*

We appreciate that the reviewer recognizes the merit in our evidence that prior MASC and mGSC *in vitro* systems are poor models for SPG. We also agree that our use of the data from mGSCs should be limited in scope given that fact. In the revised manuscript, we have shorted our presentation of the mGSC data, noted where the primary support for the conclusion derives from our *in vivo* experiment, rewritten the sentence mentioned above, and (where appropriate) moved data to the supplemental information. Thus, we better highlight the more relevant *in vivo* data.

- 4) *Line 195-197: “These results imply that ZBTB16 ensures a slow cell cycle in self-renewing early uSPG (THY1+/KIT-) but helps accelerate the cell cycle in late uSPG (THY1+/KIT+).” The results could also be consistent a model where ZBTB16 does not directly regulate the cell cycle, but rather uncouples expression of differentiation markers from the cell cycle. In the subsequent section on cell cycle control and cyclin/Cdk expression (lines 209-228), it is also not clear how the correlations between CCND1/CCND2 or CDK6/CDK4 and ZBTB16 support a role for ZBTB16 in cell cycle control.*

We agree that it remains possible that ZBTB16 impacts specific cell cycle markers without directly regulating the cell cycle. However, we note that the markers we chose, including cyclins and CDKs, are proteins essential in driving the cell cycle, and are not merely bystanders – and our approach is standard in the field. Our analysis also demonstrates that *Kit* remains unaffected by the loss of ZBTB16.

In terms of cell cycle regulation, we emphasize that ZBTB16 specifically binds and activates *Ccnd1*, rather than other cyclins or CDKs. Our findings indicate that ZBTB16 plays a direct role in promoting the undifferentiated-to-differentiating spermatogonia transition by activating *Ccnd1*, which facilitates quiescence exit in late-stage undifferentiated spermatogonia (uSPG). Furthermore, our findings indicate that early uSPG utilize CCND2 for quiescence exit, while late uSPG employ CCND1 for the same process. CDK4, on the other hand, is involved in quiescence exit across both early and late uSPG stages. This nuanced understanding of the distinct roles of cyclins and CDKs at various stages of uSPG development provide a more detailed understanding of cell cycle regulation in spermatogonia (Line 437-451).

- 5) *In Extended Data Fig. 10a and 10c, a very high overlap is shown between SOX3/SALL4 peaks and activating marks H3K4me3 and H3K27ac, as discussed in the text. However, there is also substantial overlap with the repressive mark H3K27me3, which is not discussed.*

Yes, H3K27me3 is present at a portion of the Typical network in uSPG. Genetic experiments by others support a functional role for PRC2, including the regulation of genes in SPG that promote meiosis. However, we elected for a light treatment as the H3K27me3 mark itself does not follow a clear overall regulatory logic throughout spermatogenesis (and we are managing space limitations). For example, although others have evidence that H3K27me3 may repress genes in SPG, we observe high H3K27me3 at virtually all meiotic genes during meiosis (during their peak expression), and after meiosis almost every locus in the genome that lacks DNAm acquires H3K27me3 (Hammoud et al., 2014, PMID: 24835570).

- 6) *The paper overall is somewhat confusingly written and poorly organized. It is often hard to keep track of which cell types and models are being compared in a given analysis. The figures rely heavily on Venn diagrams that are difficult to read and interpret: it is frequently hard to tell which labels correspond to which regions of the diagram and sometimes unclear which gene set is indicated by which label. Showing the relationships between datasets in a clearer way would substantially improve the presentation. In addition, there are cases where data is shown in the figures but is not discussed or explained in the text. For example, Fig 2e-i includes a set of genes annotated as bivalent, but the relationship between ZBTB16 and bivalency is not referenced at all in the main text.*

We acknowledge the complexity of this study, which involves two connected stories – one focusing on ZBTB16 itself, and the other on the poising network, which involves ZBTB16. In response, we have improved the clarity and organization of the manuscript during revision. Regarding the relationship between ZBTB16 and bivalency in Fig. 2e-i, we have now incorporated a relevant discussion in the Results section (around line 118).

Additionally, we have better labeled several Venn diagrams and re-examined how we relate them to datasets. Furthermore, we have now ensured that all data panels in figure are discussed. This has improved the clarity and organization of the paper.

Minor concerns

- 7) *Methods for isolation of pachytene spermatocytes and round spermatids are not described in either the main text or methods. This information should be provided.*

Thank you for bringing this to our attention. The isolation of pachytene spermatocytes and round spermatids, along with the analysis of Hi-C data was conducted as described in the co-submitted work by Kitamura et al. In response to your comment, we have outlined the cell isolation methods in our manuscript and provided a reference to the co-submitted work by Kitamura et al. for details. (see Line 734-745)

- 8) *Extended Data Fig. 1c: Based on the RNA-seq browser tracks, it looks like KIT⁺ cells are also positive for Thy1 expression. Can contamination with the THY1⁺ cell population be excluded?*

We and others consider *Kit* expression in THY1⁺ uSPG to be a biological phenomenon, as *Kit* is indeed expressed in THY1⁺ uSPG. Specifically, *Kit* expression is observed in *Id4*-GFP^{Bright} uSPG (24.17 FPKM), as demonstrated by bulk RNA-seq data from Helsel et al. 2017 (PMID: 28087628). This finding supports the notion that *Kit* expression in THY1⁺ uSPG is not a result of contamination by KIT⁺ dSPG. Additionally, this observation lends additional support to the conclusion that

ZBTB16 does not inhibit *Kit* expression in uSPG. Furthermore, *Kit* expression is expected during the transition from THY1⁺ uSPG to KIT⁺ cells dSPG, which involves a double-positive intermediate stage. Finally, it is known from the Geyer lab (Niederberger et al., 2015, PMID: 25446031) that the c-Kit mRNA is regulated by translation in a retinoic acid dependent manner, which helps restrict KIT⁺ protein to the differentiation phase.

Ensembl Gene ID	Gene symbol	Id4 -GFP ^{Bright} (FPKM)	Id4 -GFP ^{Dim} (FPKM)
ENSMUSG00000021379	Id4	49.45	6.04
ENSMUSG00000066687	Zbtb16	79.06	42.30
ENSMUSG00000044312	Neurog3	8.80	15.60
ENSMUSG00000032011	Thy1	3.29	0.58
ENSMUSG00000005672	Kit	24.17	108.23
ENSMUSG00000045179	Sox3	53.24	92.62
ENSMUSG00000050966	Lin28a	89.13	88.56
ENSMUSG00000027547	Sall4	95.65	57.07

Reviewer Note: Table 1. Gene expression in *Id4*-GFP^{Bright} and *Id4*-GFP^{Dim} is based on RNA-seq data after flow cytometry isolation using *Id4*-GFP mice from Helsel et al. 2017 (PMID: 28087628)

- 9) *Extended Data Fig. 1d: The text referring to this figure says that “genes occupied by ZBTB16 were... nearly identical”, but the figure appears to show the correlations between ChIP signal, which this is not the same thing as the set of occupied genes.*

Thanks for this comment. We have revised the sentence to read: 'Line 91: First, the peaks (ChIP signal) occupied by ZBTB16 in the datasets derived from the rabbit and the goat antibodies were numerous and nearly identical (r values ~0.93, Extended Data Fig.1d).'

- 10) *A minor inconsistency is that in Fig. 1d-e, the enriched functional terms cited in text don't quite correspond to those shown in the figure. For example, no terms for cell migration are shown in Fig. 1d.*

Thanks for pointing this out. We have addressed this by updating the text to include 'regulation of locomotion' to the enriched terms related to cell migration to read: 'Line 98: Analysis of affected genes by gene ontology (GO) using Panther²⁰ identified enriched GO terms related to receptors and transcription factors for development and cell migration (regulation of locomotion; for genes downregulated in the null; Fig.1d) and terms for meiosis and carboxylic acid biosynthetic process (for genes upregulated in the null; Fig.1e).'

- 11) *Fig. 3h: The abbreviations used in labeling the x-axis should be defined in the legend.*

Thanks for the comment. We have updated the legend to include the abbreviations, which now read as follows: 'Line 1036: h. The quantification in panel g reveals an increase in spermatogonia (SPG) and a decrease in spermatocytes in tubules from *Zbtb16* null mice at P14. A total of 100 circular tubules were counted for each genotype (n=3). The values represent mean ± SD (* p<0.05), ** p<0.01, *** p<0.001. SPG: spermatogonia, PS: SYCP3⁺/pH2AX^{XY} body pachytene spermatocytes, and SPC: spermatocytes.'

- 12) *Line 249: Cyp26b1 and Lin28a are identified as two of the 12 genes that are “co-occupied [by ZBTB16 and SOX3] in both uSPG and NPCs”, but the cited figure 4g does not show*

data for occupancy of ZBTB16 in NPCs. It is not clear what the relevance of the NPC data is for ZBTB16 function.

To clarify, we compared SOX3-bound genes in uSPG with those in NPCs, but not with ZBTB16-bound genes in uSPG. This comparison was made because SOX3 is expressed in NPCs, and the objective was to determine the shared functions of SOX3 in these two cell types. The updated sentence now reads: 'Line 266: To identify shared functions of SOX3 in uSPG and NPCs, we conducted an intersection analysis of ChIP-seq data in uSPG with data from NPCs, revealing 12 co-occupied genes (Fig.4g).'

13) Line 211: typo: *Id4-GFP(bight)* -> *Id4-GFP(bright)*

Thanks - corrected.

14) Line 211-214: *this sentence is a fragment.*

Thanks - corrected.

Reviewer #2: *In the present study Yi and co-authors describe the role of ZBTB16 in regulating self-renewal and differentiation of spermatogonial stem cells. Authors combine a plethora of high-throughput genomic analysis including RNAseq, ChIPseq and HiC to sustain their results. They first define ZBTB16 binding sites and transcription activity in vivo to then correlate with histone modification profiles. Then authors explore the contributions of the network members ZBTB16, SALL4, and SOX3 in transcriptional regulation of uSPG.*

The manuscript is well written, and results are presented in a clear way. I find the authors provide compelling evidence of the multifaceted role of ZBTB16 in key uSPG processes in mice meiosis. I have, however, some general comments that need further clarification.

We thank the reviewer for their helpful and constructive comments. We appreciate the overall comments regarding the quality of our approaches and the clarity of the results.

1) *Lines 89-94: It is relevant that nearly 40% of ZBTB16 binding sites reside in distal intergenic regions.*

Thanks for your interest in the intergenic binding of ZBTB16. This is a very interesting issue, and due to the word limitation in the initial submission, we initially omitted the data and discussion. However, we agree that it is important to address this point. We have now included this information in Extended Data Fig.4 and briefly discussed the results in the revised manuscript, specifically on line 125. Briefly, others have identified a large set of enhancers (super enhancers) that are active during spermatogenesis. Here, we show that the vast majority of these super enhancers co-bind ZBTB16, SOX3 and SALL4, are included in the 3D interaction network, have the chromatin marks H3K4me3 and H3K27ac – and thus are indistinguishable from promoters in the network.

2) *Moreover, only 4.7% of ZBTB16-bound genes were affected in Zbtb16-deficient THY1⁺ uSPG. Can the authors expand in these results?*

Thank you for raising this question. Indeed, omission of a single factor, such as ZBTB16, had a very limited effect, which aligns with our model and interpretation. Our genomics data suggests that the main function of ZBTB16, SALL4 and SOX3 is to poise permissive chromatin at promoters

and enhancers within the network in uSPG, rather than directly regulating a large number of genes. These factors have a secondary function to also help regulate a small subset of the genes within the network, but that is not their main function. We also note that recent evidence indicates that transcription factors often directly regulate a relatively small percentage, typically around 1-10%, of the genes they bind (Biggin 2011, PMID: 22014521; Revilla-i-Domingo et al. 2012, PMID: 22669466), highlighting the redundancy and cooperative nature of transcription factors in gene regulation.

- 3) *Figure 2g show differences in H3K27ac enrichment at the TSS of downregulated or upregulated genes in Zbtb16 null. Since differences are also seen in regions that have neither bivalent nor H3K4me3 signal, how relevant are such differences? Have the authors considered to conduct Cut&Run histone validations in Zbtb16 null mice?*

We appreciate this comment. Here, we emphasize that the vast majority of ZBTB16-occupied genes exhibit high H3K4me3 and moderate H3K27ac. Although we observed slight differences in H3K27ac levels at promoters of *Zbtb16*-affected genes, the source of variation remains unclear. As the reviewer correctly points out, we cannot be certain that these slight variations are significant. Therefore, we have decided to omit this figure panel from our manuscript.

- 4) *It seems that different strategies were used to isolate uSPG and meiotic cells depending on the downstream analysis conducted (ChIPseq & RNAseq vs HiC). This should be clarified in the text and extended in the M&M description. While the ChIPseq methodology is well described, the HiC procedures are not. For example, it is not clear how many cells per type were used for the HiC analysis, neither how enrichment analyses were conducted. Also, it will be good to describe the HiC protocol in detail (i.e., cells analyzed, number of replicates, etc...). Figure 1 should include the method used for the isolation of uSPG and meiotic cells.*

Thank you for your comment. We initially did not include the detailed Hi-C methods mentioned because they were conducted as described by Kitamura et al. in their accompanying paper. However, we agree that each paper should stand on its own from a methods standpoint, your point is well taken. As a result, we have now outlined the cell isolation methods in our manuscript and provided a reference to the co-submitted work by Kitamura et al. for the details (see Line 734-745). Additionally, we have expanded the Methods & Materials section to clarify the strategies used for isolating uSPG and meiotic cells, as well as the procedures for the Hi-C analysis, including the number of cells analyzed, the number of replicates, and the enrichment analyses conducted. We have also updated Figure 1 to include the method used for the isolation of uSPG and meiotic cells.

- 5) *Related to the HiC results, can the authors expand on the CTCF results? Not sure 26% is a notable fraction.*

Thank you for the reviewer pointing out. We misstated this in the original version, and have recalculated CTCF-bound gene promoters. The following sentence is now present in the main text. 'Line 416: Furthermore, in uSPG, we find that 56% of CTCF-bound promoters are Typical promoters, whereas only 2.9% of CTCF-bound promoters are Atypical, mirroring the pattern observed in PSs (Fig.6d).'

Minor comments:

- 6) *Lines 304-306: The statement 'Our prior work revealed an unexpected chromatin logic during spermatogenesis, where 'Atypical promoters' marked by DNAm and often*

H3K27me3 represent the majority of active promoters during spermiogenesis' needs a reference.

Thanks - corrected.

Reviewer #3: *In this study, Yi and colleagues aimed to uncover a molecular network that drives the biology of undifferentiated spermatogonia in mammalian testes. Because the undifferentiated component of the spermatogonial population provides foundation for continual spermatogenesis, clearly defining the mechanisms that underpin fate decisions is important. While several sophisticated analyses were conducted to generate potentially novel information about mammalian spermatogonia, the experimental approaches used have several major limitations that have not been properly considered or addressed. Thus, I find that much of the data cannot be validly interpreted in regard to a molecular network regulating spermatogonial functions.*

We thank the reviewer for their efforts and comments, which have improved the manuscript. We believe that a lack of clarity (and occasionally accuracy) on our part has led to misinterpretation of certain approaches and results in the paper, and have taken several steps to address in our response, below.

Introduction

- 1) *Lines 38-41: These statements are not fully supported by primary research and should be reconsidered. First, the referenced studies only examined mouse spermatogonia which may not fully reflect the biology of spermatogonia across Mammalia.*

Thank you for correctly raising this concern. As our studies and conclusions are restricted to mouse, we have revised the text to more accurately reflect the scope of the referenced studies. The statement now acknowledges that the referenced studies specifically examined mouse spermatogonia, and we have removed the term 'mammalian' to ensure precision in our description (Line 36-39).

- 2) *Second, the undifferentiated spermatogonial population in mouse testes encompasses all A_{single} , A_{paired} , and A_{aligned} cohorts and in total this population is more abundant than 0.03% of the total testis cell population (see PMID: 21521798).*

Upon closer examination, we agree. We did not quantify this ourselves, instead we relied on estimations from others, where there has been some variation and 0.03% was on the low end of those estimates. As our work does not rely on the accuracy of that number – we simply omitted that number from the manuscript.

- 3) *Third, the notion of undifferentiated spermatogonia being highly mobile is based on live imaging of select seminiferous tubules in mice that are not under normal physiological conditions including being under anesthesia and the testes exposed outside of the body cavity for long periods of time. The impacts of this non normal condition on behaviors for the cells has not been explored. Thus, I encourage the authors to revise the statement to consider limitations of the experimental approaches that were used to generate data suggesting that undifferentiated spermatogonial are highly mobile.*

Thank you for this important context. We have revised the statement to ensure proper context/limitations. Additionally, we have added another reference (Niu et al., 2016, PMID:

26904129) that utilized an *in vitro* transwell migration assay with cultured THY1⁺ undifferentiated spermatogonia, providing a complementary perspective on their migratory behavior (Line 38).

- 4) *Lines 41-44: I suggest that the authors not use the terminology of male germline stem cells (mGSCs). These in vitro derivatives are primary cultures of spermatogonia in which the actual functional stem cell content is low (1-20% as reported in the peer-reviewed literature). Referring to the cultures as mGSCs implies that all the cells harbor stem cell capacity and is misleading. It is more accurate to refer to the cells as primary cultures of undifferentiated spermatogonia.*

We agree that this terminology has liabilities and will emphasize that mGSCs is not our terminology – it is the term used by the authors that conducted those studies and is used by other groups as well (PMID:17446391, 18442644, 12700182, 15601913, 16306420, 18094355, 16107472, 23290695 and 23779100). As we do not wish to introduce another acronym, we have chosen to stay with mGSCs but have included in the revision (when first described) the reviewer's stated description that these are primary cultures of uSPG that have adapted to culturing.

- 5) *Lines 44-47: These statements should be reconsidered for accuracy. First, previous studies have shown that ZBTB16 expression in the mouse male germline extends to at least a portion of A1 differentiating spermatogonia (see PMID: 30026551).*

Thanks for the comment. We have addressed by updating the legend with definitions of the abbreviations to read as follows: 'Line 44: ZBTB16 (PLZF), which is specifically expressed in undifferentiated A-single (A_s), A-paired (A_{pr}), and A-aligned (A_{al}) uSPG⁷ and A1 differentiating spermatogonia (dSPG)^{8,9}.'

- 6) *Second, expression of Zbtb16 is not germline specific (see any number of testis scRNA-seq datasets).*

Here we note that our analysis of scRNA-seq data from the mouse testis from others (Jung et al., 2019, PMID: 31237565) did not detect *Zbtb16* transcripts in somatic testis cells. Also, *Zbtb16* transcript was not found in scRNA-seq data from macrophages isolated from peritoneal cells (Lants et al, 2020, PMID: 32868786). Furthermore, *in situ* hybridization results (Costoya et al., 2004, PMID: 15156143) demonstrated that *Zbtb16* is specifically expressed in undifferentiated spermatogonia (uSPG). Upon reviewing DeFalco et al., 2015 (PMID: 26257171), we also confirmed that ZBTB16 protein is not detected in F4/80⁺ macrophages or other interstitial cells in the adult mouse testis. Our data further show that the ZBTB16 protein is specifically detected in uSPG, but not in somatic cells (Fig.4a). Therefore, we are confident that *Zbtb16* is specifically expressed in germ cells in the mouse testis.

Reviewer Note: Figure 1. scRNA-seq data from the adult mouse testis (Jung et al., 2019, PMID: 31237565) revealed that *Zbtb16* is expressed in uSPG but not in macrophages (top). Additionally, while *Thy1* expression is relevant, it is not detected in the testis due to the low expression levels of *Thy1*, which fall below the detection threshold of scRNA-seq. in mouse testis (bottom). Red arrows indicate uSPG. Blue arrows indicate Leydig cells.

Reviewer Note: Figure 2. scRNA-seq data from the adult mouse macrophages (Lants et al, 2020 PMID: 32868786) shows that *Zbtb16* expression is not detected in any type of macrophages.

- 7) *Third, the phenotype of Zbtb16 KO mice does not fully support the statement that gradual undifferentiated spermatogonial exhaustion occurs. In the models that have been described in the peer-reviewed scientific literature, only a minor percentage of seminiferous tubules have loss of the germline due to exhaustion of the spermatogonial population. In fact, most of the seminiferous tubule mass has ongoing spermatogenesis well into adult life. Thus, ZBTB16 is not required or essential for either maintenance of the undifferentiated spermatogonial population or spermatogonial differentiation.*

We agree and have revised our description of the *Plzf/Zbtb16* null phenotype. Our original interpretation was influenced by Costoya et al., 1994, whose *Nature Genetics* article is titled “Essential role of *Plzf* in maintenance of spermatogonial stem cells”, and Buass et al., 2004, whose *Nature Genetics* article is titled “*Plzf* is required in adult male germ cells for stem cell self-renewal.” However, a closer examination supports a significant but non-essential role for ZBTB16 in SPG function. First, the studies report a markedly reduced sperm count in *Zbtb16* null. Second, as the reviewer points out, only a minor fraction of seminiferous tubules completely lost germ cells. However, many others exhibited abnormal spermatogenesis, and this was a condition that worsened with age. Third, *Zbtb16*-deficient uSPG failed to successfully transplant into recipient mice, a key assay in the field. Fourth, ZBTB16 is involved in regulating the slow cell cycle in EOMES⁺ uSPG. Fifth, CCND1⁺ SPG are decreased in *Zbtb16* null animals, which is more pronounced during aging. These findings collectively provide strong evidence for the importance of ZBTB16 in the long-term maintenance of uSPG. To ensure proper framing we now state: 'Line 45: ZBTB16 is not essential for spermatogenesis, but rather helps ensure long-term uSPG maintenance, as its absence results in the gradual depletion of uSPG in a significant subset of seminiferous tubules'. Furthermore, we present an unreported *Zbtb16* null phenotype, demonstrating that ZBTB16 promotes the transition from undifferentiated to differentiating spermatogonia by activating *Ccnd1* (Figure 3, Line 188-223).

- 8) *A major limitation of the study is that the authors designed the experiments and drew interpretations from the findings based on an understanding that ZBTB16 is essential/required for spermatogonial biology. Unfortunately, I do not agree that findings presented in the peer-reviewed scientific literature supports this understanding.*

As described above, we fully agree that ZBTB16 is not essential. Our study design and interpretations were not predicated on the assumption that *Zbtb16* is essential for SPG biology.

Instead, we intentionally studied multiple other transcription factors during the course of our work which might function in conjunction or redundantly with ZBTB16. This multi-factor approach helped us better compare ZBTB16's role with that of other non-essential factors in genome occupancy, chromatin dynamics and transcriptional impact in SPG. Importantly, our interpretation is that ZBTB16 is largely redundant with SOX3 and SALL4 within the network, which is fully consistent with the observed non-essential phenotype.

- 9) *In adult life, most of the seminiferous tubule mass has complete spermatogenesis in both the luxoid mutant and Zbtb16 genetic knockout mouse models. Thus, although some regions lose germline due to defects in spermatogonial maintenance, the majority of the tissue mass does not. In the regions of normal spermatogenesis, the spermatogonial population is functionally intact in the absence of ZBTB16. The authors should take this into account for interpreting the results and be careful in making claims of the essentiality of a molecular network that involves ZBTB16. The persistence of a spermatogonial population and complete spermatogenic lineage in large portions of seminiferous tubules of ZBTB16 deficient mice indicates that spermatogonial stem cell self-renewal and differentiation occurs normally in the absence of the proposed network*

We agree and have addressed these points in our two responses above and note that the knockout animals display a more pronounced SPG loss phenotype as the animal age, which may require. As emphasized above, we describe ZBTB16 as non-essential and the network as not depending on ZBTB16, as suggested by retention of SALL4 and SOX3 in the *Zbtb16/luxoid* null.

- 10) *A major concern of the study that has not been properly considered is that the RNA-seq, ChIP-seq, flow cytometric, and Hi-C data were generated from isolated cell populations that are not pure germ cells. Neither the THY1+ nor KIT+ fractions of mouse testes are pure germ cells. While the molecules are expressed by subsets of spermatogonia, both are also expressed by populations of testicular somatic cells. THY1 is also expressed by Leydig cells, macrophages, and potentially other somatic cells such as peritubular myoid cells and monocytes (see PMIDs: 19270176, 34001436, 28780938, and multiple previous scRNA-seq atlases of mouse or human testicular tissue). Indeed, a major portion of the THY1+ testis population in mouse testes is macrophages (PMID: 19270176).*

We considered issues of contamination very carefully during design, and we are confident that our THY1⁺ populations are SPG. First, regarding whether THY1⁺ testis cells are primarily macrophages – the study that the reviewer points to (PMID: 19270176) is a strong study by Oatley et al., 2009, which shows that the F4/80⁺ cell population (a well-known marker for macrophages) within the juvenile mouse testes completely lack THY1 expression. Oatley et al. stated: 'Expression of *Csf1r* by macrophages has been well established and previous studies have indicated that *Csf1r* expression in the adult mouse testis is localized to macrophages (Cohen et al., 1996). Thus, we used flow cytometric analysis (FCA) to examine whether enriched *Csf1r* expression by the isolated Thy1⁺ cell fraction from mouse testes could be due to co-enrichment of macrophages. Examination of the F4/80⁺ cell population, a macrophage marker, in 8-dpp mouse pup testes revealed complete absence of Thy1 expression, indicating that testicular macrophages are not Thy1⁺ (see Fig. S2 in the supplementary material). Thus, we were confident that enriched *Csf1r* expression was restricted to the Thy1⁺ germ cell fraction.' We consider the study to have provided clear evidence that testicular macrophages are not THY1⁺.

In addition, both Hammoud et al., 2015 (PMID: 26545815) and Maezawa et al., 2017 (PMID: 29126117), validated the high enrichment of germ cells isolated in the same manner that we isolated, and showed that the large majority of THY1⁺ cells were uSPG. This is reinforced in our RNA-seq experiments on THY1⁺ enriched cells, which are dominated by known markers of

SPGs and not by markers of macrophages. Finally, we note that macrophages are rarer than SPG; they comprise only 0.4% of the testis population, compared to 12% for spermatogonia. These findings collectively support THY1 expression by a subset of SPG, and not by testicular macrophages, supporting the notion that our genomics experiments were conducted on an enriched SPG population.

11) Considering that ZBTB16 is also expressed by macrophages (see multiple previous scRNA-seq atlases of mouse or human testicular tissue including the senior author's), it is highly possible that some, potentially a lot, of the RNA-seq, ChIP-seq, and Hi-C data represents differential gene expression due to ZBTB16 deficiency and ZBTB16 binding sites in macrophages and other somatic cells rather than undifferentiated spermatogonia. Similarly, ZBTB16 is expressed by Leydig cells, thus a portion of the gene expression profiling and DNA binding analysis likely does not represent what is occurring in differentiating spermatogonia. Without being able to parse the results by germ cell vs somatic cell, I believe that most of the data presented cannot be validly interpreted.

Here, we consider the absence of THY1 protein in testis macrophages (detailed above) by itself sufficient to rule out macrophage contamination. However, additional evidence supports our conclusion that our genomics analysis of ZBTB16 reflects its occupancy in uSPG. First, our own scRNA-seq analyses (PMID: 32868786) confirm that macrophages do not express *Zbtb16*, and our review of other scRNA-seq datasets reveals either very low or no *Zbtb16* RNA expression in macrophages. Furthermore, if ZBTB16 were present in macrophages, it would be illogical for it to selectively occupy genes associated with SPG and male meiosis, as our data shows, rather than genes linked to macrophage function. We further note our previous demonstration that THY1⁺ selection effectively captures the *Id4*-GFP⁺ contingent of spermatogonia from the adult mouse testis (Lord et al., 2018, PMID: 29398482 and 35433696). Thus, even though different approaches were used for spermatogonial enrichment in the aforementioned studies, the populations being selected for are largely analogous.

Regarding Leydig cells, our analysis indicates that THY1 protein expression is not detectable in these cells (Hammoud et al., 2015, PMID: 26545815). Furthermore, our staining and other prior staining only shows THY1 positivity for cells within the tubules, and adjacent the lamina, which is the location of SPG. Collectively, these findings strongly support the validity of our approach of using THY1 to enrich for SPG for RNA-seq, ChIP-seq, flow cytometry and Hi-C analyses.

12) KIT is also expressed by Leydig cells and potentially other somatic cells (see PMIDs: 1712701, 11076685, 12773427, and multiple previous scRNA-seq atlases of mouse or human testicular tissue including the senior author's).

Indeed, *Kit* is expressed in Leydig cells and other somatic cells. Here, it is important to note that interstitial cells, including Leydig cells, constitute only ~2% of adult testis cells, with Leydig cells themselves making up approximately 1%, according to scRNA-seq analysis (Green et al., 2018, PMID: 30146481). While KIT is not an exclusive marker for differentiating spermatogonia, it remains a practical and widely used marker for enriching these cells despite its expression in some somatic cells. Furthermore, Maezawa et al., 2017 (PMID: 29126117) validated the enrichment/purity of KIT⁺ cells, demonstrating that ~95% of KIT⁺ cells were indeed germ cells.

Currently, no alternative cell surface marker consistently enriches differentiating spermatogonia as effectively as KIT, so it is the field standard. Given these limitations, researchers must be careful when interpreting data involving KIT expression, ensuring that genes identified in KIT⁺ cell populations are not incorrectly ascribed to Leydig cells or other somatic cells.

This is particularly relevant when using scRNA-seq data to inform the specificity of gene expression in testicular cell populations.

Cell Type	#Cells	#Markers	%Cells	
Innate Lymphoid	64	581	0.18	
Macrophage	139	596	0.40	
Endothelial	179	496	0.52	
Myoid	49	281	0.14	
Leydig	314	576	0.91	
Sertoli	2131	369	6.15	0.23
Unknown	2205	645	6.37	
Spermatogonia	4239	260	12.24	9.79
Spermatocyte	8792	332	25.39	10.13
Round Spermatid	9923	351	28.65	19.48
Elongating	6598	421	19.05	40.40
Total	34633	4908	100.00	

Reviewer Note: Table 2. The proportion of cell types in the adult testis, based on scRNA-seq data from Green et al., 2018 (PMID: 30146481).

13) *Another aspect of the study that makes valid interpretations challenging is that cells from mice at the development age point of P7 were used. At this age, both the somatic and germline populations are in a state of development that is very different compared to the adult stage of life. The complete spermatogenic lineage is not established until P30-40 in mice and many of the somatic support cell populations are not mature until P10-12. Thus, the data generated in this study are relevant to cell lineages that are in an immature state and valid interpretations about networks underpinning their biology in a steady-state condition should not be made. The authors should either investigate whether some of the key findings made with cells at P7 hold true in the same cells in adult mice or revise their interpretations and conclusions to reflect that the findings are for cells in an immature developmental state. This includes revising the title and abstract to reflect the fact that spermatogonia in steady-state conditions of adult spermatogenesis were not investigated.*

We agree that this study focuses on juvenile mice and that our conclusions pertain specifically to that developmental stage. In response to this concern, we have revised the title to more accurately reflect the developmental stage of the cells studied. The revised title is: "Line 3: ZBTB16/PLZF regulates self-renewal and differentiation of juvenile spermatogonial stem cells through an extensive transcription factor-chromatin poising network."

We note that a considerable fraction of research on mouse spermatogenesis has been conducted using juvenile mice, due to the synchronous first wave of spermatogenesis - and has yielded important information for the field. Observations made in juvenile mice are often relevant during the adult stage, although we recognize the importance of accurately framing the results and their limitations. To address this, we have clarified the scope and conclusions of our findings (juvenile mice) in several places throughout the manuscript, ensuring that it is clear the study pertains to cells in an immature developmental state.

Author's Response to Reviewers' comments: Yi et al., and Cairns

We sincerely thank the reviewers for their constructive and insightful feedback on our manuscript. Your valuable comments have greatly guided our revisions, resulting in significant improvements to the clarity and depth of the study. Below, we provide a point-by-point response to each of the reviewers' comments. The changes reflect these considerations and have further strengthened the overall manuscript.

Authors' response to Reviewers' Comments:

Reviewer #1: The authors have thoughtfully addressed most of my concerns and have substantially improved both the clarity and content of the paper. Most notably, they have incorporated new analysis of the PRO-seq data from Kaye et al. 2024 as well as new ChIP-seq data for TBP and PolII that shows paused PolII at most Typical genes and supports the model of a pre-established open chromatin network for pre-meiotic and meiotic genes. This is a significant addition that improves the study, and provides further support for functional segregation of Typical and Atypical promoters related to a pre-activation state in spermatogonia. In addition, they have appropriately adjusted claims that were based on in vitro MASC and mGSC data and rewritten the results to better clarify where each dataset comes from. This section is now much clearer and more scientifically accurate. They have also toned down the language for the claim about SSAs promoting transcription on the poised network, allowing for a better supported model. All my minor concerns have been adequately addressed. Overall, the paper now provides a generally well-supported conceptual advance. I have a few remaining concerns:

1) The major remaining issue is that all the ZBTB16 KO data is still largely negative: the phenotype is mild, there is little effect on SALL4/SOX3 binding, and the effect on gene expression is much less than the widespread ZBTB16 binding would imply and not closely linked to binding sites at the functional level. The authors explain this as being based on redundancy between ZBTB16, SALL4, and SOX3. This is plausible, but is not directly tested (e.g. by double or triple KO), and the fact remains that there is just not much in this study to support a functional role of ZBTB16. In addition, the data remains fundamentally correlative. However, I agree that the very high correlation in binding profiles and correlation with PolII states is striking, and the study overall offers a conceptual advance. I suggest that the issue of experimental support for a functional role for ZBTB16 should be acknowledged and addressed in the Discussion section.

We agree with the reviewer that the null phenotype is modest at the gene expression level. However, we do observe an impact on testis weight, cell cycle and spermatogonia number, consistent with the long-term maintenance function observed by others. Therefore, we have adjusted our claim to now state, " Notably, the number CCND1⁺ uSPG and dSPG significantly decreased after P14 in *Zbtb16* null mice¹⁰, which corresponds with the reduction in testis size observed from P14 onwards (Fig.3d,e and Extended data Fig.8d,e), which may partly underlie ZBTB16's role in supporting long-term uSPG maintenance^{7,10,11}.on Lines 454-457. This updated phrasing acknowledges the evidence while tempering our previous conclusion.

2) The authors indicated that the relationship to H3K27me3 is not discussed because it is complicated, and was therefore excluded due to space considerations. However, since the data is included in the paper it should at least be briefly discussed, or else the data should be left out. Low overlap with H3K27me3 is mentioned briefly in line 117, but the relationship should also be mentioned in the results section related to Fig 5d.

Thank you for pointing this out. We have added a brief discussion to address the role of H3K27me3 in relation to our findings on Line 347-351. Notably, H3K27me3 is highly enriched at

meiotic, spermiogenic, and a subset of mitotic genes at Typical promoters (Fig. 5d and Extended Data Fig. 11j). Similar enrichment has been observed at particular loci within ES cells, where H3K27me3 is correlated with active transcription at a subset of promoters, rather than repression (Young et al., 2011, PMID: 21652639). These findings suggest that H3K27me3 alone may not be sufficient for robust gene repression, indicating a more nuanced, context-dependent role in gene regulation.

3) *In Figure 4h, why are the SOX3 ChIP-seq tracks from published uSPG dataset (GSE146706) and this paper's dataset so different? There appears to be no signal from the published dataset.*

Thank you for highlighting this issue. The quality of ChIP-seq data is often evaluated using the FRiP (Fraction of Reads in Peaks) score, which measures the proportion of reads falling within significant peaks. According to the ENCODE guidelines, a FRiP score above 1% indicates acceptable quality (Landt et al., 2012, PMID: 22955991). In our dataset, the SOX3 ChIP-seq data exceeds 5%, signifying high-quality data. In contrast, the published dataset (GSE146706) has a FRiP score below 0.01%, which suggests poor data quality and explains the observed discrepancies in signal.

SOX3 ChIP	This work		GSE146706		
Replicates	Replicate 1	Replicate 2	Replicate 1	Replicate 2	Replicate 3
FRiP (%)	12.6734953	5.89851613	0.00953895	0.00641411	0.00988561

Reviewer Note: Table 1. ChIP-seq Data Quality: SOX3 FRiP Scores Comparison

4) *In the model diagram (Figure 7b), the right-hand side with spermatozoa should be removed, since there is no data nor analysis of spermatozoa in this study.*

Thank you for pointing out that we did not link the spermatozoa data to our Extended Data. We have added a description addressing this on Lines 351-356. The text reads: "Additionally, Atypical genes such as those encoding protamine, gained H3K4me3 in pachytene spermatocytes (along with many Typical genes), and retained an active histone modification profile with both H3K4me3 and H3K27ac marks in mature sperm. In contrast, Atypical genes that acquired H3K4me3 and became transcriptionally active in round spermatids subsequently lost these active histone modifications in mature sperm (Extended Data Fig. 11j)."

These features are depicted in the model figure, along with features in published work.

5) *Typo in line 474: promoters à promoters.*

Thanks - corrected.

Reviewer #2: *Authors have addressed most of my initial concerns in the revised version. However, the information provided in lines 734-740 does not fully suffice to assess the validity of the data. The rebuttal letter and the methodological section mention an accompanying paper for the description of the HiC methods, but each paper should be able to stand on its own. Therefore, I recommend the authors to extend their description of the methods section.*

Thank you for your feedback. We have revised the manuscript to include a more detailed description of the Hi-C methods within the Methods section (Lines 753-768).

Reviewer #3: *In this revised version the authors have addressed most but not all the concerns raised in the first round of review. I would like to again suggest that the authors reconsider accuracy of some of the statements that have been made.*

1) *Please reconsider dropping mGSC from the manuscript and refer to the cells as primary cultures of spermatogonia; this does not require coming up with a new acronym, just describe them for what they are. It is appreciated that the authors did not start the misleading trend of labeling the cultured cells as mGSCs but sticking with it perpetuates the problem.*

We understand your concern and have decided to indeed drop the mGSC and adjusting the description on Line 39-40. However, we often need to refer to the data from the papers that use the mGSCs nomenclature to compare with our *in vivo* uSPG. Therefore, we have introduced a new acronym, PC-uSPG (Primary Cultures of Undifferentiated Spermatogonia), for clarity and ease of reference both for readers and ourselves.

2) *I still question statements about the role of ZBTB16 in ensuring long-term maintenance of undifferentiated spermatogonia and that its absence results in gradual depletion of the population in a significant subset of seminiferous tubules. Experimental evidence in previous studies referenced by the authors that a significant portion of seminiferous tubules are depleted of the undifferentiated spermatogonial population in *Zbtb16* knockout mice is not very convincing. I reiterate that those studies showed that only a minor portion of seminiferous tubule cross-sections seem to have a depletion of the spermatogonial population. Neither Costoya et al., 2004 nor Buaas et al., 2004 quantified seminiferous tubules lacking spermatogonia, rather images of testis cross-sections were provided, but Sharma et al., 2019 (PMID: 31149899) did and reported ~10-15% of cross-sections lacked spermatogonia at 6 months of age. The significance of a degree of depletion in advanced age that is <25% of the seminiferous tubules is debatable. Note that disrupted spermatogenesis may or may not be due to problems with the undifferentiated spermatogonial population and is therefore not a definitive indicator of impaired long-term maintenance. In addition, the transplantation analysis performed in previous studies with *Zbtb16* knockout cells are fraught with experimental deficiencies including use of cross-sectional histology to assess recipient seminiferous tubules which cannot accurately account for potential donor-derived spermatogonial colonization throughout the tissue mass. For these reasons, I again encourage the authors to temper statements about the functional importance of ZBTB16 in maintenance of the undifferentiated spermatogonial population.*

It is true that this phenotype is incompletely penetrant and could be multi-factorial. After thoroughly reviewing the previous studies again (see below) in relation to our data, we believe that the reported role (by others) of ZBTB16 in uSPG maintenance is a valid interpretation, and here is some of our reasoning.

(a) Cyclin D1 (CCND1) marks both undifferentiated and differentiating spermatogonia (Extended Data Fig. 9c).

(b) CCND1⁺ SPG numbers significantly dropped after P14 in *Zbtb16* null mice (Costoya et al., 2004).

(c) This corresponded with a reduction in testis size from P14 in *Zbtb16* null mice (Fig. 3ef and Extended Data Fig. 8de).

(d) Testes halted growth at P56, then shrank by P70 (Extended Data Fig. 8de), consistent with germ cell loss with aging.

(e) *Zbtb16*-deficient early uSPG showed faster cell cycling, impairing long-term maintenance (Sharma et al., 2019 and Fig. 3d).

(f) ZBTB16 directly activates critical genes like *T (Brachyury)*, underscoring a regulatory role in uSPG.

(g) Regarding your concern about sample size in Buaas et al., 2004 and Costoya et al., 2004, we examined the methods sections which reported an analysis of 289 histological sections and ~200 tubules in their transplantation studies. They also included controls and consistent methodologies, which provides a measure of confidence in their conclusions. While more data is always beneficial, these numbers appear sufficient for meaningful conclusions, aligning with other independent findings.

Thus, our interpretation of the collective data for ZBTB16 aligns with a moderate and non-essential role that involves, at least in part, maintaining uSPG long-term.